# Soil microbiomes show consistent and predictable responses to extreme events

Christopher G. Knight[1,23✉], Océane Nicolitch[1,23], Rob I. Griffiths[2,3✉], Tim Goodall[3], Briony Jones[4], Carolin Weser[1], Holly Langridge[1], John Davison[5], Ariane Dellavalle[6,7], Nico Eisenhauer[8,9], Konstantin B. Gongalsky[10], Andrew Hector[11], Emma Jardine[11,12], Paul Kardol[13,14], Fernando T. Maestre[15], Martin Schädler[8,16], Marina Semchenko[1,5], Carly Stevens[17], Maria A. Tsiafouli[18], Oddur Vilhelmsson[19,20], Wolfgang Wanek[21] & Franciska T. de Vries[1,22,23✉]

Increasing extreme climatic events threaten the functioning of terrestrial ecosystems[1,2]. Because soil microbes govern key biogeochemical processes, understanding their response to climate extremes is crucial in predicting the consequences for ecosystem functioning[3,4]. Here we subjected soils from 30 grasslands across Europe to four contrasting extreme climatic events under common controlled conditions (drought, flood, freezing and heat), and compared the response of soil microbial communities and their functioning with those of undisturbed soils. Soil microbiomes exhibited a small, but highly consistent and phylogenetically conserved, response under the imposed extreme events. Heat treatment most strongly impacted soil microbiomes, enhancing dormancy and sporulation genes and decreasing metabolic versatility. Microbiome response to heat in particular could be predicted by local climatic conditions and soil properties, with soils that do not normally experience the extreme conditions being imposed being most vulnerable. Our results suggest that soil microbiomes from different climates share unified responses to extreme climatic events, but that predicting the extent of community change may require knowledge of the local microbiome. These findings advance our understanding of soil microbial responses to extreme events, and provide a first step for making general predictions about the impact of extreme climatic events on soil functioning.

Understanding the response of soil microbial communities to climate extremes, such as droughts, floods and temperature shifts, is crucial for understanding changes in ecosystem functioning and improving climate change projections[3,4]. Factors driving the microbial response to extreme events are multiple and involve complex interactions between intrinsic and extrinsic factors, including microbial community composition and diversity, soil properties and historical climate[5]. Microbial traits, such as osmolyte production, that protect microbial communities against drought stress may explain their response to extreme events, whereas other traits such as those involved in carbon and nutrient cycles—by physiological trade-offs linked with investment in response traits—may predict the consequences for soil functioning[6]. However, the impact of extreme climatic events depends not only on the resident microbial community but also on the soil system[7]: for example, soils with a high organic matter content can be more resistant to freezing[8]. Moreover, climatic properties may affect ecosystem functioning by alteration of soil microbial communities[9] and select for soil properties and microbial traits that dampen or amplify microbial responses to climatic disturbances[10,11]. Although interest in soil microbial community responses to extreme climatic events has surged in recent years, most of our knowledge comes from experiments focusing on a single soil or system[10,12–15], potentially exacerbating exaggeration bias[16]. Moreover, our understanding of microbial community response to drought far exceeds that for other disturbances such as floods, heatwaves or soil freezing, which are also increasing with climate change.

[1]Faculty of Science and Engineering, University of Manchester, Manchester, UK. [2]School of Natural Sciences, Bangor University, Bangor, UK. [3]UK Centre for Ecology and Hydrology (UKCEH), Wallingford, UK. [4]UK Centre for Ecology and Hydrology (UKCEH), Bangor, UK. [5]Institute of Ecology and Earth Sciences, University of Tartu, Tartu, Estonia. [6]Faculty of Biology, Medicine and Health, University of Manchester, Manchester, UK. [7]Faculty of Natural Sciences, Imperial College London, London, UK. [8]German Centre for Integrative Biodiversity Research (iDiv) Halle-Jena-Leipzig, Leipzig, Germany. [9]Institute of Biology, Leipzig University, Leipzig, Germany. [10]A.N. Severtsov Institute of Ecology and Evolution, Russian Academy of Sciences, Moscow, Russia. [11]Department of Biology, University of Oxford, Oxford, UK. [12]Animal and Plant Sciences Department, University of Sheffield, Sheffield, UK. [13]Department of Forest Mycology and Plant Pathology, Swedish University of Agricultural Sciences, Uppsala, Sweden. [14]Department of Forest Ecology and Management, Swedish University of Agricultural Sciences, Umeå, Sweden. [15]Biological and Environmental Science and Engineering Division, King Abdullah University of Science and Technology, Thuwal, Kingdom of Saudi Arabia. [16]Department of Community Ecology, Helmholtz Centre for Environmental Research, Leipzig-Halle, Germany. [17]Lancaster Environment Centre, Lancaster University, Lancaster, UK. [18]Department of Ecology, School of Biology, Aristotle University of Thessaloniki, Thessaloniki, Greece. [19]Natural Resource Sciences, University of Akureyri, Akureyri, Iceland. [20]BioMedical Center, University of Iceland, Reykjavík, Iceland. [21]Centre for Microbiology and Environmental Systems Science, University of Vienna, Vienna, Austria. [22]Institute for Biodiversity and Ecosystem Dynamics, University of Amsterdam, Amsterdam, the Netherlands. [23]These authors contributed equally: Christopher G. Knight, Océane Nicolitch, Franciska T. de Vries. ✉e-mail: chris.knight@manchester.ac.uk; robert.griffiths@bangor.ac.uk; f.t.devries@uva.nl

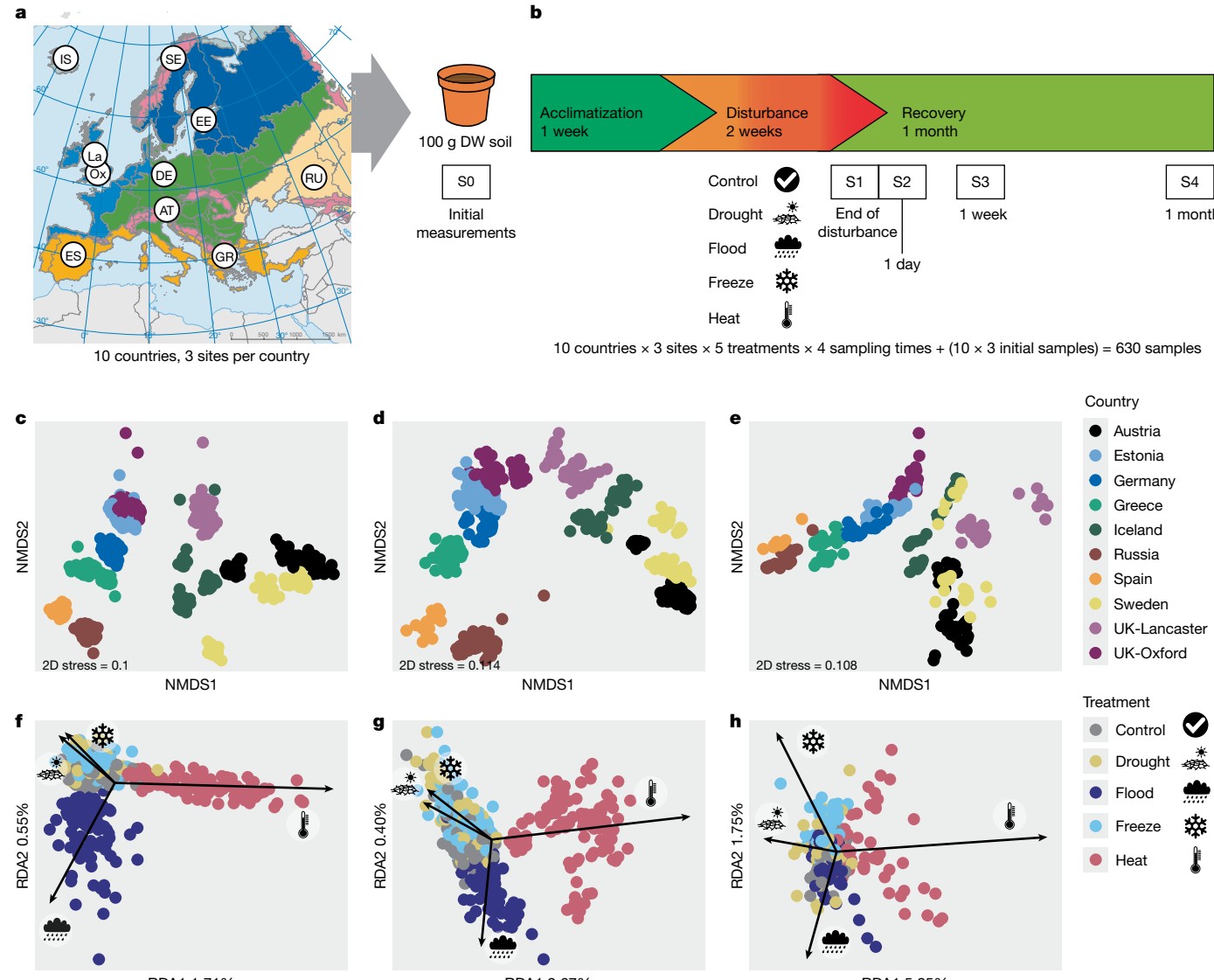

**Fig. 1 | Experimental design for imposing extreme climatic events and their effects on soils collected from across Europe. a,b**, Sampled site locations (**a**) and experimental set-up (**b**) for simulation of extreme climatic events. The 10 circles represent those countries where three replicate grassland sites within 11 km of each other were sampled, resulting in 30 sites in total. Sites represent the diversity of biogeographic regions present in Europe: alpine (AT, Austria), subarctic (SE, Sweden), Arctic (IS, Iceland), Atlantic (Ox, Oxford and La, Lancaster, both UK), boreal (EE, Estonia), continental (DE, Germany), Mediterranean (ES, Spain and GR, Greece) and steppe climate (RU, Russia). **c–h**, The simulated climate extremes consistently shift soil microbial communities in the same direction despite their contrasting composition. Non-metric multidimensional scaling (NMDS) ordinations of prokaryotic (*n* = 576) (**c**), fungal (*n* = 574) (**d**) and shotgun metagenome (*n* = 308) (**e**) communities, based on Bray–Curtis dissimilarities show that the origin of the sample (country and site) is the main driver of microbial community composition, followed by the type of disturbance and the time elapsed following the disturbance (Extended Data Table 1). The colour of points indicates the country of origin. Partial redundancy analysis (RDA) ordinations show disturbance effects on communities after controlling for site effects (black arrows). Prokaryotic (**f**), fungal (**g**) and shotgun metagenome (**h**) communities exhibit a consistent shift in response to individual disturbances. Percentage of variance explained, having conditioned on country and site, is given on the RDA axes; only the first two axes are shown. Total variance explained by all four constrained RDA axes is 2.7, 3.6 and 8.8% for prokaryotic, fungal and shotgun metagenomes, respectively. Conditional variance (country and site within country), as a proportion of total variance, was 71, 68 and 91% for prokaryotic, fungal and shotgun metagenomes, respectively. 2D, two-dimensional; DW, dry weight. Map in **a** adapted from ref. 45, European Environment Agency CC BY 4.0.

Here we sought to ascertain whether there is a unified microbial community response to extreme events across soil types and biogeographic regions. We hypothesized that different extreme climatic events shift soil microbial communities in distinct and consistent directions. Specifically, we expected similar disturbances to shift soil microbial communities in the same direction. For example, drought and soil freezing, and potentially also heat, cause osmotic stress[17,18] and may thus elicit a similar microbial response, whereas flooding would have an opposing effect. In addition, we hypothesized that local

climatic conditions select for soil microbial taxa exhibiting traits that allow them to cope with extreme climatic events regularly occurring in their climatic niche. Thus, for example, soil microbial communities from drylands should be more resistant to extreme drought through the selection of drought-resistance traits.

To test these hypotheses, we imposed 4 different climatic disturbances to soils collected from 30 European grasslands in 10 countries, covering all important European biogeographic zones (Fig. 1a). Grasslands occupy a wide range of soil types and climatic conditions, covering

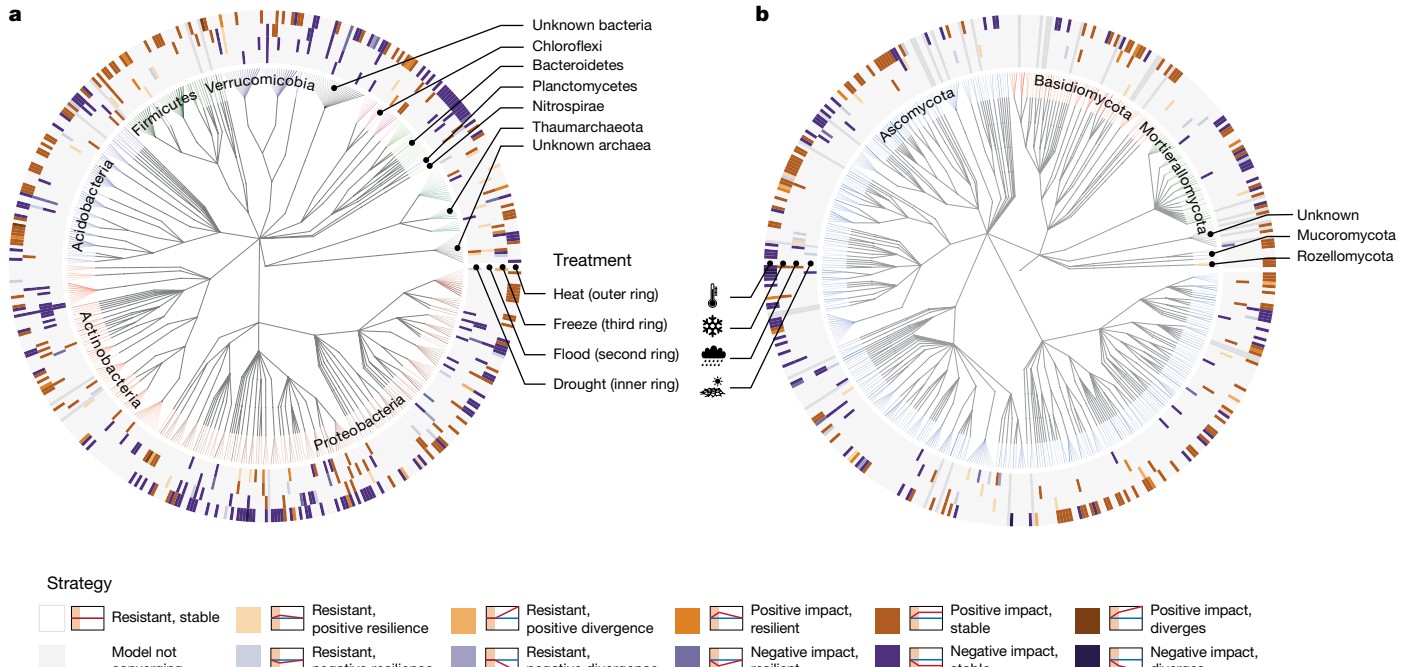

**Fig. 2 | Ecological resistance and resilience strategies associated with the 500 most abundant ASVs. a,b,** ASVs associated with prokaryotes (**a**) and fungi (**b**). The central tree indicates the taxonomy of the 500 most abundant ASVs, with tips coloured by phylum (more abundant ASVs are more intensely coloured). In the rings surrounding the tree (one per disturbance treatment), purple/blue colours indicate ASVs that perform significantly worse following disturbance ($P \leq 0.05$, two-tailed Wald test, relative to other organisms and to the control treatment as fitted by a linear mixed-effects model across all soils, treatments and samplings, without correction for multiple testing; $n = 548$ and 586 samples for bacteria and fungi, respectively). Orange colours indicate ASVs that perform relatively better following disturbance using the same criteria. The shade of colour indicates the dynamics of the response as shown in the key: the darkest shades indicate a statistically significant divergence from the control at the end of the disturbance, followed by a statistically significant change in the same direction over the following month (that is, not resilient). The palest shades indicate no significant change at the end of the disturbance, but a significant divergence over the following month. ASVs in which a model did not converge are indicated by pale grey tiles across all four perturbation rings.

40% of the Earth's surface (Fig. 1a and Supplementary Table 1). They are an important reservoir of biodiversity and provide many benefits to humans[19]. In controlled-climate cabinets, we subjected microcosms of each soil to either a 2 week drought ('drought': 10% of soil water-holding capacity (WHC), 18 °C), flooding event ('flood': 100% WHC, 18 °C), soil freezing ('freeze': 60% WHC, −20 °C) or heatwave ('heat': 60% WHC, 35 °C), alongside a control treatment that was maintained at constant moisture and temperature ('control': 60% WHC, 18 °C), followed by 4 weeks of recovery. These particular disturbances were chosen to represent extremes that are increasing with climate change[2] and are considered extreme across all our sampled environments (Methods), although not necessarily comparable in their severity. We assessed the response of soil microbial communities at the end of the disturbance (sampling 1, or S1) and 1 day (S2), 1 week (S3) and 4 weeks (S4) after ending the disturbance, by prokaryotic, fungal and shotgun metagenomic sequencing (for assessment of shifts in microbial functional genes). We assessed soil functioning by measuring microbial enzymatic activities, microbial ability to use multiple substrates and carbon and nitrogen pools and fluxes (Methods). In total, this experimental design resulted in 600 independent microcosms (Fig. 1a,b).

Prokaryotic, fungal and functional gene communities were strongly shaped by their origin (permutational multivariate analysis of variance (PERMANOVA) $R^2 = 0.72$, 0.74 and 0.68, respectively, for country and site main effects combined, $P \leq 0.001$ in all cases; Fig. 1c–e, Extended Data Table 1 and Extended Data Fig. 1a–c,g), reflecting the importance of soil and climatic factors in the distribution of soil microbial communities globally[20–22]. Despite the wide variation in microbial community composition, the responses of prokaryotic, fungal and functional gene community structure to the imposed extreme climatic events

were small but consistent among soils from diverse origins (Fig. 1f–h, Extended Data Table 1 and Extended Data Fig. 1; disturbance type and sampling time combined, including their interaction, PERMANOVA $R^2 = 0.017$, 0.018 and 0.018 for prokaryotes, fungi and metagenome, respectively). Whereas the proportion of total variance explained by extreme event treatments was small at the global scale (due to distinctness of soil communities across sites), the disturbances explained 10–29, 12–29 and 19–64% of variance, respectively, in prokaryote community, fungal community and metagenome composition at the local scale (Extended Data Fig. 1h). With the inclusion of a higher number of countries in our study, the proportion of variance explained by the disturbances became lower and less variable among included countries (Extended Data Fig. 1h). Thus, there is a common response to the simulated extreme events that can be reliably quantified only when assessing a wide range of soils from different climates. As hypothesized, drought and freeze shifted communities largely in the same direction, demonstrating that drought-resistant microbes are also resistant to freezing, probably because drought and freeze both decrease soil water availability[17]. Increased soil water availability through flooding shifted these communities in the opposite direction (along axis 2; Fig. 1f–h), and heat treatment had the strongest effect on community composition across prokaryotes, fungi and the metagenome (along axis 1; Fig. 1f–h).

We modelled the response of all fungal and prokaryotic amplicon sequence variants (ASVs) to extreme climatic events. We measured resistance (that is, the ability to withstand a disturbance[23]) by displacement in relative abundance at the end of the disturbance (S1), and we measured resilience (that is, the ability to recover following a disturbance[23]) by the slope of the change over the following month (S2–S4; Methods). In line with recent findings from a field drought experiment

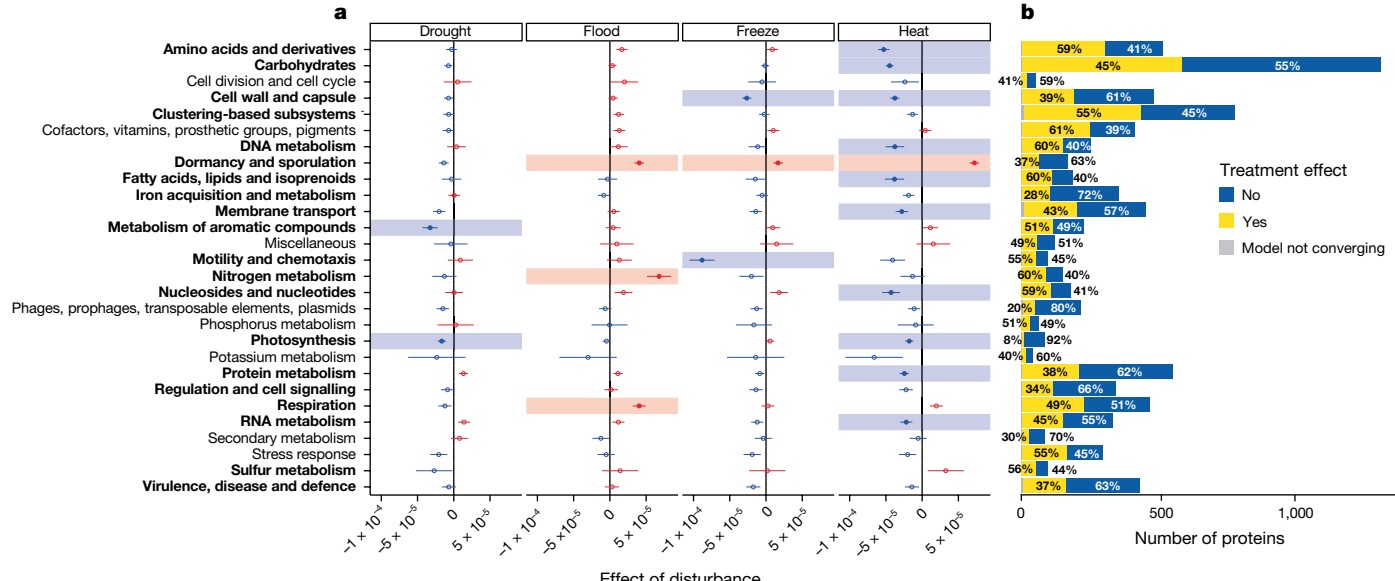

**Fig. 3 | Changes in abundance of protein functions with extreme climatic event disturbances. a,b,** Functional classifications relative to control in metagenomic samples. **a,** Effects of disturbance on each of the 28 highest-level functions; red indicates an increase in abundance following the perturbation given at the top of the column, blue a decrease. One linear mixed-effects model for the proportion of reads was fitted per function using data from S1 and S4 ($n = 280$ metagenomic samples; Methods). The change in proportion with disturbance, relative to control ($\pm$s.e.m.), as estimated by this model, is shown on the horizontal axis; functions with significant changes among treatments ($P \leq 0.05$, $F_{3,280}$; two-tailed test, following false discovery rate correction for multiple models on the analysis of variance (ANOVA) $P$ value of the relevant mixed-effects model, shown in bold). When a Dunnett's test of a particular disturbance versus control accounting for multiple testing within a mixed-effects model is significant ($P \leq 0.05$, two-tailed test), the point is filled and shown on a coloured background. **b,** Numbers (horizontal axis) and proportions (figures in bars) of individual proteins with and without significant effects of disturbance in each of the 28 highest-level functions ($P \leq 0.05$ following false discovery rate correction for multiple models). In a few cases a linear mixed-effects model did not converge for a particular protein (shown in grey).

across UK soil types and climates[24], ASVs that are fully resistant formed the largest group (that is, no significant difference between the relative abundance of ASVs in disturbed and control samples and no change in this difference over time (uncoloured in Fig. 2)). Among ASVs that showed significant changes either positively or negatively (orange/red and blue/purple, respectively, in Fig. 2), we identified ten different ecological response strategies (Fig. 2 and Extended Data Fig. 2). The most common strategy that diverged from control was an increase or decrease in relative abundance that then remained stable ('positive impact, stable' and 'negative impact, stable', respectively), which is in contrast with the aforementioned field experiment in which most dominant fungal and bacterial ASVs were resilient[24]. The distribution of these strategies across ASVs was notably similar across disturbances and between fungi and prokaryotes (Extended Data Fig. 2). The proportion of resistant ASVs was lowest in response to heat, confirming soil microbial community sensitivity to warming[25,26] (Extended Data Fig. 2). To evaluate whether the ASV response could be predicted from phylogenetic information, we calculated the phylogenetic conservatism of resistance and resilience (Extended Data Fig. 3). In both prokaryotic and fungal communities, resistance to flood and heat was more phylogenetically conserved than that to drought and freeze. Resilience to extreme climatic events was less conserved than resistance, particularly in fungi, and resistance to heat was more conserved in prokaryotes than in fungi. These findings are in contrast with previous work that shows little difference in phylogenetic conservation between microbial response to different global change drivers, and between fungi and bacteria[27,28]. Considering that traits relying on complex genetic systems are more deeply conserved phylogenetically[29], our results suggest that heat and flood resistance mechanisms might be more complex than those for other climate extremes.

We identified the functional genes responding to extreme climatic events based on shotgun metagenomic sequencing (Methods).

Overall, 46% of the total annotated genes differed in relative abundance between control and disturbed samples at the end of the disturbance (S1, 4,036 of 8,772 genes), with the proportion varying across functions (8–61% across the highest-level functional categories; Fig. 3b; further details of lower-level categories are shown in Supplementary Fig. 5). Relatively few genes showed a significant change over time (9.6%), but those that did tended to be resilient (Extended Data Fig. 4 and Supplementary Fig. 6). Functional gene abundances were strongly correlated with initial soil properties and climatic conditions, with a clear divide between wet and dry environments. In organic soils of environments with high precipitation, genes involved in nitrogen, potassium, aromatic compound and sulfur metabolism were more abundant, as were those related to phages, signalling, motility and virulence, disease and defence functions (Extended Data Fig. 5a). By contrast, dry, hot soils with high pH (Extended Data Fig. 5b) had a higher abundance of genes involved in dormancy and sporulation, phosphorus and protein metabolism, carbohydrate metabolism and cell division and cell cycle. These findings suggest a broad distinction between copiotrophic microbial strategies (organisms that preferentially metabolize labile soil organic carbon pools and exhibit high growth rates when resources are abundant, found in environments rich in resources[30]) in wet climates and oligotrophic strategies (living in environments with very low levels of resources[30]) in warm, dry climates[31].

Genes involved in dormancy and sporulation increased in relative abundance across flood, freeze and heat (Fig. 3a; change in proportion ranging from $7.21 \pm 0.67 \times 10^{-5}$ s.e.m. in heat to $1.7 \pm 0.67 \times 10^{-5}$ s.e.m. in freeze, $P \leq 0.0001$ and $P = 0.047$, respectively), suggesting community-level selection for these traits across a variety of stresses. Heat treatment elicited the greatest number of significant functional changes (11 at $P < 0.05$, Dunnett's test), with 10 out of 11 of those changes being negative (Fig. 3a). Consistent with this impact of heat, for bacteria

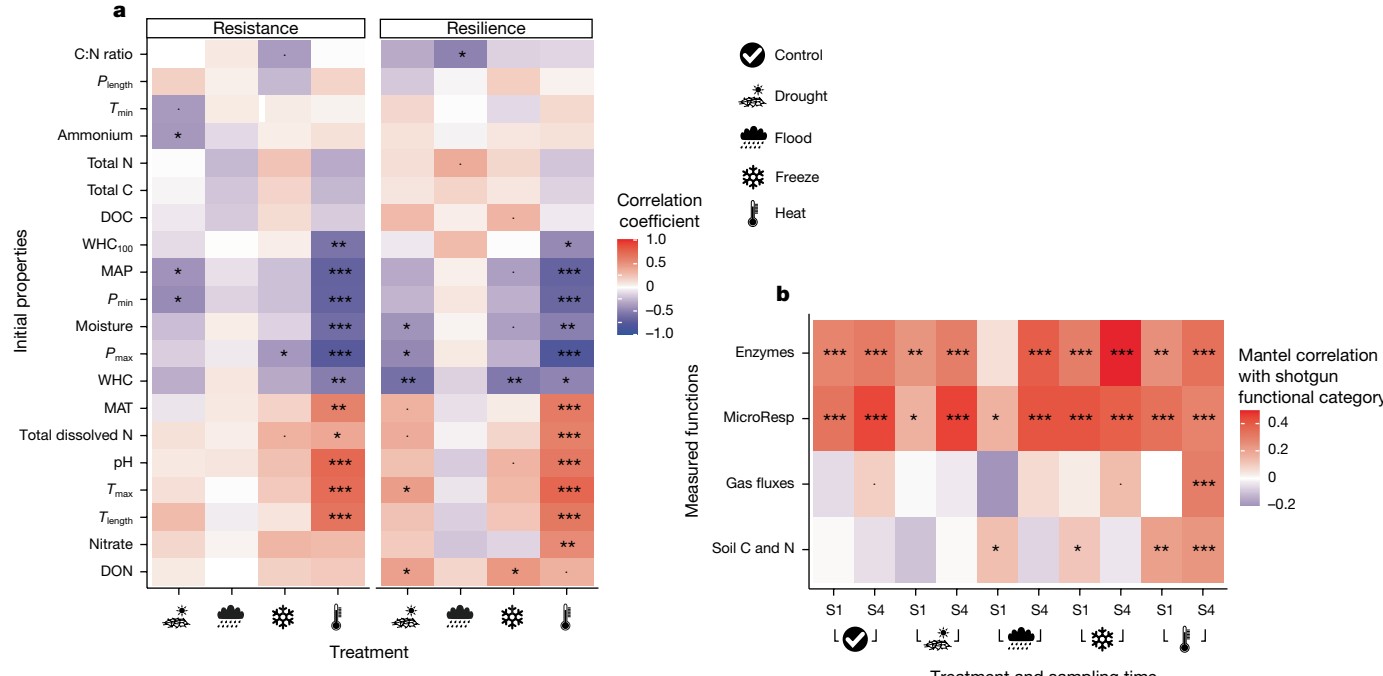

**Fig. 4 | Correlations of initial soil and climatic properties with biofunctional responses to perturbation. a**, Rank correlations between initial properties and the resistance (left) and resilience (right) of community function following disturbance. Resistance and resilience are quantified as the negative Bray–Curtis dissimilarity of metagenome-derived functional (protein-level) matrices between control and treatment directly following either the disturbance (S1) or 1 month of recovery (S4), respectively. The order of initial properties on the y axis is determined by complete-linkage clustering according to their correlations with resistance–resilience across perturbations. **b**, Relationship of soil functional measurements (enzyme activities, microbial substrate use, gas fluxes and soil C and N concentrations) and the relative abundance of finest-scale functional categories (proteins) from the metagenomes. Distances among soil samples were calculated for both the relative abundances of all functional categories in the metagenomes and each of the four classes of functional soil measurements. For each treatment and time point, coloured tiles show the rank correlation between these two distance matrices, with asterisks indicating the significance (two-tailed Mantel test, $n_{perm}$ = 999; \*\*\*$P \leq 0.001$, \*\*$P \leq 0.01$, \*$P \leq 0.05$, ·$P \leq 0.1$, uncorrected for multiple testing). Note the different colour scales in **a** and **b**. Ammonium, plant available ammonium concentration; C:N ratio, soil carbon:nitrogen ratio; DOC, dissolved organic carbon concentration; DON, dissolved organic nitrogen concentration; MAP, mean annual precipitation; MAT, mean annual temperature; Nitrate, plant available nitrate concentration; $P_{length}$, difference between $P_{max}$ and $P_{min}$; $P_{max}$, annual precipitation maximum; $P_{min}$, annual precipitation minimum; $T_{length}$, difference between $T_{max}$ and $T_{min}$; $T_{max}$, maximum annual temperature; $T_{min}$, minimum annual temperature; Total C, total soil carbon content; Total dissolved N, total dissolved nitrogen concentration; Total N, total soil nitrogen content; WHC, soil moisture content expressed as percentage of moisture content at 100% water-holding capacity; $WHC_{100}$, soil moisture content at 100% WHC.

we find that both realized growth (origin:terminus ratio; Methods) and capacity for growth (measured by the estimated copy number of the 16S ribosomal RNA operon, which is highly variable among organisms and related to maximum growth rate[32–34]), decreased relative to control in heat at the end of the disturbance (S1) (Extended Data Fig. 6b,c). Subsequently, during recovery, organisms with greater realized growth increased (Extended Data Fig. 9b; $F_{1,187}$ = 12, $P$ = 0.00054; sampling time effect in linear mixed-effects model), potentially facilitated by the greater capacity for dormancy and sporulation immediately following heat (Fig. 3a). The greatest growth responses following heat occurred in copiotrophic communities from cold, acidic environments (eight of the ten most actively growing communities were from Austria and Sweden; Extended Data Fig. 6a). There was also a notable relationship of bacterial response to heat with that of fungi—the more fungal communities were impacted, the more positive was the bacterial growth response (slope, 0.71 ± 0.23 s.e.m., $t_{187}$ = 3.1, $P$ = 0.0028, all measured relative to control, Wald test, mixed-effects model). This increased bacterial growth with more impacted fungal communities was not found in other disturbances (interaction between fungal impact and treatment $F_{3,187}$ = 4.1, $P$ = 0.0073; Extended Data Fig. 6a). Together with a decrease in carbohydrate metabolism (change in proportion, 2.5 ± 0.6 × 10$^{-5}$, $P$ = 2.0 × 10$^{-15}$; Fig. 3a), which suggests lower metabolic versatility, this is consistent with a relative increase in copiotrophs following heat[30,31]. Thus, whereas both heat and flood favour organisms

with a 'boom-and-bust' strategy, and both initially cause a bust (reduction in growth; Extended Data Fig. 6b,c), it is heat that enables a subsequent bacterial boom specifically where the fungi are most disrupted (Extended Data Fig. 6a), suggesting a release from competition or antibiosis[35,36].

In line with our hypothesis, the magnitude of microbial community response to extreme climatic events depended on their original environment. We calculated the resistance and resilience of metagenome responses to extreme climatic events (Methods), and found that resistance and resilience to heat, drought and, to a lesser extent, freeze were lower in communities from cold and wet climates, whereas resistance and resilience to heat and, to a lesser extent, drought were higher in communities from warmer and drier climates (Fig. 4a). Moreover, we were able to construct a predictive model that explains 58 ± 4% of variance (mean ± s.d.) in the response across perturbations in unseen soils (by cross-validation) based only on initial soil properties and climate (Extended Data Fig. 7a; the addition of soil and country effects to this model did not improve its fit). Model prediction is most effective for heat perturbation, where soils from warmer than average climates resisted heat treatment increasingly well, but freeze increasingly poorly, with higher mean annual temperatures of the site (Extended Data Fig. 7b). A similarly distinct response to flood and drought was associated with soil moisture (Extended Data Fig. 7c), despite prediction being much less effective for these perturbations (Extended

Data Fig. 7a). As with the effect of our experimental perturbations on community composition (Extended Data Fig. 3), this model relies on the inclusion of a broad range of countries representing different biogeographic regions (Fig. 1a): prediction of resistance in whole unseen countries had relatively poor explanatory power, restricted to heat perturbation and was itself more variable ($26 \pm 20\%$ mean $\pm$ s.d.; Extended Data Fig. 7d). These results support the idea that, in extreme environments, deterministic processes are more important than stochastic processes in determining microbial community composition[37–39], and are in line with studies finding reduced microbial response to either a second disturbance or a disturbance similar to their native climate[10,13,40]. They highlight that predicting the magnitude of microbial response to extreme events requires knowledge of local microbiomes. We also found that communities more resistant and resilient to heat had a higher abundance of genes involved in dormancy and sporulation, as well as carbohydrate, protein, amino acid, phosphorus and DNA metabolism, but lower abundance of nitrogen and potassium metabolism (Extended Data Fig. 8), again emphasizing the different response to heat of communities from cold and wet environments and those from warm, dry climates[31].

Our imposed extreme climatic events strongly affected fluxes of both greenhouse gases ($CO_2$, $CH_4$ and $N_2O$) and soil C and N availability, but less so enzyme activities and the microbial ability to use different substrates, which were mainly driven by country of origin (Extended Data Table 1 and Supplementary Figs. 1–3). Importantly, we found a direct link between metagenome structure and soil functioning: when metagenomes were more similar, measured soil functions were typically more similar within the end-of-disturbance sampling (S1) and sampling following 4 weeks of recovery (S4) in each treatment. Specifically, in all treatments, both at S1 and S4, metagenome composition was strongly correlated with enzyme activities, and with substrate-induced respiration—that is, the ability to use different carbon substrates (Fig. 4b). On further inspection, we found that genes involved in photosynthesis, virulence and phages were consistently driving these correlations (Extended Data Fig. 9). However, these correlations do not allow for partitioning between the direct effects of soil microbial communities on soil processes and the feedbacks of altered soil nutrient and C availabilities to microbial community composition and process rates, which will both have a role in longer-term soil response to altered climatic conditions[41]. For example, alterations in photosynthesis genes (Fig. 3) may directly drive soil processes whereas the impact of alterations in phage-related genes may be indirect, which warrants further investigation[42].

Together, our results indicate that contrasting extreme climatic events shift soil microbial communities in distinct, predictable directions. The response of functional microbial community composition could be predicted by both the type of disturbance and soil and climatic properties, and was linked to enzyme activities and the ability to use different carbon sources, suggesting that extreme events may have consequences for soil carbon storage through their impacts on soil microbial communities[6,43]. Our results show that extreme climatic events will probably affect soil functioning most strongly in regions that do not normally experience similar extreme conditions, which are also those that are warming most rapidly[44]. They also emphasize that information on local soil properties and climate is key to predicting the impacts of climate extremes on soil functioning. These insights into the microbial community and environmental attributes that determine soil microbial responses to extreme climatic events provide a first step for making general predictions of the impact of extreme climatic events on soil functioning.

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

## Methods

### Study sites and soil sampling

We limited our study to natural or extensively managed grasslands across Europe, aiming to cover the widest range possible of climates and soil types. Our grassland network consisted of ten locations (referred to as countries) covering most biogeographical regions in Europe[19] (Fig. 1a,b and Supplementary Table 1): alpine (AT, Austria), subarctic (SE, Sweden), Arctic (IS, Iceland), Atlantic (UK-Ox, Oxford and UK-La, Lancaster, both UK), boreal (EE, Estonia), continental (DE, Germany), Mediterranean (ES, Spain and GR, Greece) and Steppic (RU, Russia). Climatic data associated with the collection sites were obtained from the WorldClim database in April 2018 (ref. 46), based on observations recorded between 1960 and 1990 at 2.5 min resolution, using the coordinates of the sites. We captured a wide range of MAP values, varying from 223 mm in Russia to 1,383 mm in Austria, whereas MAT varied between −2 °C in Sweden and 14.5 °C in Spain. Seasonal variability varied depending on the location, with Spain having the highest precipitation seasonality (38 mm) and UK-Oxford the lowest (14 mm). Russia varied the most in regard to temperature seasonality (38.5 °C) and Iceland the least (14 °C). We aimed to collect soil samples at peak microbial biomass, which is close to peak vegetation biomass, when the average temperature was the closest to 18 °C—that is, in spring for southern locations and following snow melt, in summer, for northern locations. In May 2018 we collected soil from Russia, Greece, Spain and Estonia, followed by Germany and Oxford in June, Austria and Iceland in July and Lancaster and Sweden in August.

Three replicate sites per country were selected to encompass a wide range of soil types present in European grasslands, for a total of 30 replicate sites, closely situated within countries (0.05–11.76 km apart). According to the World Reference Base[47] (https://soilgrids.org), our study included seven soil type categories, most of these belonging to the Haplic Cambisols (16), followed by Haplic Podzols (6), Haplic Kastanozems (3), Haplic Luvisols (2), Petric Calcisols (1), Rendzic Leptosols (1) and Haplic Gleysols (1). The most acidic soils were collected in Austria (pH 4.73), with those from Spain having the highest pH (7.89). Within each replicate site, seven $1 \times 1 \, m^2$ plots were arranged at least 5 m apart. We removed vegetation and stones and randomly sampled four soil cores from each plot with a 3-cm-diameter soil corer (0–15 cm). We carefully sterilized the collecting tools with ethanol 70% and distilled water and sampled the soil with gloves to avoid cross-contamination. Soil cores were pooled and sieved at 4 mm to form one homogenized composite soil sample per replicate site. Soil samples were transported in cold boxes to the laboratory and stored at 4 °C before analysis or to the establishment of microcosms. We measured WHC by immersing 100 g of fresh soil in water overnight and measuring the mass of water at saturation (100% WHC) following 4 h of drainage at 18 °C. We determined soil moisture content following drying at 105 °C for 48 h. This information was used to calculate the amount of fresh soil corresponding to 100 g of dry soil, and to maintain the moisture of the microcosms during the experiment.

### Microcosm procedure and harvest

For each replicate site, fresh soil was thoroughly mixed and dispatched into 20 pots (240 ml, 7.6 cm diameter across the top). We subsampled 20 and 5 g of soil for instant determination of initial soil moisture and nutrient concentrations, and stored 4 g of soil at −20 °C for DNA-based analyses. Considering the variability in soil density, WHC or initial moisture of soil derived from the different sites, the amount of soil per microcosm was standardized by dry weight (100 g dry weight per pot). Microcosms were placed for 7 days in a Percival AR-66L2 growth chamber (CLF PlantClimatics), open to the air, at 18 °C to allow acclimatization of microbial communities and adjustment of soil moisture to 60% WHC with milliQ water. Following this period, microcosms were randomly divided into five treatments. Temperature and moisture parameters were manipulated as follows for 14 days: control (18 °C, 60% WHC), drought (18 °C, 10% WHC), flood (18 °C, 100% WHC), freeze (−20 °C, 60% WHC) and heat (35 °C, 60% WHC). Application of these treatments for 2 weeks imposed extreme disturbance on soils from all locations used—for example, temperatures of −20 °C do not commonly occur even in Arctic soils[48] whereas 35 °C is rarely exceeded in semiarid systems and during extreme heatwaves[49,50]. All treatments were placed in the same growth chamber and grouped by replicate site, where control exhibited the same environmental parameters as the acclimatization period: for drought we stopped watering the pots when moisture reached 10% WHC (Supplementary Fig. 4) and, for flood, microcosms were placed in plastic cups and submerged 1 cm above the soil surface. Freeze and heat samples were placed in a separate freezer and growth chamber, respectively, for the duration of the disturbance. At the end of the disturbance, one set of microcosms was harvested (see below) and the remainder returned to the main growth chamber, randomized within block (replicate site). Moisture content for drought was brought back to 60% WHC and, for flood, microcosms were drained until they reached the appropriate moisture content (Supplementary Fig. 4). To maintain moisture content at 60% WHC, the respective microcosms were watered by weight every 2 days with milliQ water. We sacrificed one microcosm per treatment and per replicate site at four sampling times: end of disturbance (S1) and at 1 (S2), 8 (S3) and 26 (S4) days following the end of the disturbance ($5 \times 3 = 15$ pots per harvest). This resulted in a total of 600 microcosms: 10 countries × 3 replicate sites × 5 treatments × 4 sampling times. Ten Russian samples at S2 were excluded from the study because of insufficient soil amount.

At each sampling time, we measured the emission of greenhouse gases in the dark by placing the microcosms into 500 ml airtight Kilner jars covered in aluminium foil. The jars were sealed and 10 ml gas samples taken sequentially in the headspace with a syringe at 0, 10, 20 and 30 min. Gas samples were transferred to 20 ml pre-evacuated glass vials, and $CO_2$, $N_2O$ and $CH_4$ concentrations analysed by gas chromatography (GC Agilent 7890B). All gas concentrations were converted into fluxes based on soil dry weight (microgram of gas per gram of dry soil per hour). Fluxes were calculated considering the variation in both headspace volume (soils have different bulk densities depending on their origin, and headspace gas volume decreased by gas sampling) and temperature at each measurement. Microcosms were then thoroughly homogenized, with immediate subsampling of 20 and 5 g to determine soil moisture and nutrient concentrations. In addition, we stored 52 g ($40 + (6 \times 2 \, g)$) of soil at −20 °C for DNA-based analyses, MicroResp and enzymatic profiles. DNA was extracted within 3 months following freezing.

### Soil analyses and microbial activity

Dissolved carbon and nitrogen in initial soil samples ($n = 30$) and all microcosms ($n = 590$, $600 − 10$ Russian samples as described above) were extracted from 5 g of soil (fresh weight) in 35 ml of milliQ water, shaken for 10 min at 220 rpm and centrifuged for 15 min at 4,000 rpm. Water extracts were filtered through cellulose papers (Whatman no. 1, 11 μm) and syringe filters (0.45 μm), and filtrates were stored at 4 °C for a maximum of 15 days. Filtrates were analysed on a Seal AutoAnalyzer3 Segmented Flow Multi-chemistry analyser (Mequon) to measure dissolved nitrogen. Dissolved organic nitrogen was obtained by subtracting total dissolved inorganic N ($NH_4^+$-N and $NO_3^-$-N) from total dissolved N. Dissolved organic carbon was obtained from the same filtrates analysed on a 5000A TOC analyser (Shimadzu). Soil pH was measured on initial samples (30 samples) by the water method using a volume ratio of 2:5 fresh soil:milliQ water. Total soil C and N were determined by high-temperature combustion (150 mg of oven-dried, ground subsamples), followed by thermoconductometric detection on a Vario EL cube (Elementar Analysensysteme).

Substrate-induced respiration profiles were measured by the MicroResp method[51] on soil samples collected at S1 and S4 (280 samples).

Soils were defrosted overnight at 4 °C. We used the microplate filler device (300 µl) to standardize the volume of soil added in each well of the deep-well microplate (1.2 ml) and recorded the mass of soil added. We prepared detection microplates with a pH indicator gel to estimate the amount of $CO_2$ produced by a colourimetric method (3% agar, 2.5 mM $NaHCO_3$, 150 mM KCl and 12.5 µg ml$^{-1}$ cresol red). We dispensed 25 µl of eight substrate solutions relevant to soil microbial activity separately to the wells: water, glucose, fructose, carboxymethyl cellulose, citric acid, malic acid, alanine or asparagine. Carbon substrates were prepared to a concentration of 12 mg C g$^{-1}$ soil/water, corresponding on average to 30 mg glucose g$^{-1}$ soil/water solution. Initial absorbance of the detection plates was determined at 570 nm on a Clario Star microplate reader. We incubated the sealed system deep-well microplate/detection plate for 6 h at 25 °C in the dark before measurement of final absorbance. Respiration rates (microgram of $CO_2$-C per gram of dry soil per hour) were calculated according to ref. 51.

Enzymatic activity assays were performed on soil samples at S1 and S4. Microbial potential enzymatic activities were assessed for acetyl-lesterase, β-glucosidase, phosphatase and leucine-aminopeptidase, with methylumbelliferyl- and 7-amino-4-methylcoumarin-conjugated substrates (Sigma-Aldrich), using the protocol described in ref. 52. In brief, a total of 1.5 g of frozen soil was mixed with 20 ml of milliQ water and shaken for 20 min at 400 rpm. We added 30 µl of soil solution to a microplate containing 170 µl of substrate solution at 300 mM, and incubated it at 28 °C for 3 h.

## DNA extraction and sequence processing

We extracted total DNA from 250 mg of frozen soil from the initial samples ($n = 30$) and microcosm samples ($n = 600 - 10$) using the DNeasy PowerSoil Pro Kit, following the manufacturer's instructions (Qiagen). Tubes were vortexed for 10 min at 1,200 rpm on a FastPrep-96 instrument (MP Biomedicals) at maximum speed, and DNA was eluted in 80 µl of the elution buffer. DNA integrity and quality were validated by running DNA samples on a 1.5% agarose gel at 75 V for 55 min, and by checking 280:260 absorbance ratios with the Clario Star microplate reader (BMG LABTECH). We quantified DNA concentrations by fluorimetry using the Quant-iT dsDNA Broad-Range Assay Kit (Life technologies) with the Clario Star microplate reader. DNA solutions were split among three tubes and stored at −20 °C until use for whole-metagenomic sequencing and prokaryotic and fungal metabarcoding.

Prokaryotic and fungal community composition of the initial and microcosm samples at all sampling times ($n = 620$) was assessed by sequencing the V4–5 region of 16S rRNA genes using the primers 515 forward GTGYCAGCMGCCGCGGTAA and 806 reverse GGACTACN VGGGTWTCTAAT[53], and established primers fITS7 (GTGARTCATCG AATCTTTG) and ITS4 (TCCTCCGCTTATTGATATGC) coding the ITS2 region[54], respectively. We followed the PCR protocols of the Earth Microbiome Project[55], and sequencing was performed using a two-step Nextera approach on the Illumina MiSeq platform with V3 chemistry (Illumina). We used the DADA2 v.1.24 pipeline[56] in R to trim, quality-filter denoise and dereplicate the sequences, for generation of ASV tables and to assign taxonomies. The UNITE dynamic database, released on 2 February 2019, and SILVA SSU r132, released in March 2018, were used for fungal and bacterial taxonomic assignment, respectively. Median read numbers were 32,200 (interquartile range (IQR) 25,631–38,508) and 38,410 (IQR 31,831044,246) for 16S and ITS, respectively. Two replicate sites from Spain (sites 1 and 3; $n = 42$) plus two further individual samples were excluded from analyses due to low DNA yield and poor recovery of reads, particularly for the prokaryotic amplicons, leaving 574 samples in total.

To determine the functional gene composition of our initial samples and our microcosms at S1 and S4, the whole metagenome was sequenced by the Centre of Genomic Research (28 initial samples (ES1 and ES3 excluded) + 28 replicate sites × 2 sampling times ×

5 treatments = 308 samples). The Illumina unamplified fragment library was prepared using the TruSeq PCR-free kit (350-base-pair (bp) inserts) and shotgun sequenced in four lanes of the NovaSeq platform using S4 chemistry (2 × 150 bp paired-end). Illumina adaptor sequences were detected and removed with Cutadapt (v.1.2.1), before trimming with Sickle 1.200 using a minimum window quality score of 20 and with exclusion of reads shorter than 15 bp. Forward reads were then functionally annotated using DIAMOND BLASTX (v.0.8)[57] using scripts and procedures from the SAMSA2 pipeline (v.2) and the included SAMSA formatted SEED subsystems database[58]. Median read number for metagenomes was 6,398,524 (IQR 4,704,883–8,216,799).

## Statistical analyses

All statistical analyses were performed in R software v.4.2 or above[59], and most plots were generated using ggplot2 v.3.3 (ref. 60). R packages used in custom code (Code availability) to assist core analysis and plotting were: broom.mixed 0.2 (ref. 61), car 3.1 (ref. 62), cowplot 1.1 (ref. 63), eulerr 7.0 (ref. 64), fs 1.6 (ref. 65), ggordiplots 0.4 (ref. 66). ggrepel 0.9 (ref. 67), Hmisc 5.1 (ref. 68), lmerTest 3.1 (ref. 69), magrittr 2.0 (ref. 70), pacman 0.5 (ref. 71), CARTOcolors 2.1.2 (ref. 72), pracma 2.4 (ref. 73), RColorBrewer 1.1 (ref. 74), seqinr 4.2 (ref. 75) and tidyverse:2.0 (ref. 76).

For visualization of the relative importance of the effect of treatment, country, site and sampling time on taxonomic and functional gene abundance data, as well as measures of soil functioning, we ran NMDS analysis using the function metaMDS in the vegan package[77], Bray–Curtis dissimilarity, followed by partial RDA, to visualize the relationships between disturbance treatments and community composition, controlling for country and site effects (vegan R function rda). To test the effect of inclusion of different numbers of countries, total variance explained by disturbance treatments was summed across all four RDA axes. To quantify treatment effects on taxonomic and functional gene abundance data, as well as measures of soil functioning, PERMANOVA was performed on all except the initial samples, testing the effects of disturbance, sampling time and their interactions with country and site. This was implemented using the adonis2 function in the vegan package on Bray–Curtis dissimilarities among Hellinger-transformed counts and 999 permutations. All above assessments of beta-diversity used rarefied data (rrarefy function in vegan), standardized to the sample with the lowest numbers of reads (2,043 bacterial reads, 4,091 fungal reads and 496,126 functional reads).

## Individual resistance and resilience

For identification of different response categories, we fitted linear mixed-effects models to each ASV using the lme function in the nlme R package v.3.1 (ref. 78). These models fitted fixed effects of disturbance treatment, sampling time (in days from S1) and their interaction on ASV relative abundance. Random effects on the intercept were included for country and site within country. Differences in variance (heteroscedasticity) were accounted for with sampling depth (accounting for both additive and proportional error, using the varConstProp function also in the nlme package), and with each of the sampling points (S1–4). In each case, the sign and significance of difference from zero ($P \le 0.05$ or not by Wald test) were recorded for treatment effects (differences in relative abundance from control at S1) and interaction terms (differences in slope of relative abundance over time from control). These were used to classify the ASV response to each disturbance in terms of resistance (either significantly positively or negatively impacted or resistant, with no significant impact at S1) and resilience (either resilient—significant change in relative abundance over time in the opposite direction to the impact, or diverging—significant change over time in the same direction as the impact; or stable—no significant change over time).

Phylogenetic clustering of these strategies was determined by constructing a tree of the 500 most abundant 16S and ITS ASVs, each aligned using kmer and secondary structure-guided alignment

with AlignSeqs from the DECIPHER R package v.2.24 (ref. 79). Trees were constructed using FastTree v.2.1 (ref. 80), with pseudocounts and a generalized time-reversible (gtr) model and rooted at the split between archaea and bacteria for both 16S sequences and fungi at the split between an outgroup comprising ASVs identified as belonging to the basal-branching phylum Chytridiomycota[81] and the other taxa. For each disturbance, a phylogenetic least-squares model of both impact of disturbance and change over time (that is, the treatment and treatment:time effects for each ASV from the mixed-effects models described above) was fitted using the caper v.1.0 package in R[82]. This approach estimates a value and confidence interval for phylogenetic signal in terms of Pagel's lambda[83].

For estimation of the resistance and resilience of functional gene categories, the arcsine square-root-transformed relative abundances of functional genes at each level of the SEED Subsystems classification[84] were first calculated. Mixed-effects models were fitted to these values with the lme4 v.1 package[85]. This model used treatment and sampling time as fixed effects, with site nested in country and all lower SEED Subsystems levels nested (level2/level3/level4) as random effects on the intercept. In addition, observation-level random effects (that is, one level for a particular read-count in a particular sample, for a particular gene in the SEED category being modelled) were fitted for each country, to allow for different levels of variance among countries (that is, heteroscedasticity). Estimated resistance and resilience effects were then extracted from the model in the same way as for the ASV models above (treatment and treatment:time interactions, respectively). *P* values were obtained for the significance of these treatment contrasts, accounting for multiple comparisons with the glht function in the multcomp v.1.4 package in R[86], using Dunnett's test and explicit planned contrasts for treatment and treatment:time, respectively.

## Growth rates
We quantified bacterial growth across whole communities, using the ratio of bacterial origins and bacterial termini. Because bacterial chromosomes replicate from a single origin to a terminus, actively growing bacterial populations have, on average, a gradient in copy number increasing from terminus to origin[87], the basis of tools used to identify the growth rates of particular microbes from metagenomic data[88–90]. We counted matches to *dnaA* protein sequences, the replication initiator protein as a marker of bacterial origins, where it is consistently found[91], and to the *dif* DNA sequence, a broadly conserved 28 bp binding site for the chromosome dimer resolution machinery and a marker of bacterial termi[92], markers specifically validated in a community context[89]. For each sample, we matched one of the paired-end metagenomic read files (R1) to databases of 39,401 *dnaA* sequences taken from RefSeq[93] release 215, and to a set of 578 *dif* sequences[92]. For identification of *dnaA* sequences, we used DIAMOND v.2.0 (ref. 57) in translated (blastx) mode set to 'very sensitive', counting every sequence in which at least one match had an e-value of 0.001 or less as a hit. For identification of *dif* sequences, we used nucleotide BLAST v.2.13 (blastn[94]) with the blastn-short task settings, counting every sequence in which at least one match had an e-value of 0.01 or less as a hit. Counts of origin and terminus hits were converted to estimates of relative bacterial growth by taking the difference between $\log_2$ of the ratio of origin hits to terminus hits in each treatment sample and the equivalent value in the unperturbed control sample. These growth rate estimates were analysed using a mixed-effects model with fixed effects of treatment, sampling time, Bray–Curtis dissimilarities from control of the bacterial and fungal communities and all possible interactions among these effects. Random effects on the intercept of site nested within country were included, along with an effect of treatment on variance, using the lme function in R as above. This complete model was reduced to a minimal adequate model by stepwise removal of non-significant effects using the stepAIC function from the MASS package v.7.3 (ref. 95). This reduced model contained only main effects of treatment, sampling time, the distance of the fungal community from control and the interaction between that distance and treatment.

## Growth capacity
Growth capacity was quantified using RasperGade16S v.0.0.1 (ref. 96) to estimate 16S copy number per cell for each prokaryotic ASV. This gave 1 as the most frequent copy number, 2 as the median copy number and a range of 1–18. The degree of confidence in that estimate (complete confidence, 1) was highly variable (median, 0.58; range, $2.6 \times 10^{-4}$ – 1.00). These values were combined with ASV abundances to create a weighted average expected copy number for each sample, which was compared between treatment and control.

## Community resistance and resilience
The resistance and resilience of microbial community structures (prokaryotes, fungi and metagenome, as opposed to individual ASVs or functional categories) were defined as the negative Bray–Curtis dissimilarity between disturbance and control at S1 and S4, respectively (that is, if the distance was high, the resistance/resilience was low)[14,97]. To test for relationships between initial soil, climatic and microbial properties and microbial community responses to extreme events (resistance and resilience), as well as between those initial properties and the initial relative abundance of each of the highest-level functional categories and between the community response and those initial functional categories, we used rank (Spearman) correlations across samples. We also used rank correlations between distance matrices measuring Bray–Curtis dissimilarity among the relative abundances of functional categories and the Euclidean distance among rank-transformed individual soil properties for each set of soil properties (enzymes: phosphatase, β-glucosidase, leucine-aminopeptidase and acetylesterase; MicroResp: water, alanine, cellulose, citric acid, fructose, glucose, glycine and malic acid; soil C and N availability: DOC, DON, nitrate and ammonium; gas fluxes: $CO_2$, $CH_4$ and $N_2O$) at either the S1 or S4 time point. *P* values were calculated using 999 permutations in the mantel() function in the vegan package. Heatmaps were clustered where necessary using complete-linkage hierarchical clustering on the values of the correlations.

## Predictive model
An explicit predictive model of resistance and resilience (negative log-transformed Bray–Curtis dissimilarity of metagenome gene abundances in treatment from their control at S1 and S4, respectively) was fitted based on the 20 initial soil and environmental properties (four temperature variables, four precipitation variables, eight carbon and nitrogen variables, three water-holding variables and pH), along with treatment and sampling time. These explanatory variables were each standardized to a mean of zero, standard deviation of 1 and used as explanatory variables in a random forests model[98]. That regression forest was fitted using the randomforest R package (v.4.7)[99] with 10,000 trees and a default mtry parameter (one-third of explanatory variables randomly sampled at each split). A modified grouped cross-validation of this model was carried out, in which the data were split into training and test sets and with each training set containing data from two replicate sites in each country; the test set contained data from the third site. There are 19,683 unique ways of doing this split (after exclusions, see above). A random forest model, as above, was fitted to each of these training sets and used to make predictions for each respective test set. The proportion of variance explained (coefficient of determination) was calculated for each test set using the random forest function. An alternative grouped cross-validation was carried out where training sets comprised the data from six countries, and test sets comprised those from the remaining four countries (these proportions were chosen because they enable an estimation of error for each country and are close to the threefold cross-validation proportions used above). There are 210 unique ways of doing this split, with

models fitted and coefficients of determination calculated for each one as above. For both cross-validations, Pearson correlations were calculated between observed values and the mean value predicted across cross-validation models for each site within each perturbation. Partial dependence plots, which visualize the relationship between a variable of interest and the response, also accounting for the average effect of other predictors[100], were created using the pdp R package (v.0.8)[101].

## Ethics and inclusion statement

This project was planned and executed in close collaboration with local partners. Sampling adhered to local laws, and permission was obtained from landowners. All local partners have been included as coauthors.

## Reporting summary

Further information on research design is available in the Nature Portfolio Reporting Summary linked to this article.

## Data availability

All raw sequence files are deposited in the European Nucleotide Archive at project accession PRJEB52753, with paired fastq files for each sample under Run accession nos. ERR9712737–ERR9713356 (16S amplicons), ERR9713357–ERR9713976 (ITS amplicons) and ERR9924623–ERR9924930 (whole-genome metagenomes). Derived data are available at https://doi.org/ngfr, with explicit accessions for raw data at ENA. Climatic data are available from WorldClim.org.

## Code availability

Code is available at https://doi.org/ngfr.

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

**Acknowledgements** This work was funded by NERC Standard Grant no. NE/P01206X/1 to F.T.d.V., C.G.K. and R.I.G. In addition, F.T.d.V. was supported by BBSRC DP Fellowship no. BB/L02456X/1. N.E. gratefully acknowledges the support of iDiv funded by the German Research Foundation (DFG, no. FZT 118, 202548816). J.D. was supported by the European Regional Development Fund (EcolChange) and the Estonian Research Council (no. PRG1065). F.T.M. was supported by the European Research Council (ERC Grant agreement no. 647038), Generalitat Valenciana (no. CIDEGENT/2018/041), the Spanish Ministry of Science and Innovation (no. PID2020-116578RB-I00) and by the King Abdullah University of Science and Technology (KAUST) and the KAUST Climate and Livability Initiative. M. Semchenko was supported by the Estonian Ministry of Education and Research (AgroCropFuture). P.K. was supported by a

project grant (no. 2015-04214) awarded by the Swedish Research Council (Vetenskapsrådet). We thank W. Pritchard, Y. Liu, Y. Zhang, D. I. Korobushkin, V. Ochoa, B. Gozalo, S. Asensio, J. Prommer and L. Prommer for help in the field and laboratory.

**Author contributions** The study was conceived by F.T.d.V., C.G.K., R.I.G. and O.N. Planning and field work was done by O.N., T.G., C.W., H.L., J.D., N.E., K.B.G., A.H., E.J., P.K., F.T.M., M. Semchenko, M. Schädler, C.S., M.A.T., O.V. and W.W. Laboratory work was performed by O.N., C.W., T.G. and H.L. Computational and data analyses were performed by C.G.K., R.I.G., O.N., T.G., B.J. and A.D. The manuscript was written by F.T.d.V., with significant input from C.G.K., R.I.G., O.N. and the other authors.

**Competing interests** The authors declare no competing interests.

**Additional information**
**Correspondence and requests for materials** should be addressed to Christopher G. Knight, Rob I. Griffiths or Franciska T. de Vries.

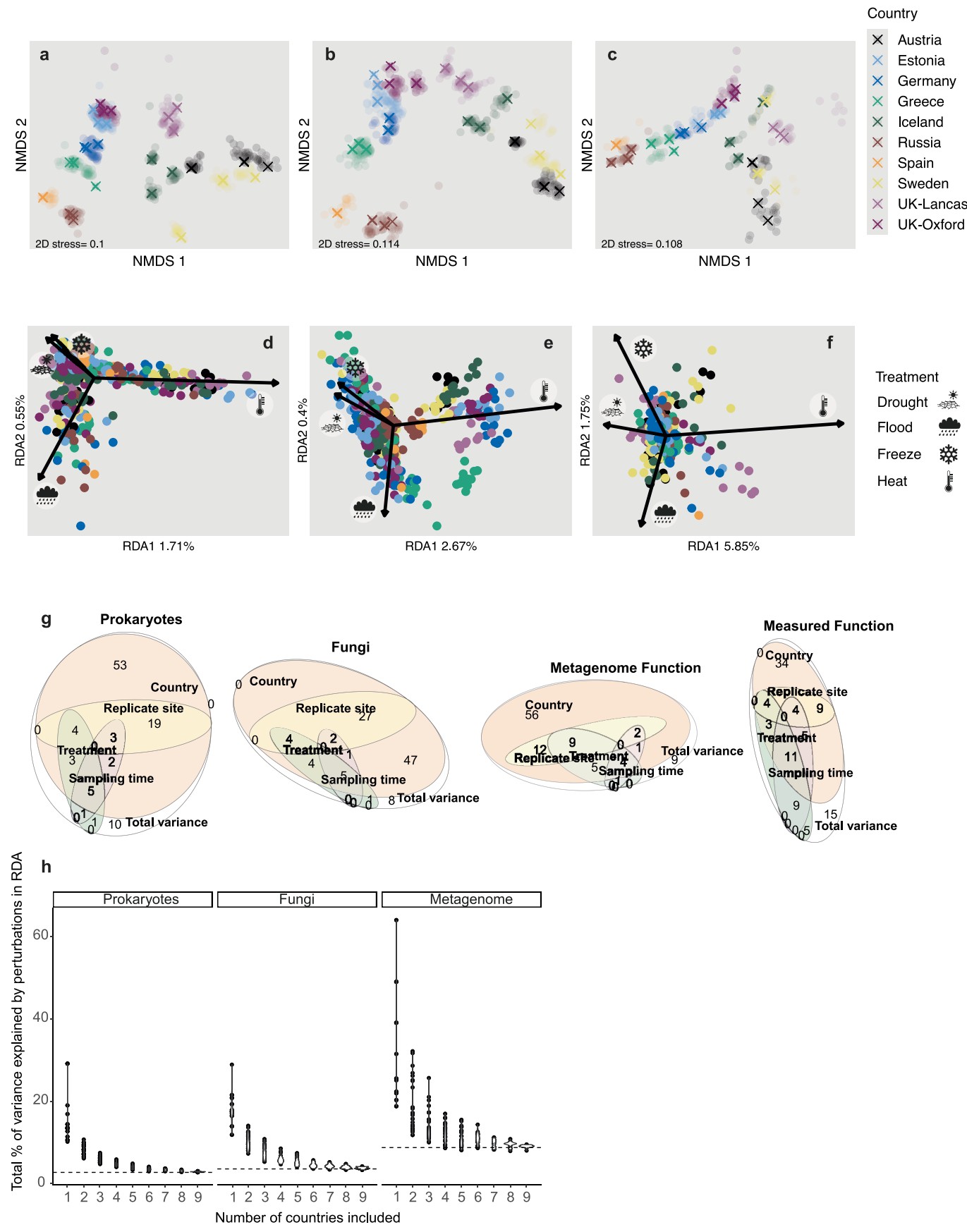

**Extended Data Fig. 1** | See next page for caption.

**Extended Data Fig. 1 | Alternatively coloured versions of the NMDS and RDA analyses in Fig. 1 and percentage variance explained.** Analyses are as in Fig. 1c–h, but in the NMDS analyses **(a-c)**, initial samples are shown as crosses, with all other samples semi-transparent and in the RDA analyses **(d-f)** points are coloured by country (in the same way as in Fig. 1c–e). Euler diagrams **(g)** show the percentage of variance explained. Ellipses have areas approximately proportional to the variance explained by model terms that include the effect in question – Country (orange); Replicate site (yellow); Sampling time (pink) and Treatment (green). Overlaps of ellipses correspond to interactions among effects. The residual variance corresponds to the area only in the white, Total Variance, ellipse. Values annotated on particular areas are the percentage variance explained by that area to the nearest 1%. Thus, for example, the interaction among Country, Replicate site, and Treatment, represented by the overlap of orange, yellow and green ellipses, accounts for 4% of the variance in prokaryotes, fungi and measured functions, but 9% of variance in metagenomically inferred functions. Note that this representation is inevitably imperfect, specifically there are small ellipse overlaps that do not represent any variance; these are explicitly marked with a zero. Data from Extended Data Table 1. The variance in communities explained by the perturbations depends on the number of countries included **(h)**. The total percentage community variance attributed to drought, flood, freeze and heat perturbations in the RDA analysis when repeated with fewer countries included in the data. The dashed line indicates the percentage variance explained by perturbations when including all ten countries (Fig. 1f–h). Points, summarised by an overlying 'violin' (indicating the density of points), correspond to the percentage variance for a reduced dataset missing all data from one or more of the countries. 30 randomly selected combinations of dropped countries were analysed for each size of dataset for Prokaryotes, Fungi and metagenomically encoded functions respectively.

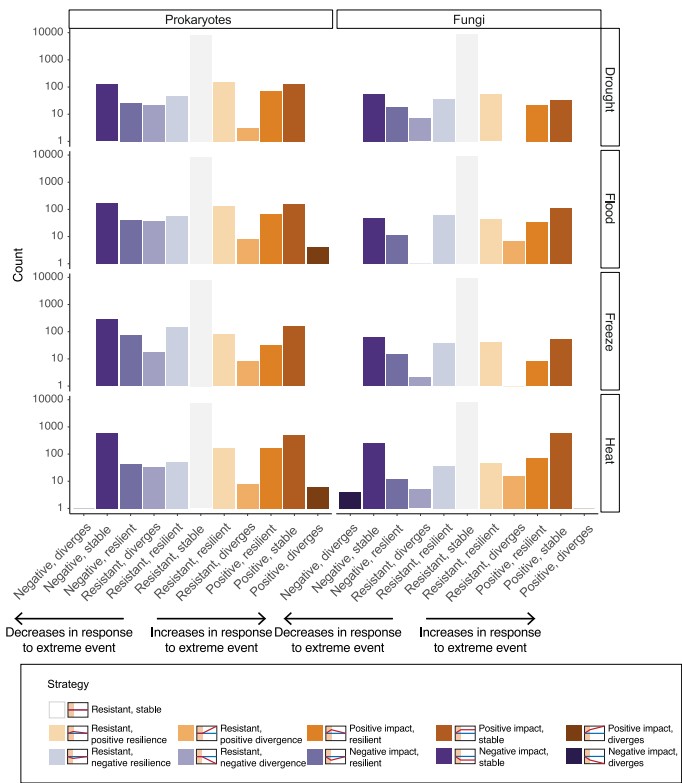

**Extended Data Fig. 2 | Total numbers of 'strategies' across all ASV sequences for which a resistance-resilience model could be fitted.** A single model (see Methods) was fitted for each ASV (accounting for the experimental structure and differences in variance among treatments) and both the initial change at S1 (impact) and change over time in relative abundance, compared with the control, classified as positive or negative and significant or not for each extreme climatic event. This gives 11 possible strategies, illustrated in the key (where the blue line indicates a stable control over time and the red line the effect of the treatment, shown by a shaded background) and indicated by colour. Note the logarithmic scale. The taxonomic distribution of these strategies is shown for the 500 most abundant ASVs in Fig. 2.

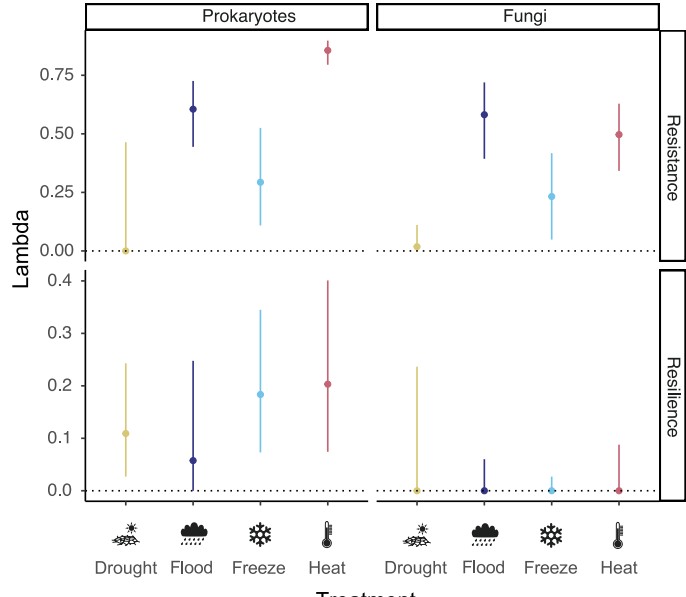

**Extended Data Fig. 3 | Phylogenetic signal in resistance and resilience.**
Each point represents an estimate (by phylogenetic generalised least squares
regression using the caper package, see Methods) of Pagel's lambda for
resistance or resilience in response to a particular extreme climatic event
across a phylogenetic tree of either prokaryotes or fungi (the tree estimated
from the ASVs' 16S or ITS sequences, $N$ = 493 and 466 samples respectively,
see Methods). A value of 1 would correspond to resistance or resilience to the
disturbance that evolved only randomly (according to Brownian motion) along
the tree, whereas a value of 0 corresponds to no detectable similarity in the
resistance or resilience of related strains at all. Error bars correspond to 95%
confidence intervals from the model. Resistance and resilience here simply
refer to the intercept and slope of relative abundance over time from the mixed
effects models of the 500 most abundant prokaryotic or fungal ASVs for which
such a model could be fitted (494 and 466 ASVs respectively).

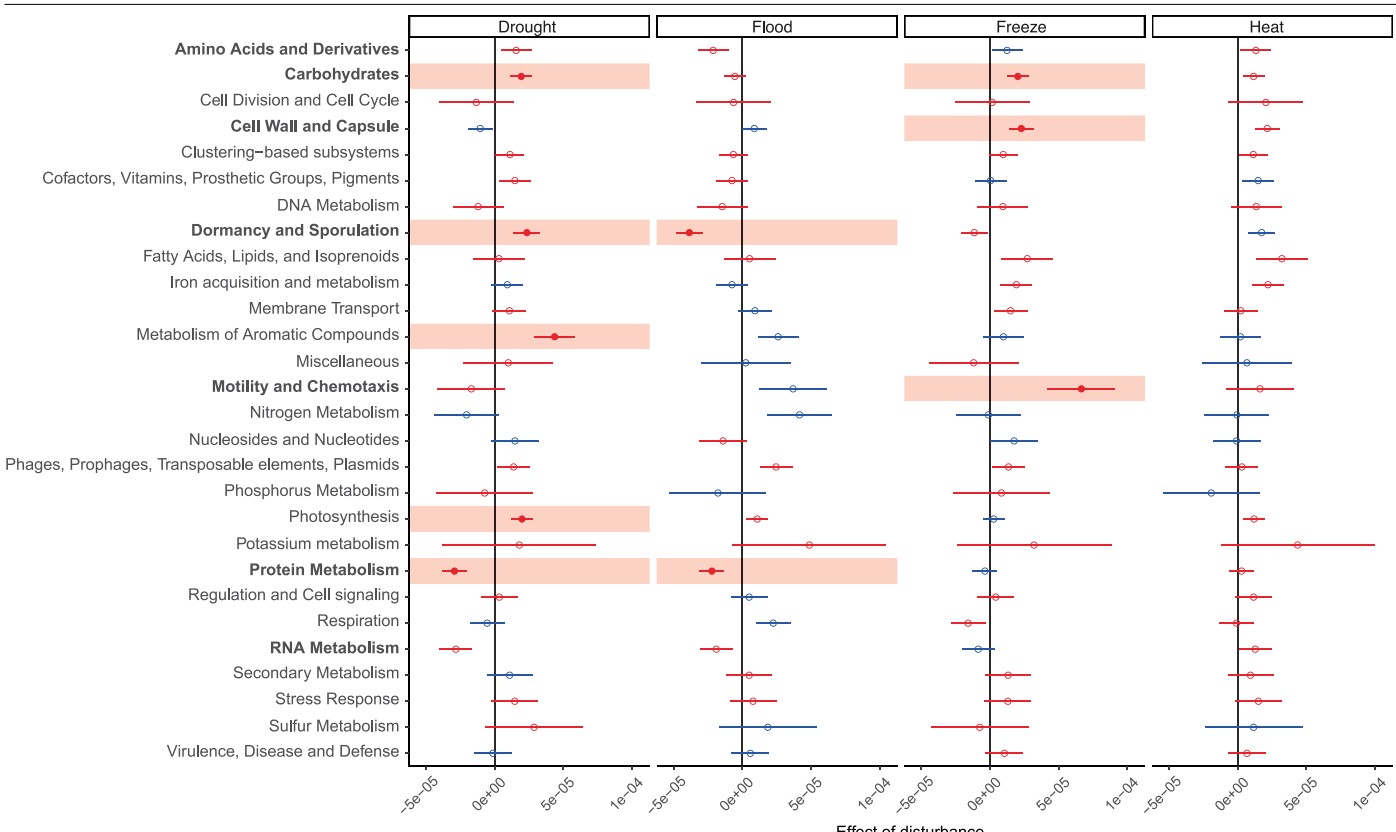

**Extended Data Fig. 4 | Resilience in the 28 highest-level functions in response to the extreme climatic events.** Analysis from the same models as Fig. 3a ($N$ = 280 metagenomic samples, see Methods) and showing the same as that figure, except that values (points representing estimates from a mixed-effects model ±SE) indicate how the abundance of the function in question changes, relative to control, after a month of recovery (i.e. the interaction between treatment and time in a mixed-effects model). As in Fig. 3a,b, points are filled and contribute to the background colour if significantly different from the control by Dunnett's test. However, the colour in each case here indicates resilience, i.e. if the change over time (relative to control) goes in the opposite direction to the initial disturbance (resilience), that is coloured red, whereas non-resilient functions, where change over time goes in the same direction as the initial change, are coloured blue.

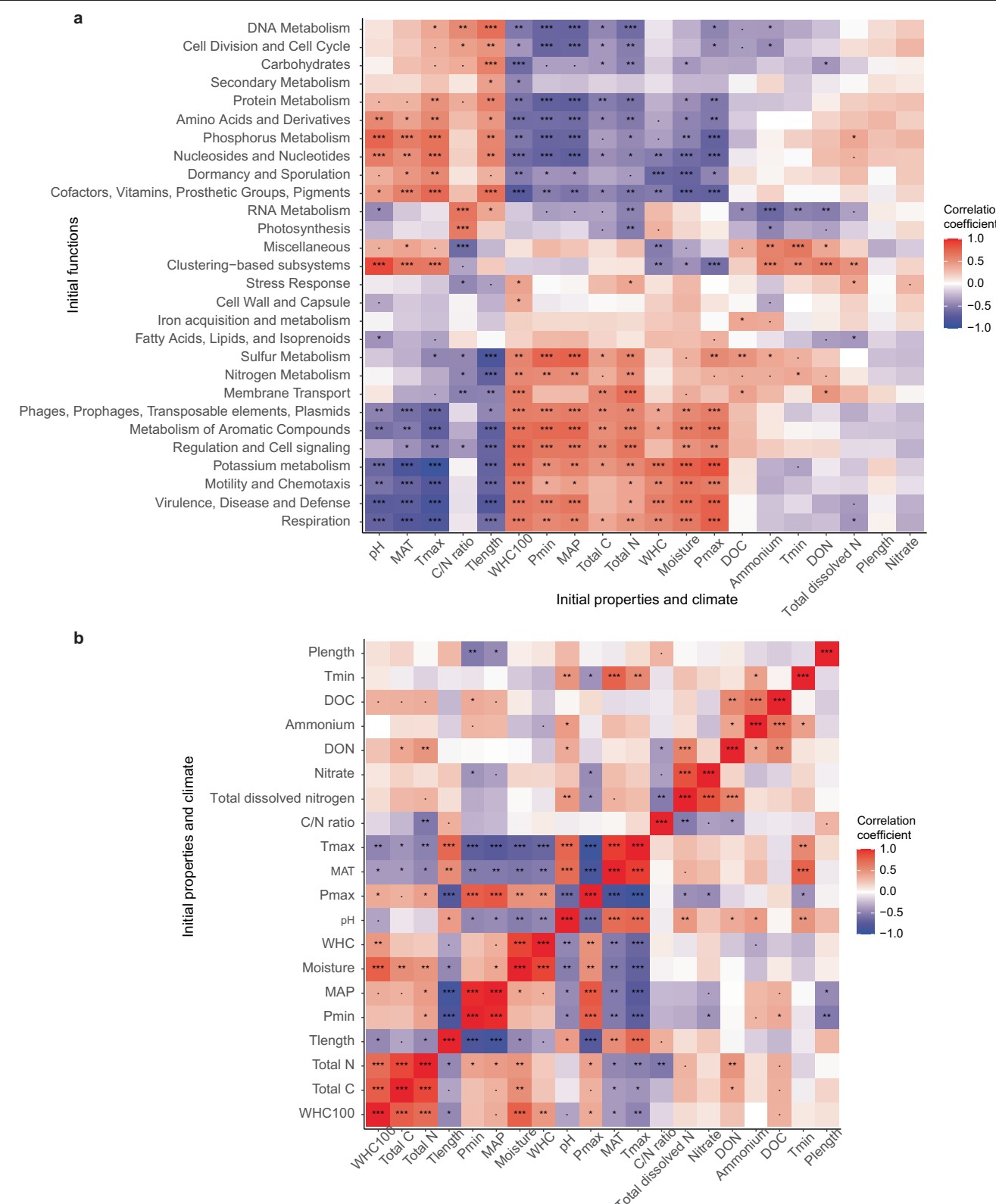

**Extended Data Fig. 5** | See next page for caption.

**Extended Data Fig. 5 | Correlations between initial soil and climatic properties and the 28 highest level functions.** Correlations between the measured properties of the initial soil samples and the relative abundance of the 28 highest level functional categories in the metagenomes of those same samples (**a**), and correlations between initial soil and climatic properties (**b**). The order of the columns and rows is clustered by the value of the correlations. Asterisks indicate the *P*-value for a 2-tailed correlation test of the rank correlation differing from zero: *** ≤ 0.001 < ** ≤ 0.01 < * ≤ 0.05 < · ≤ 0.1, uncorrected for multiple testing. Abbreviations: MAT – mean annual temperature; Tmax – maximum annual temperature; C/N ratio – soil carbon to nitrogen ratio; Tlength – difference between Tmax and Tmin; WHC100 – soil moisture content at 100% water holding capacity; Pmin – annual precipitation minimum; MAP – mean annual precipitation; Total C – total soil carbon content; Total N – total soil nitrogen content; WHC – soil moisture content expressed as percentage of moisture content at 100% water holding capacity; Pmax – annual precipitation maximum; DOC – dissolved organic carbon concentration; Ammonium – plant available ammonium concentration; Tmin – minimum annual temperature; DON – dissolved organic nitrogen concentration; Total dissolved N – total dissolved nitrogen concentration; Plength – difference between Pmax and Pmin; Nitrate – plant available nitrate concentration.

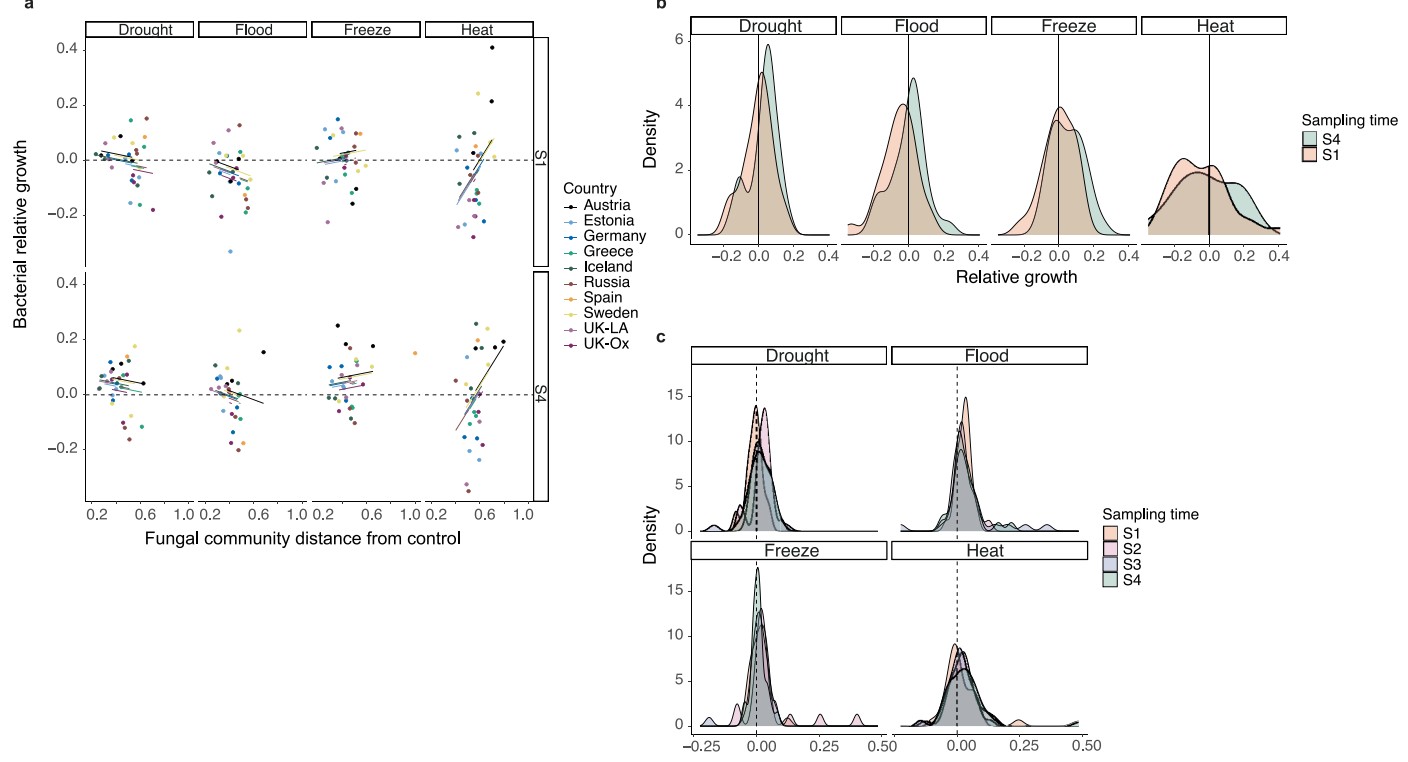

**Extended Data Fig. 6 | Bacterial growth and growth capacity relative to control.** Growth estimated via the ratio of origins to termini (see Methods), is shown in relation to the distance from the control of the fungal community in the same sample **(a)**, and the distribution across samples at different time points **(b)**, which is also shown for growth capacity estimates **(c)**, from 16S rRNA copy numbers. Only bacteria are considered because only bacterial chromosomes replicate from a single origin towards a single terminus, enabling growth to be estimated in this way. **a**, Growth equal to the control is indicated by the dashed black line. Coloured lines indicate country-level fits from the mixed effects model (see Methods). Growth is estimated by the difference in relative abundance of origins of replication (identified by matches to the *dnaA* gene) to termini (identified by matches to the *dif* sequence) in disturbed samples relative to control in the metagenomes. Distance to control is the Bray-Curtis dissimilarity, between disturbed and control samples, of the relative abundances of ITS sequence ASVs. Countries are indicated by colour, as in Fig. 1. Facets are ordered by perturbation

(horizontally) and sampling time (vertically). **b**, Bacterial growth in disturbed samples relative to control in the metagenomes. Zero indicates the same degree of growth as the undisturbed control. Comparisons are made on a $\log_2$ scale, so a value of 1 would indicate twice as much growth (i.e. twice the ratio of origins to termini) in disturbed than control and −1 half as much growth. **c**, Increased bacterial growth capacity is associated with increased number of copies of the 16S rRNA[32–34]. The average expected number of 16S rRNA copies per cell for each sample is therefore estimated from our 16S data, weighted by confidence. Both expected copy number per cell and confidence in that estimate were calculated using the RasperGade16S software[96] from the 16SrRNA data. The difference between treatment and control samples is calculated on a $\log_2$ scale (i.e. double growth capacity in perturbed versus control would give a value of 1) for each sample at each sampling time (colour) and in each treatment. Times after disturbance are coded by facet **a** and colour **b-c**: S1 = directly after disturbance; S2 = 1 day; S3 = 1 week; S4 = 1 month.

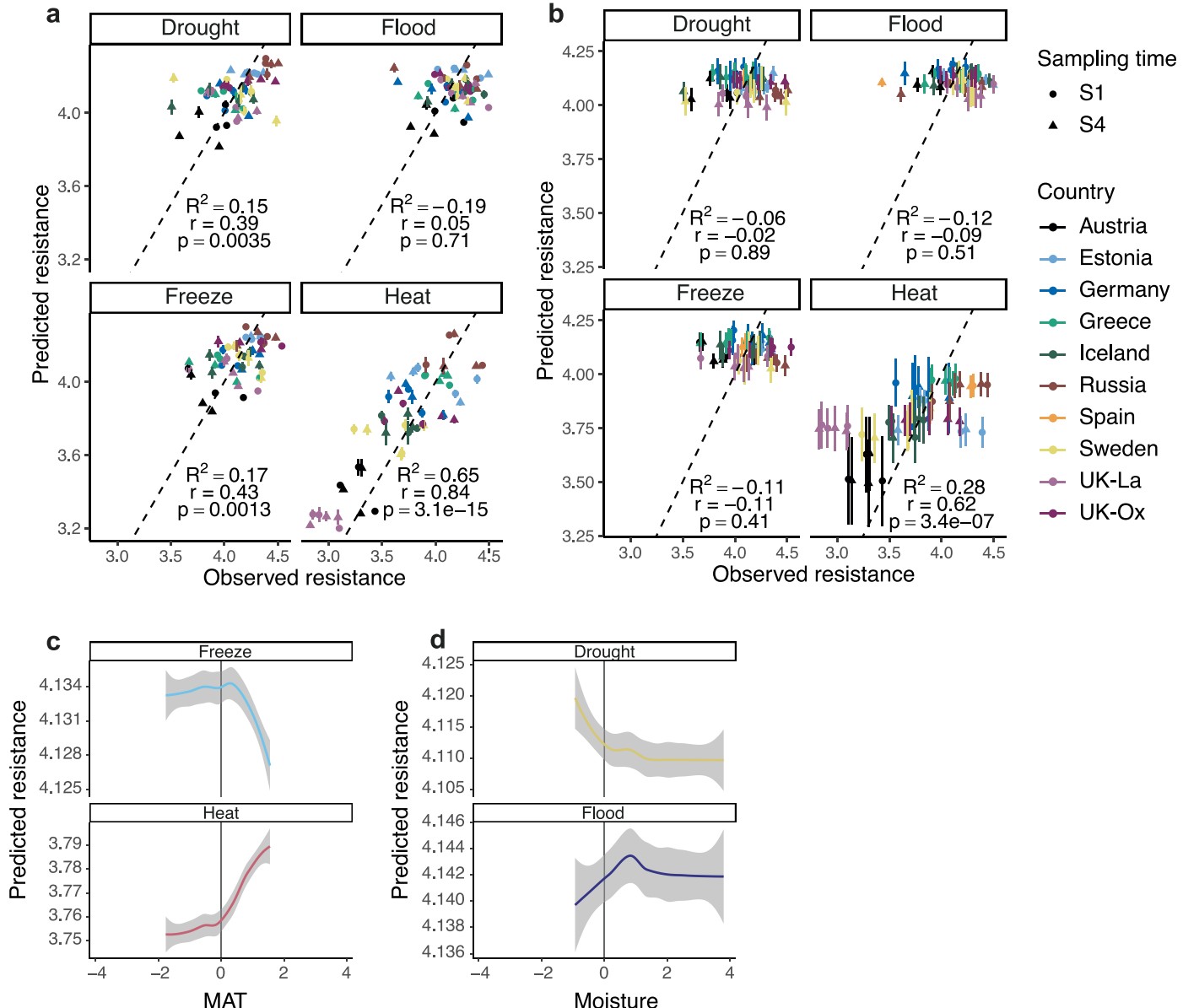

**Extended Data Fig. 7 | Model predicting the magnitude of metagenomic resistance and resilience from climate and initial soil properties. a** The horizontal axis represents the observed resistance (measured as the distance between control and perturbed treatments in metagenome gene abundances, using negative log Bray Curtis dissimilarity) either immediately after perturbation (S1, circles) or after a month of recovery (S4, triangles). Colours represent countries as in Fig. 1. The vertical axis represents the prediction of a random forest model (see Methods, N = 152 samples in the training set of each cross-validation run) trained on data not including any samples of the replicate site for which the impact is predicted. Points and error bars therefore represent mean and sd respectively for the predictions from models fitted to each possible split of the data into training and test sets. The mean coefficient of determination ($R^2$) for all training-test set splits of the data is 0.58 (±0.04, sd). The coefficient of determination ($R^2$) of the observed versus mean predicted resistance within each perturbation, a 2-tailed correlation test of the Pearson correlation coefficient (r) and its probability (p) of being different from zero, are given in the lower

right of each facet. Note the lack of Spanish samples as only one Spanish site was included, meaning cross-validation was not possible among sites, giving N = 54 predictions for the correlations. **b** Alternative cross-validation grouped by countries (largely confounded with biogeographic regions, Fig. 1a) in which models are fitted to data from 6 countries and tested on data from the remaining four, giving N = 56 predictions for the correlations. Other aspects as in part **a**. **c** Based on a model containing all the data (coefficient of determination $R^2$ = 0.61, N = 224 samples), the partial dependencies on mean annual temperature (MAT) of outcome in response to freeze and heat and **d** moisture of response to drought and flood are contrasting for warmer climates (greater than average temperatures) and drier soils (less than average moisture) respectively. Note that the horizontal scales in **c** and **d** are standardised, so a value of zero corresponds to the mean across all sites, and the scale is in units of standard deviation. In each case, the smooth line and error bound is a loess smoother of the partial dependence within the random forest model.

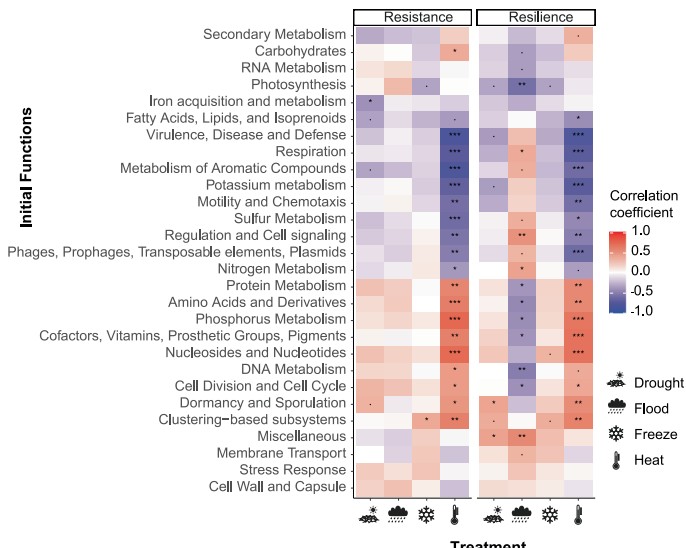

**Extended Data Fig. 8 | Correlations between soil initial functions and resilience and resilience of community function across samples from 28 different sites.** Rank correlations are shown between the initial relative abundance of each of the 28 highest level functions (from the metagenomics) and either the resistance (left) or resilience (right) of overall community function. Resistance and resilience are measured as the negative Bray-Curtis dissimilarity between control and disturbances treatment of the relative abundance of each of the finest scale functions (proteins) directly after the disturbance or after one month of recovery, respectively. Initial functions are clustered, with asterisks indicating a 2-tailed correlation test where the rank correlation is significantly different from zero (*** ≤ 0.001 < ** ≤ 0.01 < * ≤ 0.05 < · ≤ 0.1, uncorrected for multiple testing).

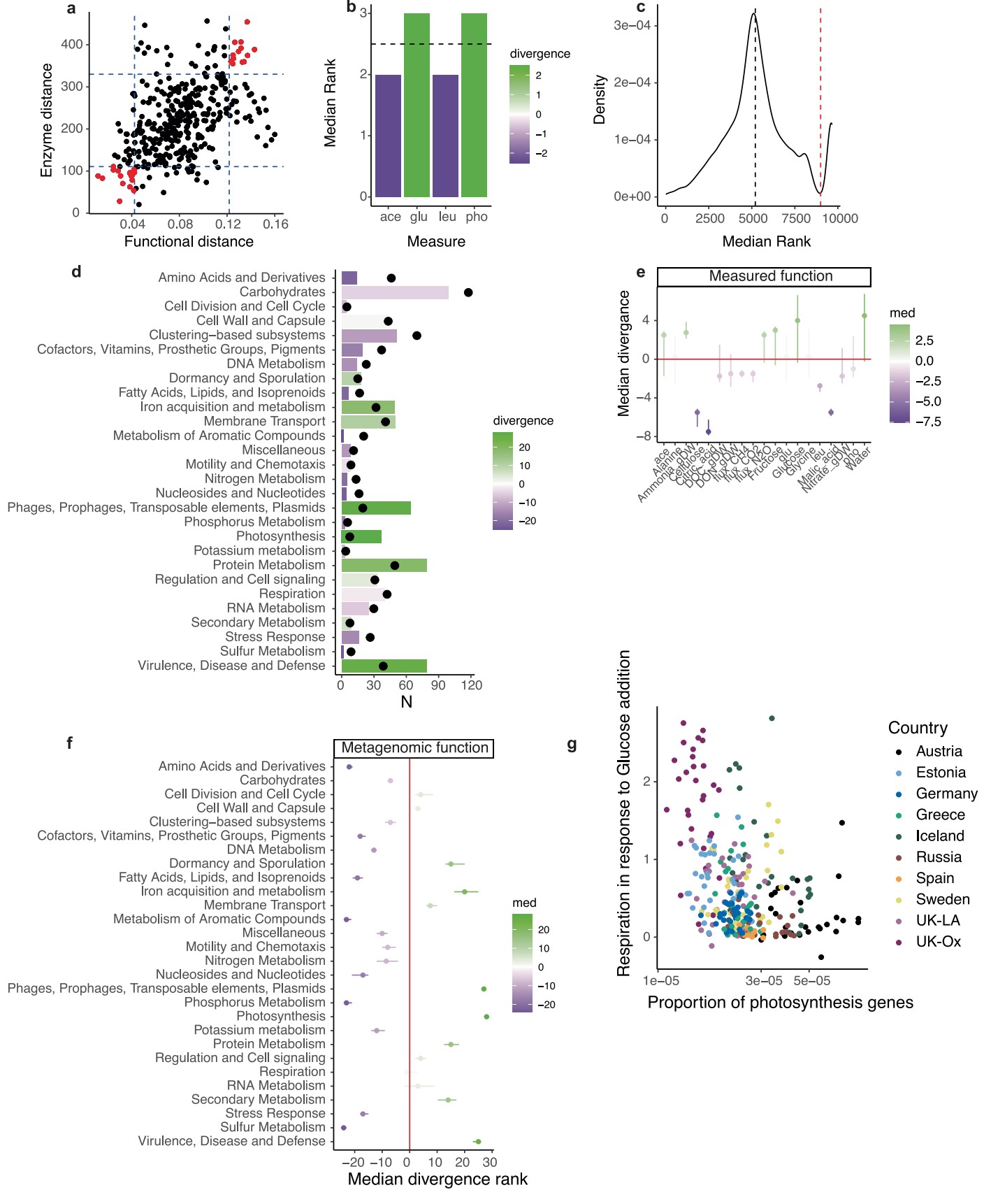

**Extended Data Fig. 9** | See next page for caption.

**Extended Data Fig. 9 | Drivers of correlations between metagenomic and measured functions in Fig. 4b.** Example is shown in **a** of the correlation between the distance matrix among samples for metagenome-derived functions and the distance matrix among samples of measured enzymatic activities for the freeze treatment at time point S4 (included in Fig. 4b). This is a rank correlation of 0.50, which is highly significantly different from zero (2-tailed Mantel test, $N = 756$ inter-site comparisons, Nperm=999 $P \leq 0.001$). The blue dashed lines indicate the 10% and 90% quantiles of each distribution. The points falling beyond these lines on both axes (shown in red) correspond to comparisons between particular samples that are driving the correlation. Both for measured functions **b** (in this case four enzyme activities – ace = Acetate esterase enzyme activity; glu = Beta-glucosidase enzyme activity; leu = Leucine amino peptidase activity; pho = Phosphatase enzyme activity each originally in nmol/h/g dry soil) and the metagenomic functional data **c** (many, 10,408, individual proteins) we ranked the individual distances within each of these site comparisons and plotted the median rank across the selected site comparisons. In each case high ranks indicate large distances for comparisons from the upper right of part **a** and small distances for comparisons from the lower left of **a**. Thus a high median rank suggests that the variable is consistently important for determining the distance among samples in these extreme comparisons. In **b** we show with a dashed line the expected median rank if all measures behaved similarly and coloured by the divergence from this (ranking |(observed – expected)|/expected, then signed according to whether it the observed value is greater or less than expected). In **c** the distribution of median ranks has two main modes, one in the middle of the distribution, close to the null value (vertical black line), corresponding to measurements that are not consistently far or close between samples and therefore not driving the correlation, and one on the right, corresponding to comparisons that are consistently associated in the correlation-driving comparisons. We selected this second mode (taking all comparisons with a median rank above 9,000, indicated by the vertical red line) and looked at their corresponding highest-level functions in **d**. Here, the bars indicate the numbers of these high-ranking comparisons that involve proteins in the given functional category, whereas the black blobs indicate the numbers of proteins in that category that would have been expected, had there been a random selection. Colour then indicates the ranked divergence between these two, coloured by direction (as in **b**). There is under-representation of a wide range of smaller categories, such as Sulfur Metabolism and Metabolism of Aromatic compounds. We conducted these same analyses for all pairs of distances, all treatments and timepoints shown as a Mantel correlation in Fig. 4b, taking a median and interquartile range (point and error bars shown) across these correlations for measured functions **e** and metagenomic functions, **f** ($N = 756$ site-site comparisons in each case). Among laboratory functional measurements, none was particularly strongly associated with positive correlations, though specific substrate activities (citric acid, Cellulose and Malic acid) were consistently not driving the correlation. In contrast, the metagenomic functions, **f**, showed several high-level categories that were particularly over-represented in positive associations: Photosynthesis, Phages, Prophages, Transposable elements and Plasmids and Virulence, Disease and Defence; followed by Iron acquisition and metabolism, Dormancy and Sporulation, Protein Metabolism, Secondary metabolism. A more disparate range of metagenomic functions was un-associated with the measured functions. This suggests that particularly important relationships for driving the metagenomic association with measured functions may be those scoring highly in both **e** and **f**, such as **g**, the negative association between the proportion of photosynthetic genes and the response to glucose addition ($N = 278$ samples, rank correlation = $-0.40$, $P = 3.8 \times 10^{-12}$), each of which varies substantially across countries (colour).

**Extended Data Table 1 | PERMANOVA results of the effect of treatment (extreme climatic event), sampling time, country, and site on microbial communities**

| Variable | Prokaryotes $R^2$ | $F$ | $P$ | Fungi $R^2$ | $F$ | $P$ | Metagenome $R^2$ | $F$ | $P$ | Function $R^2$ | $F$ | $P$ |
|---|---|---|---|---|---|---|---|---|---|---|---|---|
| Treatment | **0.0083** | $F_{4,208}$ 4.5 | ≤0.001 | **0.010** | $F_{4,206}$ 6.4 | ≤0.001 | **0.011** | $F_{4,72}$ 2.2 | 0.005 | **0.052** | $F_{4,589}$ 20 | ≤0.001 |
| Sampling time | **0.0028** | $F_{3,208}$ 2.0 | ≤0.001 | **0.0025** | $F_{3,206}$ 2.1 | ≤0.001 | 0.0015 | $F_{1,72}$ 1.2 | 0.24 | 0.000 | $F_{3,589}$ -0.086 | 1.0 |
| Country | **0.53** | $F_{9,208}$ 126 | ≤0.001 | **0.47** | $F_{9,206}$ 128 | ≤0.001 | **0.56** | $F_{9,72}$ 51 | ≤0.001 | **0.34** | $F_{9,589}$ 58 | ≤0.001 |
| Site | **0.19** | $F_{18,208}$ 23 | ≤0.001 | **0.27** | $F_{18,206}$ 36 | ≤0.001 | **0.12** | $F_{18,72}$ 5.4 | ≤0.001 | **0.093** | $F_{20,589}$ 7.1 | ≤0.001 |
| Treatment: Sampling time | 0.0060 | $F_{12,208}$ 1.1 | 0.21 | **0.0056** | $F_{12,206}$ 1.1 | 0.04 | 0.0055 | $F_{4,72}$ 1.1 | 0.28 | **0.086** | $F_{12,589}$ 11 | ≤0.001 |
| Treatment: Country | **0.029** | $F_{36,208}$ 1.7 | ≤0.001 | **0.0039** | $F_{36,206}$ 2.7 | ≤0.001 | **0.054** | $F_{36,72}$ 1.2 | 0.027 | 0.029 | $F_{36,589}$ 1.2 | 0.14 |
| Sampling time: Country | **0.016** | $F_{27,208}$ 1.3 | 0.003 | **0.014** | $F_{27,206}$ 1.3 | ≤0.001 | 0.012 | $F_{9,72}$ 1.1 | 0.32 | **0.052** | $F_{27,589}$ 2.9 | ≤0.001 |
| Treatment: Site | **0.041** | $F_{72,208}$ 1.2 | ≤0.001 | **0.043** | $F_{72,206}$ 1.5 | ≤0.001 | 0.090 | $F_{72,72}$ 1.0 | 0.34 | 0.044 | $F_{80,589}$ 0.84 | 0.90 |
| Sampling time: Site | 0.026 | $F_{54,208}$ 1.0 | 0.27 | 0.023 | $F_{54,206}$ 1.0 | 0.24 | 0.022 | $F_{18,72}$ 1.0 | 0.38 | 0.040 | $F_{60,589}$ 1.0 | 0.46 |
| Sampling time: Treatment: Country | 0.050 | $F_{104,208}$ 1.0 | 0.22 | **0.046** | $F_{104,206}$ 1.1 | ≤0.001 | 0.044 | $F_{36,72}$ 1.0 | 0.46 | **0.11** | $F_{105,589}$ 1.5 | ≤0.001 |
| Residual | 0.097 | | | 0.084 | | | 0.087 | | | 0.15 | | |
| Total | 1.0 | | | 1.0 | | | 1.0 | | | 1.0 | | |

See Methods for model descriptions. Bold $R^2$ values indicate those for which the uncorrected probability of being greater than zero $P ≤ 0.05$, in a one-tailed permutation test with the $F$ and $P$ values given. $R^2$ values are plotted out in Extended Data Fig. 1g.

# Reporting Summary

## Statistics

For all statistical analyses, confirm that the following items are present in the figure legend, table legend, main text, or Methods section.

| n/a | Confirmed | |
|-----|-----------|---|
| ☐ | ☒ | The exact sample size (*n*) for each experimental group/condition, given as a discrete number and unit of measurement |
| ☐ | ☒ | A statement on whether measurements were taken from distinct samples or whether the same sample was measured repeatedly |
| ☐ | ☒ | The statistical test(s) used AND whether they are one- or two-sided<br>*Only common tests should be described solely by name; describe more complex techniques in the Methods section.* |
| ☐ | ☒ | A description of all covariates tested |
| ☐ | ☒ | A description of any assumptions or corrections, such as tests of normality and adjustment for multiple comparisons |
| ☐ | ☒ | A full description of the statistical parameters including central tendency (e.g. means) or other basic estimates (e.g. regression coefficient) AND variation (e.g. standard deviation) or associated estimates of uncertainty (e.g. confidence intervals) |
| ☐ | ☒ | For null hypothesis testing, the test statistic (e.g. *F*, *t*, *r*) with confidence intervals, effect sizes, degrees of freedom and *P* value noted<br>*Give P values as exact values whenever suitable.* |
| ☒ | ☐ | For Bayesian analysis, information on the choice of priors and Markov chain Monte Carlo settings |
| ☐ | ☒ | For hierarchical and complex designs, identification of the appropriate level for tests and full reporting of outcomes |
| ☐ | ☒ | Estimates of effect sizes (e.g. Cohen's *d*, Pearson's *r*), indicating how they were calculated |

*Our web collection on statistics for biologists contains articles on many of the points above.*

## Software and code

Policy information about availability of computer code

| Data collection | Data collection not involving software |
|-----------------|----------------------------------------|
| Data analysis | Most analysis was carried out with R version 4.2, using the following key packages:<br>broom.mixed 0.2, caper 1.0, car 3.1, cowplot 1.1, DADA2 1.24, DECIPHER 2.24, eulerr 7.0, fs 1.6, ggordiplots 0.4, ggplot 3.3, ggrepel 0.9, Hmisc 5.1, lme4 1.1, lmerTest 3.1, magrittr 2.0, MASS 7.3, multcomp 1.4, nlme 3.1, pacman 0.5, pals 1.9, pdp 0.8, pracma 2.4, randomForest 4.7, RasperGade16S 0.0, RColorBrewer 1.1, seqinr 4.2, tidyverse 2.0, vegan 2.6<br>In addition, the following stand-alone software was utilised<br>FastTree (2.1)<br>Diamond (0.8 and 2.0)<br>BLAST (2.13) |

For manuscripts utilizing custom algorithms or software that are central to the research but not yet described in published literature, software must be made available to editors and reviewers. We strongly encourage code deposition in a community repository (e.g. GitHub). See the Nature Portfolio guidelines for submitting code & software for further information.

## Data

Policy information about [availability of data](availability of data)

All manuscripts must include a [data availability statement](data availability statement). This statement should provide the following information, where applicable:

- Accession codes, unique identifiers, or web links for publicly available datasets
- A description of any restrictions on data availability
- For clinical datasets or third party data, please ensure that the statement adheres to our [policy](policy)

All raw sequence files are deposited in the European Nucleotide Archive at project accession PRJEB52753, with paired fastq files for each sample under Run accessions ERR9712737-ERR9713356 (16S amplicons); ERR9713357-ERR9713976 (ITS amplicons), and ERR9924623-ERR9924930 (whole genome metagenomes). Derived data and all code are available at https://doi.org/ngfr with explicit accessions for the raw data at ENA. Climatic data are available from WorldClim.org

## Human research participants

Policy information about [studies involving human research participants and Sex and Gender in Research.](studies involving human research participants and Sex and Gender in Research.)

| Reporting on sex and gender | NA |
|---|---|
| Population characteristics | NA |
| Recruitment | NA |
| Ethics oversight | NA |

Note that full information on the approval of the study protocol must also be provided in the manuscript.

# Field-specific reporting

Please select the one below that is the best fit for your research. If you are not sure, read the appropriate sections before making your selection.

☐ Life sciences      ☐ Behavioural & social sciences      ☒ Ecological, evolutionary & environmental sciences

For a reference copy of the document with all sections, see [nature.com/documents/nr-reporting-summary-flat.pdf](nature.com/documents/nr-reporting-summary-flat.pdf)

# Ecological, evolutionary & environmental sciences study design

All studies must disclose on these points even when the disclosure is negative.

| Study description | The experiment was a factorial design between treatment (a factor with five levels) and sampling time (a factor with four levels, S1-S4, though some analyses only used S1 and S4 and there was also an Initial sample for each soil). A nested sampling approach was taken with three replicate sites sampled within each of 10 countries. A single pot of soil was utilised for each replicate site at each sampling time in each treatment. |
|---|---|
| Research sample | Samples from each replicate site comprised 4 pooled 3cm diameter x 15cm long soil cores |
| Sampling strategy | For a fixed number of samples to be analysed, the strategy was to maximise the diversity of soils sampled. This results in the design with a single sample for each soil/treatment/time combination. |
| Data collection | Laboratory data collection on soils carried out in Manchester by ON, CW and HL with sequencing data collected by CGR, Liverpool and the UK Centre for Ecology and Hydrology (RIG, TG and BJ) |
| Timing and spatial scale | Samples were collected when the average temperature of the location was closest to 18 °C, i.e. in spring for southern locations, and after the snow melt, in summer, for the northern locations. In May 2018, we collected soil from Russia, Greece, and Estonia and Spain, followed by Germany and Oxford in June, Austria and Iceland in July, and Lancaster and Sweden in August. Three spatial scales were used – European country-scale shown in Supplementary Fig. 1; replicate sites within countries, separated by 0.05 – 11.76 km; cores within sites, taken from four random points within seven 1 m x 1 m plots were arranged at least 5 m apart. |
| Data exclusions | Two replicate sites from Spain (sites 1 and 3; n = 42) were excluded from the analyses due to low DNA yield and poor recovery of reads particularly for the prokaryotic amplicons |
| Reproducibility | The clustering of replicate sites in Fig. 1 demonstrates reproducibility. |
| Randomization | Location of soil cores within plots were randomised; assignment of soil samples to treatments was done randomly; physical locations of microcosms on trays was randomised. |

| Blinding | On samples' arrival in Manchester all researchers were blinded to the identity of the samples by making their only identity the unique random numbers allocated to them during the randomisation. |

Did the study involve field work?  ☒ Yes   ☐ No

## Field work, collection and transport

| Field conditions | Conditions at each site are given in the data file associated with the samples |
| Location | Locations, including latitude and longitude are given in the data file associated with the samples |
| Access & import/export | Soils were imported under DEFRA plant health licence 116619/374208/0 Valid from 01 January 2019 to 31 December 2019 |
| Disturbance | Clearing of obstructions at the sites was limited to the 7 single square metre plots. |

# Reporting for specific materials, systems and methods

We require information from authors about some types of materials, experimental systems and methods used in many studies. Here, indicate whether each material, system or method listed is relevant to your study. If you are not sure if a list item applies to your research, read the appropriate section before selecting a response.

### Materials & experimental systems

| n/a | Involved in the study |
|-----|----------------------|
| ☒ ☐ | Antibodies |
| ☒ ☐ | Eukaryotic cell lines |
| ☒ ☐ | Palaeontology and archaeology |
| ☒ ☐ | Animals and other organisms |
| ☒ ☐ | Clinical data |
| ☒ ☐ | Dual use research of concern |

### Methods

| n/a | Involved in the study |
|-----|----------------------|
| ☒ ☐ | ChIP-seq |
| ☒ ☐ | Flow cytometry |
| ☒ ☐ | MRI-based neuroimaging |

