## [Peer Review File · Nature]

Manuscript Title: Soil microbiomes show consistent and predictable responses to extreme events

Reviewer Comments & Author Rebuttals

Reviewer Reports on the Initial Version:

Referees' comments:

Referee #1 (Remarks to the Author):

The presented manuscript details an investigation into the effect of 4 different climate stressors on grassland soils from across Europe, using a combination of amplicon (fungal and bacterial), metagenomics, soil geochemistry and enzymatics to characterise incubations. The manuscript is mostly well written, reflecting the straightforward experimental design consistently applied across the grassland sites. However, I found several of the conclusions of the study to be unsupported by the evidence presented.

In particular, the conclusion in the abstract that heat “decreased .. protein biosynthesis genes” seems unlikely. This statement is made through interpretation of a read-centric analysis of metagenomes. This data is not a reflection of gene expression but of genomic potential, and thus only changes when community members grow or die. But since protein synthesis is a universal function, a decrease in the relative abundance of these genes more likely points to an artefact of the functional gene analysis or its interpretation.

As well, the abstract concludes that “heat most strongly impacted soil microbiomes”, but it isn't clear how the different stressors can be compared, since the specific scale of each pressure is not derived with respect to future climate predictions.

I also found the conclusion that the “magnitude of this response was dampened in microbiomes from soils that are prone to conditions similar to the exposed extreme events” somewhat problematic. Observation of resilience/resistance to heat and drought in samples from warmer and drier environments is not precisely equivalent to the above, since the climates of the different regions are differently variable. Also, this conclusion was only reached for 2 of the 4 treatments, with 1 (freeze) showing the opposite correlation (line 152), so a general conclusion about climate extremes seems inappropriate. On top of this, since the “heat” condition was a consistent 35 degrees C for all samples, samples from cooler environments had a larger scale of treatment (more degrees warmed), so are subjected to a more extreme environmental change than those from warm environments.

I had also had several minor concerns:

15: “share” was confusing to this reader. Did this mean across soil samples? Or across treatments too?

76, 82-24: Resistance and resilience are defined here for the purposes of differentiating between different types of responses. But, these two words are very similar to each other. Can one of them be replaced by a more distinct word? This would aid readability.

102-103: I found this sentence hard to read.

111: Missing full stop after Fig 3.

132: The methods for assessing growth are not well established, and as such should be given some further detail in the main text. The methods for predicting bacterial growth are potentially interesting, but it is hard to assess their veracity since no benchmarking of the experiment does not appear to have been undertaken and the only results presented (Extended data figure 6) are derived, not giving raw results. For instance, did the freeze treatments show a complete stoppage in growth, as might be expected?

Potentially supplementary data is available in the figshare, but this is not yet publicly accessible. The work has generated many datasets that would benefit from distribution as raw data (as opposed to the many plots presented). These data should be specifically cited within the manuscript, rather than leaving a reader to sort through what may be a large set of supplementary files in the figshare.

169-172: This long sentence is hard to read because “functional” is used in different ways, and “correlated” is repeated. It was unclear also how these conclusions relate to the “shift” as opposed simply to their state, since each correlation only considers one timepoint.

Fig 2: It isn't immediately obvious where the treatments are shown, that they are rings. It might be more obvious if the ring annotations were drawn in the middle of the page rather than the outer, so a line could be drawn between the legend to the icons within the rings.

Fig 2: Matching the colours is taxing for a reader. Perhaps it would be better if the phyla were simply annotated around the tree, with arcs showing their breadth. The order of the as alphabetical in the legend (rather than tree order) makes it hard to read, again this would be solved by putting the phyla on the tree.

Referee #2 (Remarks to the Author):

This is an interesting and valuable paper. There has been lots of interest in exploring the factors that regulate microbial community structure and function. How pulse perturbations alter dynamics has been a common theme of much of this research. Whether communities respond in common ways to different

perturbations is a particularly interesting question. One that this study addressed. It collected soils from 10 different regions in Europe, ranging from the sub-Arctic to the Mediterranean, and then ran them through four different perturbations: drying, freezing, flooding, and heating. It analyzed both the composition of the community and the functioning of communities and related these to each other. The four showed some common patterns in response, with drying, freezing, and heating all showing some parallels in their dynamics, while flooding tended to go in a different direction. Additionally, the responses to stresses showed some common patterns relative to a site's natural conditions—dry soils were more resistant to drying stresses, for example.

One question I had was about Figure 1 and a note on Line 743. The figure appears to show coherent patterns of response in the different microbial communities, but the caption says: “the total variance explained by all four constrained RDA axes is 2.7%, 3.5% and 7.4% for prokaryotes, fungi and shotgun metagenomes respectively.” So are we only discussing such a limited amount of variance? If so, that would be making something of a mountain out of a molehill—the lack of response across the treatments would then seem to be a larger story than the paper makes out. I'm sure I'm misreading this somehow but if so, something is unclear.

It is important to note, and they do, that the “origin of the sample (country and site) is the main driver of microbial community composition.” I wonder whether there are any patterns to this beyond climate—are there soil properties that can explain the patterns they observed? Clay content? Mineralogy? Or is it just climate as they suggest? Can we explain anything more about the patterns of community composition?

I also wonder what the effects of the physical disturbance was—this was a lab experiment done on sieved soils. Thus, the resource environment of the soils would have been substantially different from that found in undisturbed soils—sieving enhances C availability. I don't see this as a problem, as it would not have been possible to do this study on undisturbed soils, but the treatment might well have encouraged copiotrophs over oligotrophs.

I also found the argument that these findings will lead to “a better understanding of the microbial contribution to changes in ecosystem functioning, which is urgently needed for improving the accuracy of global C cycling models that currently do not take into account extreme events.” Is a bit specious—global C cycling models currently operate at a scale so far from that explored in this study, it's hard to see that this work will have any real relevance to such models. The real value of this study is in its fundamental soil biology and its exploration of microbial responses to environmental perturbations. This is terrific soil biology, but it's not what I'd consider Earth System Science.

A minor question: On line 316 they describe covering the microcosms as such: “covered in tin foil.” But I'll bet that it wasn't tin foil but aluminum foil. This is a minor point but they should be accurate.

Referee #3 (Remarks to the Author):

The publication by Devries et al. investigates the resistance and resilience of soil microbial communities to extreme events of heat, moisture, drought, and freeze. They collected soils from 10 geographically diverse sites, conducted a 4-week lab incubation, and used shifts in community diversity to assess resilience and resistance. They further explored correlations between shifts in diversity and environmental parameters. Their results suggest that microbiomes are most resilient and resistant to changes that are like their native conditions. These findings could be interesting, and the experimental design is solid. However, the impact of the findings and how it pushes the field forward are not clearly articulated. In general, the analyses are based on community diversity (Bray Curtis dissimilarity), without a strong justification or interpretation. The “small but highly consistent and phylogenetically conserved response under the imposed extreme events” only explains about 2% of the variation in amplicon data and at most 7.4% of the variation in the metagenomic data. Giving examples of what specific changes in the taxonomic and functional gene potential composition represent realized resistance or resilience of measured community activities would be a valuable addition. The experimental design is impressive and covers a wide range of soils (600 microcosms)! This is an important context that is not discussed. Finding universal responses among disparate and heterogeneous soils is an important contribution that may have implications for better understanding regional scale responses to disturbance. It would be valuable to explore this in the discussion and to clarify the potential impact of this work. Below are specific comments to help improve the manuscript.

Abstract:

L5 please specify this is lab study

” These findings fundamentally advance our understanding of soil microbial responses to extreme events, and improve our ability to implement their responses in global biogeochemical models.”

It would be helpful to support this statement with a specific model that would benefit from these data or findings.

L 38 – “responses in response to” is redundant

Extended Data Table 2. What about interaction effects?

L 77 the genes that underlie this response (e.g. osmoprotectants²³) - what specific genes? It would be wonderful to see a supplemental table with differentially expressed genes.

L Fig 2 Optional edit: I understand these figures are notoriously difficult to craft - yet it is challenging to see strong grouping of the strategies. Perhaps a different color scheme for the strategies, rather than shades of the same color would help.

L 116 defense misspelled

Extended data fig 6: this figure needs a bit more information in the caption/methods for readers to follow. Caption title: Is this prokaryotic data (16S amplicon) or just bacterial? Please add directly after the disturbance (S1) or after one month of recovery (S4) to the caption. Please also explain how boom and bust are interpreted from the figure.

L134 how are “copiotrophic communities’ being identified? Not in methods

L135 should this be actively growing fungal communities? “eight of the ten most actively growing communities were from Austria and Sweden, Extended Data Fig. 6a” It’s not obvious to me from the figure and caption how this statement is supported statistically. A bit more information will benefit a

broad audience.

L145 – It's not clear what evidence suggests that heat and flood favour organisms with a 'boom and bust' strategy. Perhaps instead favours communities, rather than organisms since it's a community metric?

Fig 4 b. what are 'functional categories from the metagenomics in the fig b? is this the list in extended fig 7? Are a and b both correlation coefficients? b says correlation.

L750 – categories from the metagenomics should be metagenomes; same on 752

L 155, "microbial communities lacked genes to withstand disturbances" –Is this correlation analysis indicating that functions are missing or that biodiversity confers resistance and resilience? It is not clear how a lack of gene presence can be inferred from the correlation of environmental parameters to Bray-Curtis distance matrix. It would be great to see evidence/examples of missing genes/functions.

L163 indicate that a high abundance of stress-tolerance genes – would be nice to see direct evidence of differences in abundance. What is the relative importance of abundance v the diversity metrics used?

L169 Please be more specific with the definition of "functional community composition". Is this the diversity of annotated genes from the metagenomes?

L 171 – this is a correlation of a correlation? Or typo? microbial functional community composition correlated significantly with measures of soil functioning correlated within the end-of-disturbance sampling (S1) and sampling after four weeks of recovery (S4) in each treatment.

L 182 Moreover, we show that the response of functional microbial community composition is linked to soil functioning and can be explained by traits present in these communities.

It's difficult to understand what this really means –response of functional microbial community composition is a shift in the beta diversity of functional potential – and this distance is linked to 'functioning' - which suggests gene expression or community activity. However extreme events did not affect enzyme activity or substrate use, but functional community composition was strongly correlated with enzyme activities and substrate-induced respiration. More discussion reconciling these results will be helpful. Greenhouse gas emissions and C and N availability had contrasting responses to enzymes and substrate induced respiration. What does this mean in terms of shifts in functional potential (functional genes from metagenomes) informing ecosystem resilience/ resistance? Are there specific genes /pathways where genomic potential may be more representative of functional (expressed) responses? How can this be generalized?

What traits – although traits are discussed in the introduction, there are no traits defined in the methods or results. Are you considering a gene or phenotype a trait?

What community function is linked to soil function, specifically?

L 186 – it's not clear from the discussion how this study improves our understanding of the microbial contribution to changes in ecosystem functioning. What specific functions of ecosystems does this inform? How would these data improve models (and which models)?

L 188 –fundamental insights aren't clear - Fundamental suggests we have identified causation.

L 253 was time equivalent for all treatments. In other words, how long did it take to dry samples for drought and was that drying time considered drought, or was drought started once 10% WHC was accomplished. What happened to the 'flood' while soils were being dried? Did freeze happen over time, or was this an instantaneous shift from 18 to -20?

L 342 – what was the sequencing depth?

L395 - it seems you have data on the abundance of genes encoding for SEED subsystems, which may be a more direct measure of genomic potential than the Bray Curtis dissimilarity. Please add reference to where the p values from this analysis (L402) were reported in the manuscript

L 411 – A bit more justification of this approach would be helpful. What is the question you are asking of the data – does “particular functional categories” mean that this was a targeted approach? Were specific functional categories of genes correlated with the associated biogeochemical measurements?

Author Rebuttals to Initial Comments:

The authors included a summary of the general points they addressed in the revisions. These are copied below and are referred to in their responses to the individual referees' comments.

1) A main concern was the small proportion of variance in microbial communities explained by our perturbations. As ably demonstrated in a paper published since our submission (Kimmel et al., 2023), now ref. 16), as ecologists we may have exaggerated expectations for effect sizes. We now clearly show that the variance in microbial community composition explained by our extreme events decreases when more sites/countries are included in the analysis, down to a small but consistent plateau only accessible to a study of this scale. This first demonstrates what is well known i.e. that the largest variability in soil microbial community composition is between sites (due to differences in climate, geology, human impact, biogeography etc.) which therefore also explains the largest fraction of the variance in this cross-site continental study. Moreover, this unambiguously highlights that, while we identify responses that are consistent across environments, for predicting the response of soil microbial communities to these events, we need to understand local responses. We now demonstrate that this is possible – explaining 58% of the variance in resistance and resilience of functional genes to extreme events based only on soils' climate and initial properties. We have modified the text to better highlight these findings and how they advance our understanding of soil response to extreme events.

2) We have added several analyses that add support to our main conclusions: This has resulted in augmenting one Extended data Table (EDTable 2) and one figure (now EDFig. 9) plus adding a further five more Extended data Figures. These include reporting an additional, independent analysis of bacterial growth potential to support our analysis of realised bacterial growth, as well as more in-depth analyses both determining the functional genes that are driving the correlations of metagenomic community composition and measured soil functions, and explicitly predicting soil responses based only on their initial properties and climate.

3) When running these additional statistical analyses, we discovered that two of our 620 samples had been swapped in the original analyses, namely Iceland Control sample 1 at time point S1 and Russian Flood sample 2 at time S1. We have corrected this and have redone all analyses, which has resulted in minor changes in our results. Specifically, they have resulted in a stronger and more consistent impact of heat, which corresponds with the growth analysis results, and points more consistently to a shift towards more copiotrophic communities.”

Referees' comments:

Referee #1 (Remarks to the Author):

The presented manuscript details an investigation into the effect of 4 different climate stressors on grassland soils from across Europe, using a combination of amplicon (fungal and bacterial), metagenomics, soil geochemistry and enzymatics to characterise incubations. The manuscript is mostly well written, reflecting the straightforward experimental design consistently applied across the grassland sites. However, I found several of the conclusions of the study to be unsupported by the evidence presented.

In particular, the conclusion in the abstract that heat “decreased .. protein biosynthesis genes” seems unlikely. This statement is made through interpretation of a read-centric analysis of metagenomes. This data is not a reflection of gene expression but of genomic potential, and thus only changes when community members grow or die. But since protein synthesis is a universal function, a decrease in the relative abundance of these genes more likely points to an artefact of the functional gene analysis or its interpretation.

Response: Of course, protein biosynthesis genes perform a universal function, but that does not preclude great variation in their copy number between microbial species. We also agree that our metagenomic functional analysis, including protein biosynthesis genes, is about genomic potential, which is why we have complemented and compared it with direct functional measures and estimates of actual growth (Fig. 4, now ED Figs. 7 and 9a-b).

Within the ‘protein metabolism’ classification, ribosomal genes are the most consistently decreased under heat, so largely responsible for the change in protein biosynthetic genes referred to. At the same time, rRNA gene copy number is one of the best characterised cases of essential gene copy number variation, where particular bacteria may have anything from one to at least 16 copies of this essential operon. That variation relates to the maximum growth capacity of the organisms (Roller et al., 2016). Furthermore, when assayed at a community level, that variation is functionally meaningful across both space and time (e.g. Abreu et al., 2023). We have therefore conducted an additional analysis of rRNA gene copy number.

Specifically, to test whether our more general analysis of change in relative abundance of protein biosynthesis genes represents the specific rRNA proxy of maximum growth capacity in the community, we have now applied a state of the art rRNA copy number tool (enabling weighting by certainty) to our datasets (Gao & Wu, 2023). Accounting for 16S rRNA copy number prediction uncertainty and its implications in bacterial diversity analyses. ISME Commun 3, 59. 10.1038/s43705-023-00266-0.). This new analysis showed that the heat treatment did indeed show a consistent decrease in the copy numbers of these genes directly after perturbation (new ED Fig. 9c, rRNAvTimeAbs2_Gaow), consistent with

reduced growth capacity, as identified in both the broad functional analysis (Fig. 3) and the direct growth estimates (now ED Fig. 9b).

In addition, to clarify this in the manuscript we have both augmented the text (including the citations above) and added the finer-scale functional analysis (illustrating the effect on ribosomal proteins among others) in two new sets of data figures (Extended Data Figures 17 and 18, 56 in all, covering the resistance and resilience of the individual proteins in each of the highest-level functional categories) available via the FigShare site (<https://figshare.com/s/48310a0929e4a875f5b8>). This demonstrates that

the functional gene analysis gives a detailed and nuanced view, consistent with the other data. For example, this is what it shows for the impact of heat on the relative abundance of large ribosomal subunits:

This demonstrates that many different bacterial large ribosomal subunit (LSU) proteins (one protein per row, blue are decreases and red increases relative to control, represented by the vertical bar; filled figures are significant following correction for multiple testing) are indeed significantly reduced by the heat treatment, whereas the eukaryotic and archaeal equivalents are not. It is notable that the one bacterial protein that bucks the trend and is significantly enriched following heat (the red point in the bottom of the blue section), is the one LSU annotated specifically to the Firmicutes. This is consistent with the 16S data, specifically our observation that Firmicutes, relatively, are positively impacted by the heat treatment (Fig. 2a). Together, these results suggest that our functional observations of protein metabolism do not biased by problems with annotation.

As well, the abstract concludes that “heat most strongly impacted soil microbiomes”, but it isn’t clear how the different stressors can be compared, since the specific scale of each pressure is not derived with respect to future climate predictions.

Response: The reviewer is correct that our extreme events are not scaled to future climate predictions, which would not be possible when keeping the extreme events consistent, and thus comparable, across all sites/countries. Thus, we designed these extreme events to be “extreme” for the soils across all sampled environments, based on treatments used in our previous experiments and those of others, as well as information about the variability of soil temperatures and moisture. Drought treatments commonly dry the soil to a moisture content equal to 10% of the soil’s water holding capacity, which severely limits moisture availability for plants and soil organisms (e.g. De Vries et al., 2016, De Vries et al., 2012). Freezing temperatures of -20°C do not commonly occur even in Arctic soils (Convey et al., 2018), while soil temperatures of 35°C are rarely exceeded even in semiarid systems and during extreme

heatwaves (Rehschuh et al., 2022, Sviličić et al., 2016). Flooding is not scalable, i.e. fully submerging soil is fully submerging soil. Thus, subjecting soils to two weeks of these treatments continuously would be considered extreme for any soil from any climate. We have now included more detail on our rationale for these treatments in the Methods and Materials, see lines 265-268. Regardless, we still think our statement that the heat treatment impacted the soil microbiomes most is valid. First, the arrows that indicate the direction of the heat treatment response are longest (Fig. 1), second, a larger number of taxa is impacted by heat than by the other treatments (Fig. 2), and third, most gene categories were significantly impacted by heat (Fig 3a). We agree that this doesn't necessarily mean that heat in general impacts microbiomes most, and therefore we have changed our wording to "the heat treatment" when referring to this strongest impact (e.g. line 139).

I also found the conclusion that the "magnitude of this response was dampened in microbiomes from soils that are prone to conditions similar to the exposed extreme events" somewhat problematic. Observation of resilience/resistance to heat and drought in samples from warmer and drier environments is not precisely equivalent to the above, since the climates of the different regions are differently variable. Also, this conclusion was only reached for 2 of the 4 treatments, with 1 (freeze) showing the opposite correlation (line 152), so a general conclusion about climate extremes seems inappropriate.

Response: We agree that the correlations of responses to heat and drought in Fig. 4a are the most striking, and that, as the reviewer points out, variability matters. This latter is reflected in the different associations we show for maximum, minimum and mean values, particularly for temperature with heat and drought.

We were previously clear about the fact that drought and freeze had similar effects due to their mediation through reducing water availability (now line 42-44). Coupling that with the fact that climate conditions similar to our heat and drought effects are likely to be associated during combined drought-heat events makes our expectation for the climate associations of freeze more complicated than a simple expectation of soils from cold climates showing less response to freeze. This is reflected in the much more limited simple correlations for freeze in Fig. 4a and does mean that our previous statement, which the reviewer highlights, was unduly simplistic.

However, to address this issue more directly, we have now augmented Fig. 4a with a less simplistic random forests model (extended data Fig. 14) of the same data. This shows that these climate and soil properties do explain the magnitude of resistance and resilience across perturbations, not just heat and drought (explaining 61% of the variance overall, 58% \pm 4% by cross-validation on unseen data), now shown in Extended data Fig. 14a. This model clarifies that there is a strong contrast both between the

associations of heat and freeze with the thermal climate (particularly among warmer climates Extended data Fig. 14b) and the associations of drought and flood with soil water holding content (particularly in soils with low WHC Extended data Fig. 14c). This is consistent with our statement of a dampened response in soils from conditions more similar to our extreme event treatments, just not in the simplistic way of the one-by-one correlations we can show in Fig. 4a. We have therefore modified the main text to acknowledge the nuance the reviewer rightly highlights, backed up by the new models in ED Fig. 14.

On top of this, since the “heat” condition was a consistent 35 degrees C for all samples, samples from cooler environments had a larger scale of treatment (more degrees warmed), so are subjected to a more extreme environmental change than those from warm environments.

Response: We fully agree with the reviewer that the extent of the disturbance is not the same for the communities from different environments and this is indeed part of our hypothesis: we expected communities adapted to the type of extreme event that more frequently occurs in their climate of origin to be better able to withstand that disturbance through the selection of traits that confer resistance to the extreme event. While nuanced (as set out in the response to the previous point), this is what we find. However, as outlined in the response to the point before that, these extreme events are still extreme in the environments that are more likely to experience them, and, as our results show, still affect the composition of the microbial community and cause a shift in 16S, ITS, and metagenomic profiles (ED Table 2).

We now present these changes by country more explicitly in the context of the RDA plots (Fig. 1) and in the new Extended Data Fig. 1 (parts d-f) showing, for instance, that the relatively cold and wet UK Lancaster site was consistently strongly affected by heat, but much less so by flood.

I had also had several minor concerns:

15: “share” was confusing to this reader. Did this mean across soil samples? Or across treatments too?

Response: The first: this meant a unified response to each treatment shared across soil samples from different environments. We have now reworded this sentence to make this clearer (lines 12-13).

76, 82-24: Resistance and resilience are defined here for the purposes of differentiating between different types of responses. But, these two words are very similar to each other. Can one of them be replaced by a more distinct word? This would aid readability.

Response: We acknowledge that the words may be similar, but these two concepts are widely and generally used in ecology to describe the initial ability of a system or property to withstand a disturbance and the speed and extent to which a system or property recovers after a disturbance. See for example (De Vries et al., 2018, Pimm, 1984, Orwin & Wardle, 2004, De Vries et al., 2012, De Vries & Shade, 2013, Lavallee et al., 2024, Bardgett & Caruso, 2020). For consistency with previous work, we prefer to keep these terms as they are coined in ecology, i.e. resistance and resilience. We have now explained these terms better in the manuscript in line 93-96.

102-103: I found this sentence hard to read.

Response: We have clarified this sentence, which now reads “These findings are in contrast with previous work that shows little difference in phylogenetic conservation between microbial response to different global change drivers, and between fungi and bacteria” (now line 114-116)

111: Missing full stop after Fig 3.

Response: Inserted.

132: The methods for assessing growth are not well established, and as such should be given some further detail in the main text. The methods for predicting bacterial growth are potentially interesting, but it is hard to assess their veracity since no benchmarking of the experiment does not appear to have been undertaken and the only results presented (Extended data figure 6) are derived, not giving raw results. For instance, did the freeze treatments show a complete stoppage in growth, as might be expected?

We have now added further detail in the main text as requested (line 146-160). Peak-to-trough ratios in genomic sequence are well established as a means of assessing bacterial growth in both isolates (Skovgaard et al., 2011) and mixed communities (Brown et al., 2016, Emiola & Oh, 2018, Korem et al., 2015) – these references go back to 2011 and 2015 and are all cited in the main text. The use of dnaA and dif sequences as indicative of the origin (peak of copy number) and terminus (copy number trough), respectively, which we employed in this study, has been developed and validated in this context in original ref 58. Direct benchmarking of growth in such complex samples is not possible, but we have now included an independent assay (from 16S rather than metagenome data) of community growth capacity (via expected rRNA copy number, see above) in Extended Data Fig. 9, which helps contextualise these growth estimates. This shows, as expected, relatively little change in either growth or growth capacity proxy directly after freezing.

Potentially supplementary data is available in the figshare, but this is not yet publicly accessible. The work has generated many datasets that would benefit from distribution as raw data (as opposed to the many plots presented). These data should be specifically cited within the manuscript, rather than leaving a reader to sort through what may be a large set of supplementary files in the figshare.

Response: All our raw data used for this manuscript are already freely available for download right now, as specified in the data availability statement. We included a private link in the manuscript for the reviewers to access the processed data on figshare before we make it publicly available. All processed data and code will be freely accessible and downloadable on figshare when the manuscript is published.

169-172: This long sentence is hard to read because “functional” is used in different ways, and “correlated” is repeated. It was unclear also how these conclusions relate to the “shift” as opposed simply to their state, since each correlation only considers one timepoint.

Response: We have rewritten this sentence to be clearer on the difference between metagenomes (functional community composition) and direct measures of soil functioning, see line 185. The reviewer is indeed correct that these correlations do not refer to shifts in functions and metagenomes induced by treatments but refer to correlations across soils and within treatments – we have changed the wording to more accurately reflect this (line 184-187).

Fig 2: It isn't immediately obvious where the treatments are shown, that they are rings. It might be more obvious if the ring annotations were drawn in the middle of the page rather than the outer, so a line could be drawn between the legend to the icons within the rings.

Response: We have now modified this figure and drawn arrows between the legend icons and the rings. We hope these changes make this figure easier to read.

Fig 2: Matching the colours is taxing for a reader. Perhaps it would be better if the phyla were simply annotated around the tree, with arcs showing their breadth. The order of the as alphabetical in the legend (rather than tree order) makes it hard to read, again this would be solved by putting the phyla on the tree.

Response: In addition to drawing arrows from the treatment icons and the rings, we have also plotted the names of the phyla directly on the tree.

Referee #2 (Remarks to the Author):

This is an interesting and valuable paper. There has been lots of interest in exploring the factors that regulate microbial community structure and function. How pulse perturbations alter dynamics has been a common theme of much of this research. Whether communities respond in common ways to different perturbations is a particularly interesting question. One that this study addressed. It collected soils from 10 different regions in Europe, ranging from the sub-Arctic to the Mediterranean, and then ran them through four different perturbations: drying, freezing, flooding, and heating. It analyzed both the composition of the community and the functioning of communities and related these to each other. The four showed some common patterns in response, with drying, freezing, and heating all showing some parallels in their dynamics, while flooding tended to go in a different direction. Additionally, the responses to stresses showed some common patterns relative to a site's natural conditions—dry soils were more resistant to drying stresses, for example.

Response: We thank the reviewer for their positive evaluation, and we are happy to read that the reviewer likes our experimental approach in addressing the factors that regulate microbial community response to extreme events.

One question I had was about Figure 1 and a note on Line 743. The figure appears to show coherent patterns of response in the different microbial communities, but the caption says: “the total variance explained by all four constrained RDA axes is 2.7%, 3.5% and 7.4% for prokaryotes, fungi and shotgun metagenomes respectively.” So are we only discussing such a limited amount of variance? If so, that would be making something of a mountain out of a molehill—the lack of response across the treatments would then seem to be a larger story than the paper makes out. I'm sure I'm misreading this somehow but if so, something is unclear.

Response: We thank the reviewer for this comment. The reviewer is correct that the proportion of variation explained by our four extreme events is very low across our wide gradient of environmental variation. However, we would like to stress that this is not making a mountain of a molehill. Instead, we argue that our study design enables us to more accurately represent the impact of the extreme events we applied. We now provide additional analyses to emphasize this point in our manuscript.

Previous experiments assessing the impacts of extreme events have tended to find large (functional) shifts in soil microbial community composition, because generally experiments focus on one site or ecosystem. Most notably, such studies have considered (mostly) drought (only) within a system or a soil type, and sometimes along a precipitation or temperature gradient within that system. As a result,

variation in communities and traits within that system is limited, as are sample sizes, potentially contributing to exaggeration bias both through low power (Kimmel et al., 2023) and also through effects in that system that are specific to that system (effectively the Treatment × soil interaction, which we now detail explicitly in Extended Data Table 2 and Extended Data Fig. 2). However, if we want to fundamentally understand how microbial communities respond to these extreme events, we need to look across systems. We show, that in the light of the enormous variation in microbial communities across our diverse soil systems, the shifts in microbial community composition are small, yet highly consistent, allowing us to not only infer which traits underlie these responses, but also give context to the much stronger effects within systems, and provide a more accurate effect size. We now also very convincingly show, in Extended Data Figure 3, that the more countries are included in our analysis, the smaller, but more consistent, the variation explained by the extreme events becomes. We show this consistently for 16S, ITS, and metagenomes. Thus, our study puts into context the shifts that are caused by extreme events, and emphasises that local effects on soil microbiomes cannot be used to predict global effects. We have updated the text to reflect better the importance of the differences in variance explained when including a higher number of countries, and emphasise that this result means that observations from a single site or system cannot be used to predict general patterns (line 15, line 83-85 and line 202-204). In addition, we have also changed the title to better reflect our findings, which now reads “Soil microbiomes show consistent and predictable responses to extreme events across climates”.

It is important to note, and they do, that the “origin of the sample (country and site) is the main driver of microbial community composition.” I wonder whether there are any patterns to this beyond climate—are there soil properties that can explain the patterns they observed? Clay content? Mineralogy? Or is it just climate as they suggest? Can we explain anything more about the patterns of community composition?

Response: We agree with the reviewer that certainly there are interesting patterns to show on the factors that shape soil microbial communities beyond climate (e.g. the initial properties reported in Fig. 4 and what is now Extended Data Fig. 7). Nonetheless, this was not the focus of our study and global patterns in microbial communities and the soil and climatic factors that drive them have been extensively reported in previous studies (Delgado-Baquerizo et al., 2016, Fierer & Jackson, 2006, Tedersoo et al., 2014), which we cite in line 72. The focus of our study was to use soil and climatic properties to explain microbial community responses to extreme events. We now augment that analysis with a random forests model (Extended Data Fig. 14) that requires soil properties beyond those of climate (removing these reduces the coefficient of determination, R^2 , by 5%), but does not require any further information about place (i.e. including information about country gives no improvement in the coefficient of determination; $R^2 = 0.58$ by cross-validation in either case). We discuss these issues in the main text in line 166-177.

I also wonder what the effects of the physical disturbance was—this was a lab experiment done on sieved soils. Thus, the resource environment of the soils would have been substantially different from that found in undisturbed soils—sieving enhances C availability. I don't see this as a problem, as it would not have been possible to do this study on undisturbed soils, but the treatment might well have encouraged copiotrophs over oligotrophs.

Response: This is why we included an acclimation period in the controlled growth chambers, and why we compare the impact of extreme events not to the 'before disturbance' situation, but to a control that also changes over time after sieving and under the controlled conditions. It was also why we sequenced initial samples, taken before acclimation. These initial samples generally had communities very similar to those in the acclimated and treated versions of the same soils, as we now show explicitly in Extended Data Fig. 1a-c.

I also found the argument that these findings will lead to “a better understanding of the microbial contribution to changes in ecosystem functioning, which is urgently needed for improving the accuracy of global C cycling models that currently do not take into account extreme events.” Is a bit specious—global C cycling models currently operate at a scale so far from that explored in this study, it's hard to see that this work will have any real relevance to such models. The real value of this study is in its fundamental soil biology and its exploration of microbial responses to environmental perturbations. This is terrific soil biology, but it's not what I'd consider Earth System Science.

Response: We agree with the reviewer that our results have limited relevance to Earth system models (and we are delighted that the reviewer thinks this is terrific soil biology!), but are a first step in understanding how soils from different environments will respond to extreme events in their functioning. We have modified our abstract (line 12-17) and conclusions (line 197-207) to better represent the advances that our study provides.

A minor question: On line 316 they describe covering the microcosms as such: “covered in tin foil.” But I'll bet that it wasn't tin foil but aluminum foil. This is a minor point but they should be accurate.

Response: We thank the reviewer for their attention to detail - we have changed this to aluminium foil!

Referee #3 (Remarks to the Author):

The publication by Devries et al. investigates the resistance and resilience of soil microbial communities

to extreme events of heat, moisture, drought, and freeze. They collected soils from 10 geographically diverse sites, conducted a 4-week lab incubation, and used shifts in community diversity to assess resilience and resistance. They further explored correlations between shifts in diversity and environmental parameters. Their results suggest that microbiomes are most resilient and resistant to changes that are like their native conditions. These findings could be interesting, and the experimental design is solid. However, the impact of the findings and how it pushes the field forward are not clearly articulated. In general, the analyses are based on community diversity (Bray Curtis dissimilarity), without a strong justification or interpretation. The “small but highly consistent and phylogenetically conserved response under the imposed extreme events” only explains about 2% of the variation in amplicon data and at most 7.4% of the variation in the metagenomic data. Giving examples of what specific changes in the taxonomic and functional gene potential composition represent realized resistance or resilience of measured community activities would be a valuable addition. The experimental design is impressive and covers a wide range of soils (600 microcosms)! This is an important context that is not discussed. Finding universal responses among disparate and heterogeneous soils is an important contribution that may have implications for better understanding regional scale responses to disturbance. It would be valuable to explore this in the discussion and to clarify the potential impact of this work. Below are specific comments to help improve the manuscript.

Response: We thank the reviewer for the positive comments on our solid and impressive experimental design and the importance of our results in understanding soil responses to disturbance. We agree that we could have highlighted the impact of our findings and how they push the field forward better, as also highlighted by reviewer 2, which we have now done in the abstract and the conclusion, line 12-17 and line 197-207. We have also now more explicitly highlighted the scale of our experiment in the main text, line 66-67.

In response to the reviewer’s comment about diversity analyses: we quantify the importance of country and imposed extreme events in explaining variation in 16S, ITS, and metagenome community composition. We do this using analyses based on Bray-Curtis dissimilarities, which is an appropriate and very commonly used method to quantify differences in community composition (beta-diversity). In addition, we use Bray-Curtis dissimilarities to quantify the extent to which the (16S, ITS, and metagenome) communities have changed in response to our extreme events, i.e. how dissimilar communities affected by extreme events are from control communities. We have now added references for this approach, which is also commonly used for this purpose (De Vries et al., 2018, Shade et al., 2011) in line 430-433. Where our analyses use distances for other purposes (notably Fig. 4b) we have used Euclidian rather than Bray Curtis distance.

We have addressed the fact that our extreme events explain a small fraction of variation in our communities in the response above to Reviewer 2. We now clearly show that the amount of variance

explained by our RDA decreases with the number of countries sampled, while asymptoting to a consistent level (Extended Data Fig. 3). We show, that in the light of the enormous variation in microbial communities across our diverse soil systems, the shifts in microbial community composition are very small, yet highly consistent, allowing us to not only infer which traits underlie these responses, but also give context to the much stronger, but more variable, effects reported at one site. Our results emphasise that observed local shifts in microbial communities cannot be used to predict global shifts, and show that local soil and climatic properties can be used to predict shifts in microbial communities in response to extreme events (new Extended Data Figure 14).

Finally, in response to the reviewer's comment that "examples of what specific changes in the taxonomic and functional gene potential composition represent realized resistance or resilience of measured community activities would be a valuable addition": this is exactly what we show in Figure 3, where changes in abundance of protein functions with extreme climatic events are displayed. We have now added further detail and nuance to this in Extended Data Figures 15 and 16, available on Figshare.

Abstract:

L5 please specify this is lab study

Response: We have now added "common" such that the abstract now details that these soils were exposed to extreme events under "common controlled conditions", line 6

" These findings fundamentally advance our understanding of soil microbial responses to extreme events, and improve our ability to implement their responses in global biogeochemical models." It would be helpful to support this statement with a specific model that would benefit from these data or findings.

Response: We have modified this sentence in the abstract and also in the conclusion of the manuscript to better reflect the advance our findings provide (see response to Reviewer 2 above about global models) and have removed references to global biogeochemical models.

L 38 – "responses in response to" is redundant

Response: this sentence has been modified

Extended Data Table 2. What about interaction effects?

Response: The Treatment × Sampling time interaction was included because this is a focus in the figures. However, we have now included all other possible interactions in Extended Data Table 2 and summarised them in a new figure (Extended Data Fig. 2). Including these interactions has little effect on the existing values, but links to the new Extended Data Fig. 1 d-f where the Treatment × Country interaction is apparent .

L 77 the genes that underlie this response (e.g. osmoprotectants23) - what specific genes? It would be wonderful to see a supplemental table with differentially expressed genes.

Response: As we noted above in our response, the impact of our imposed extreme events on gene abundances of the 28 highest level functions are detailed in Figure 3. In addition, we have now uploaded figures for all the significant individual responses of the lower level functions within these 28 broad categories in figshare (this considers all 10,408 individual functional annotations that we identified in these samples). The (private) link to all figshare data is available for the reviewers (line 736-738). As a side note, the specific phrase the reviewer refers to here has now been deleted.

L Fig 2 Optional edit: I understand these figures are notoriously difficult to craft - yet it is challenging to see strong grouping of the strategies. Perhaps a different color scheme for the strategies, rather than shades of the same color would help.

We have made the circular diagrams easier to read by plotting the phyla names directly on the tree, as well as including arrows for which ring represents which treatment. We have kept the colour scheme for the strategies though as this emphasises the first point from Fig. 2, which is the different directions of change. We have reordered the text to make the importance of this point clearer. The secondary issue of the individual strategies encoded by the admittedly relatively subtle differences in shade are separated much more explicitly in Extended Data Fig. 4.

L 116 defense misspelled

Response: We have consistently used UK English spelling, thus defence is correctly spelled.

Extended data fig 6: this figure needs a bit more information in the caption/methods for readers to follow. Caption title: Is this prokaryotic data (16S amplicon) or just bacterial?

Response: ED Fig. 6 (now ED Figure 9) is only bacterial, because only bacteria have single origin to terminus replication. However it is taken from the metagenome data (where we have origins and termini) rather than the 16S amplicon data. We have now included a separate analysis of growth capacity using this prokaryotic data (ED Fig. 9c) and clarified the captions and methods for both of them.

Please add directly after the disturbance (S1) or after one month of recovery (S4) to the caption.

Response: The times of each sampling point (coded by colour or facet in the figure) are now included in the legend.

L134 how are “copiotrophic communities’ being identified? Not in methods

Response: We have now included a definition of copiotrophic and oligotrophic strategies in line 134-138.

Please also explain how boom and bust are interpreted from the figure.

Response: Our identification of ‘boom and bust’ copiotrophic communities originates primarily in what is now Extended Data Fig. 7 (clustering of associations between particular functions and soil properties/climates). In this figure (now Extended Data Fig. 9) we see a shift towards increased growth capacity (panel c) and increased realised growth (panel b) across treatments, but we see specifically high growth bursts (boom) in some of the communities already identified as copiotrophic, notably the Austrian and Swedish soils following heat treatment (the blue points at the top of the right hand facets).

L135 should this be actively growing fungal communities? “eight of the ten most actively growing communities were from Austria and Sweden, Extended Data Fig. 6a” It’s not obvious to me from the figure and caption how this statement is supported statistically. A bit more information will benefit a broad audience.

Response: This continues to refer to the bacterial growth in Extended Data Fig. 9a (previously ED Fig. 6a), which is coloured by country, demonstrating that the highest points are mostly Austrian and Swedish.

L145 – It's not clear what evidence suggests that heat and flood favour organisms with a 'boom and bust' strategy. Perhaps instead favours communities, rather than organisms since it's a community metric?

Response: We have referred to copiotrophic strategies, organisms and communities reflecting the different scales at which we have connected data. The specific point highlighted is referring to growth data now in Fig. 9 where we have information about the growth of particular members of the community (the bacteria), hence the reference to organisms.

Fig 4 b. what are 'functional categories from the metagenomics in the fig b? is this the list in extended fig 7? Are a and b both correlation coefficients? b says correlation.

Response: The 'functional categories' used in both 4a and 4b are the relative abundance of the finest scale functional categories (proteins) from the metagenomics. This is now made clearer in the legend. Both 4a and 4b are correlations, but 4a are simply correlations of individual variables, whereas 4b are correlations of distance matrices (Mantel correlations). The meaning and underlying drivers of the correlations in 4b have now been explored further in Extended Data Fig. 14.

L750 – categories from the metagenomics should be metagenomes; same on 752

Response: Changed.

L 155, "microbial communities lacked genes to withstand disturbances" –Is this correlation analysis indicating that functions are missing or that biodiversity confers resistance and resilience? It is not clear how a lack of gene presence can be inferred from the correlation of environmental parameters to Bray-Curtis distance matrix. It would be great to see evidence/examples of missing genes/functions.

Response: We have, in fact, removed this sentence. But in response to the reviewer's comment on correlations between gene abundances and functions, we have now deconstructed how the correlations

of distance matrices relate to individual genetic and measured community-level functions in Extended Data Fig. 15, and added text on these findings in line 189-196.

L163 indicate that a high abundance of stress-tolerance genes – would be nice to see direct evidence of differences in abundance. What is the relative importance of abundance v the diversity metrics used?

Response: This specific sentence has now changed to highlight better our updated results, which now point to a higher importance of a range of metabolic genes, see line 177-180. We are here literally linking the abundance of specific genes to community resistance and resilience to extreme events (Extended Data Fig. 10), thus highlighting genes that are potentially driving this response, testing our hypothesis that specific genes are important in this response, and answering the reviewer's question.

L169 Please be more specific with the definition of “functional community composition”. Is this the diversity of annotated genes from the metagenomes?

Response: We agree with the reviewer that this was somewhat confusing. We have now modified this sentence, which now reads: “Importantly, we found a direct link between metagenome structure and soil functioning.” (line 184-185).

L 171 – this is a correlation of a correlation? Or typo? microbial functional community composition correlated significantly with measures of soil functioning correlated within the end-of-disturbance sampling (S1) and sampling after four weeks of recovery (S4) in each treatment.

Response: This was indeed a typo and the second ‘correlation’ has been removed.

L 182 Moreover, we show that the response of functional microbial community composition is linked to soil functioning and can be explained by traits present in these communities.

It's difficult to understand what this really means –response of functional microbial community composition is a shift in the beta diversity of functional potential – and this distance is linked to ‘functioning’ - which suggests gene expression or community activity. However extreme events did not affect enzyme activity or substrate use, but functional community composition was strongly correlated with enzyme activities and substrate-induced respiration.

Response: The reviewer is correct that shifts in community composition of metagenomes, i.e., based on shifts in abundance of functional gene categories, are linked to shifts in soil functioning, which was also transformed into a community metric, i.e., based on all process rates together for the different categories (see Methods lines 438-447). However, the reviewer is not correct in stating that extreme events did not affect enzyme activities or substrate use – we stated in our manuscript “less so enzyme activities and the ability to use different substrates” (line 182-183). Moreover, to clarify that functioning was affected by extreme events, we have now updated Extended Data Table 2 with the PERMANOVA results of all combined measures of soil functioning, as detailed in Methods line 374-379, which convincingly shows that extreme events did have a significant impact on soil processes.

Importantly, the correlations that are presented in Fig 4b are Mantel correlations of metagenomes and measured functions within treatments and sampling points, and thus these correlations are not driven by treatment effects.

More discussion reconciling these results will be helpful. Greenhouse gas emissions and C and N availability had contrasting responses to enzymes and substrate induced respiration. What does this mean in terms of shifts in functional potential (functional genes from metagenomes) informing ecosystem resilience/ resistance? Are there specific genes /pathways where genomic potential may be more representative of functional (expressed) responses? How can this be generalized?

Response: We suspect that the reviewer here is pointing to the different signs of the correlations in Fig. 4b. However, we would like to respectfully point out that these correlations are Mantel correlations of the distance matrices of metagenomes and soil functioning, and thus not directly indicative of particular functional associations. Our Fig. 4b shows that correlations between metagenome profiles and soil functioning are simply less strong for gas fluxes and soil C and N. We now address the reviewer’s questions on which specific genes are responsible for changes in functions explicitly in Extended Data Fig. 14, demonstrating that there are indeed specific functional categories of genes that are more and less representative of expressed functional responses. In particular, photosynthesis functions and, intriguingly, ‘phage’ seem more closely associated than other functional categories with the observed positive correlations.

What traits – although traits are discussed in the introduction, there are no traits defined in the methods or results. Are you considering a gene or phenotype a trait?

Response: Traits are organism properties that are determined by (sets of) genes, of which we give examples in the introduction, in line 24-27. Because in our analyses we focus on specific genes, after the introduction we use the word genes rather than traits.

What community function is linked to soil function, specifically?

Response: We have now explored correlations between gene abundances and soil functions in more depth in Extended Data Fig. 15, which highlighted that in particular, the gene categories of photosynthesis, phages and virulence were important in driving these correlations. We have also added new text on this in the main manuscript, line 189-196.

L 186 – it's not clear from the discussion how this study improves our understanding of the microbial contribution to changes in ecosystem functioning. What specific functions of ecosystems does this inform? How would these data improve models (and which models)?

Response: These sentences in the conclusion, as well as in the abstract, have now been modified to more accurately highlight the advance that our study represents.

L 188 –fundamental insights aren't clear - Fundamental suggests we have identified causation.

Response: We have removed the word fundamental.

L 253 was time equivalent for all treatments. In other words, how long did it take to dry samples for drought and was that drying time considered drought, or was drought started once 10% WHC was accomplished. What happened to the 'flood' while soils were being dried? Did freeze happen over time, or was this an instantaneous shift from 18 to -20?

Response: All treatments were imposed for two weeks. Drought was imposed by withholding watering from the start of the drought. It took a few days for the soils to reach 10%WHC, after which that weight was maintained for the remaining time of the two weeks. Flood was instantaneous, as microcosms were submerged in water, but similarly, after ending the flood, it took a while before the soils dried out to the control level of 60%WHC. Freeze and heat are instantaneous, as the microcosms were transferred to a freezer and growth chamber, respectively, but again, these soils would also take a while to reach the

extreme temperatures as well as control temperatures of 18° C. This is all detailed in the Methods, line 268-281, but we have now included a figure that shows soil moisture, expressed as percent of 100%WHC, of the treatments, but specifically of the drought and the flood treatment (Extended Data Fig. 16).

L 342 – what was the sequencing depth?

Response: Read depth typically varies per sample, so we resampled each sample to a standardised number of reads as reported in line 386-388 – these rarefied data were used for the NMDS and PERMANOVA, as detailed in the line preceding this.

Read numbers were: metagenomes 6,398,524 (median, IQR: 4,704,883 – 8,216,799);

16S (32,200; 25,631 – 38,508 median; IQR); ITS (38,410; 31,831 – 44,264 median; IQR) - this is now detailed in line 370 and line 355-356.

L395 - it seems you have data on the abundance of genes encoding for SEED subsystems, which may be a more direct measure of genomic potential than the Bray Curtis dissimilarity. Please add reference to where the p values from this analysis (L402) were reported in the manuscript

Response: We did indeed analyse how the abundance of genes within SEED subsystem categories at all four levels were affected by our treatments; this is detailed at level 1 and level 4 in Figure 3a and 3b respectively, which includes the effect sizes as well as the false-discovery rate corrected p-values, relative to a threshold of $P = 0.05$, for the change in gene abundances of each category. All scales are now included in the Extended Data Figures 15 and 16, available via Figshare.

L 411 – A bit more justification of this approach would be helpful. What is the question you are asking of the data – does “particular functional categories” mean that this was a targeted approach?

Response: The analysis described in line 411 is that reported and justified in Fig. 4b and its associated text. The reference ‘particular’ functional categories (original line 408) referred to the separation into each of the 28 highest level functional categories in what is now Extended Data Fig. 7. We have now clarified this language.

Were specific functional categories of genes correlated with the associated biogeochemical measurements?

Response: Yes. And, we have now explored these correlations in more depth in Extended Data Fig. 15, which highlighted that in particular, the gene categories of photosynthesis, phages and virulence were important in driving these correlations. We have also added new text on this in the main manuscript, line 189-196.

References

- Abreu, C. I., Dal Bello, M., Bunse, C., Pinhassi, J. & Gore, J. (2023) Warmer temperatures favor slower-growing bacteria in natural marine communities. *Science Advances*, **9**, eade8352.
- Bardgett, R. D. & Caruso, T. (2020) Soil microbial community responses to climate extremes: resistance, resilience and transitions to alternative states. *Philosophical Transactions of the Royal Society B: Biological Sciences*, **375**, 20190112.
- Brown, C. T., Olm, M. R., Thomas, B. C. & Banfield, J. F. (2016) Measurement of bacterial replication rates in microbial communities. *Nature Biotechnology*, **34**, 1256-1263.
- Convey, P., Coulson, S. J., Worland, M. R. & Sjöblom, A. (2018) The importance of understanding annual and shorter-term temperature patterns and variation in the surface levels of polar soils for terrestrial biota. *Polar Biology*, **41**, 1587-1605.
- De Vries, F. T., Brown, C. & Stevens, C. J. (2016) Grassland species root response to drought: consequences for soil carbon and nitrogen availability. *Plant and Soil*, 1-16.
- De Vries, F. T., Griffiths, R. I., Bailey, M., Craig, H., Girlanda, M., Gweon, H. S., Hallin, S., Kaisermann, A., Keith, A. M., Kretzschmar, M., Lemanceau, P., Lumini, E., Mason, K. E., Oliver, A., Ostle, N., Prosser, J. I., Thion, C., Thomson, B. & Bardgett, R. D. (2018) Soil bacterial networks are less stable under drought than fungal networks. *Nature Communications*, **9**, 3033.
- De Vries, F. T., Liiri, M., Bjørnlund, L., Bowker, M., Christensen, S., Setälä, H. & Bardgett, R. D. (2012) Land use alters the resistance and resilience of soil food webs to drought. *Nature Climate Change*, **2**, 276-280.
- De Vries, F. T. & Shade, A. (2013) Controls on soil microbial community stability under climate change. *Frontiers in Microbiology*, **4**, 265.
- Delgado-Baquerizo, M., Maestre, F. T., Reich, P. B., Trivedi, P., Osanai, Y., Liu, Y. R., Hamonts, K., Jeffries, T. C. & Singh, B. K. (2016) Carbon content and climate variability drive global soil bacterial diversity patterns. *Ecological Monographs*, **86**, 373-390.
- Emiola, A. & Oh, J. (2018) High throughput in situ metagenomic measurement of bacterial replication at ultra-low sequencing coverage. *Nature Communications*, **9**, 4956.
- Fierer, N. & Jackson, R. B. (2006) The diversity and biogeography of soil bacterial communities. *Proceedings of the National Academy of Sciences of the United States of America*, **103**, 626-631.
- Gao, Y. & Wu, M. (2023) Accounting for 16S rRNA copy number prediction uncertainty and its implications in bacterial diversity analyses. *ISME Communications*, **3**, 59.
- Kimmel, K., Avolio, M. L. & Ferraro, P. J. (2023) Empirical evidence of widespread exaggeration bias and selective reporting in ecology. *Nature Ecology & Evolution*, 15.
- Korem, T., Zeevi, D., Suez, J., Weinberger, A., Avnit-Sagi, T., Pompan-Lotan, M., Matot, E., Jona, G., Harmelin, A., Cohen, N., Sirota-Madi, A., Thaïss, C. A., Pevsner-Fischer, M., Sorek, R., Xavier, R. J.,

- Elinav, E. & Segal, E. (2015) Growth dynamics of gut microbiota in health and disease inferred from single metagenomic samples. *Science*, **349**, 1101-1106.
- Lavallee, J. M., Chomel, M., Alvarez Segura, N., de Castro, F., Goodall, T., Magilton, M., Rhymes, J. M., Delgado-Baquerizo, M., Griffiths, R. I., Baggs, E. M., Caruso, T., de Vries, F. T., Emmerson, M., Johnson, D. & Bardgett, R. D. (2024) Land management shapes drought responses of dominant soil microbial taxa across grasslands. *Nature Communications*, **15**, 29.
- Orwin, K. H. & Wardle, D. A. (2004) New indices for quantifying the resistance and resilience of soil biota to exogenous disturbances. *Soil Biology & Biochemistry*, **36**, 1907-1912.
- Pimm, S. L. (1984) The complexity and stability of ecosystems. *Nature*, **307**, 321-326.
- Rehshuh, R., Rehshuh, S., Gast, A., Jakab, A.-L., Lehmann, M. M., Saurer, M., Gessler, A. & Ruehr, N. K. (2022) Tree allocation dynamics beyond heat and hot drought stress reveal changes in carbon storage, belowground translocation and growth. *New Phytologist*, **233**, 687-704.
- Roller, B. R. K., Stoddard, S. F. & Schmidt, T. M. (2016) Exploiting rRNA operon copy number to investigate bacterial reproductive strategies. *Nature Microbiology*, **1**, 16160.
- Shade, A., Read, J. S., Welkie, D. G., Kratz, T. K., Wu, C. H. & McMahon, K. D. (2011) Resistance, resilience and recovery: aquatic bacterial dynamics after water column disturbance. *Environmental Microbiology*, **13**, 2752-2767.
- Skovgaard, O., Bak, M., Lobner-Olesen, A. & Tommerup, N. (2011) Genome-wide detection of chromosomal rearrangements, indels, and mutations in circular chromosomes by short read sequencing. *Genome Research*, **21**, 1388-1393.
- Sviličić, P., Vučetić, V., Filić, S. & Smolić, A. (2016) Soil temperature regime and vulnerability due to extreme soil temperatures in Croatia. *Theoretical and Applied Climatology*, **126**, 247-263.
- Tedersoo, L., Bahram, M., Põlme, S., Kõljalg, U., Yorou, N. S., Wijesundera, R., Ruiz, L. V., Vasco-Palacios, A. M., Thu, P. Q., Suija, A., Smith, M. E., Sharp, C., Saluveer, E., Saitta, A., Rosas, M., Riit, T., Ratkowsky, D., Pritsch, K., Põldmaa, K., Piepenbring, M., Phosri, C., Peterson, M., Parts, K., Pärtel, K., Otsing, E., Nouhra, E., Njouonkou, A. L., Nilsson, R. H., Morgado, L. N., Mayor, J., May, T. W., Majuakim, L., Lodge, D. J., Lee, S. S., Larsson, K.-H., Kohout, P., Hosaka, K., Hiiesalu, I., Henkel, T. W., Harend, H., Guo, L.-d., Greslebin, A., Grelet, G., Geml, J., Gates, G., Dunstan, W., Dunk, C., Drenkhan, R., Dearnaley, J., De Kesel, A., Dang, T., Chen, X., Buegger, F., Brearley, F. Q., Bonito, G., Anslan, S., Abell, S. & Abarenkov, K. (2014) Global diversity and geography of soil fungi. *Science*, **346**, 1256688.

Reviewer Reports on the First Revision:

Referees' comments:

Referee #1 (Remarks to the Author):

The resubmitted version of the manuscript provides some further detail and analysis. However, some of the major criticisms discussed in the first review have not been, to me, adequately addressed.

Ribosomal proteins are strikingly decreased in the heat-treated incubations according to the analysis presented. However I do not understand how this can be robustly interpreted to be a “down-regulation of protein biosynthesis and energy harvesting complexes under warming”. Most ribosomal proteins are always single copy, a fact which is used e.g. to assess the quality of metagenome-assembled genomes (see e.g. CheckM; Parks et. al. 2015, or Wu and Eisen 2008 10.1186/gb-2008-9-10-r151). Ribosomal rRNA may change its genomic copy number, as the authors state in the R2R, but ribosomal proteins do not. The analysis as stated appears to be based purely on a single result, recruiting reads to entire pathways with SAMSA/SEED. Extended Data Figure 17 only shows significantly over- or under-represented genes for brevity, but by eye I see a pattern that bacterial LSU bacterial genes appear to go down under heat, while the eukaryotic and archaeal LSU proteins go up. The same is true for SSU genes.

The R2R also highlights this observed pattern, but I contend that the pattern has been misinterpreted. Since LSU and SSU genes are usually single-copy in bacteria and archaea (whether or not they are lowering their expression of those genes in response to stress), this might indicate a change in the community profile to one more enriched in archaea or eukarya, which might mean that reference database bias towards particular species subverts the resulting read recruitment somehow. It does not indicate that individual cells are down-regulating their protein biosynthesis machinery. If my interpretation is correct, that bacteria comprise less of the community following heat treatment (or some other related explanation, like the average genome size of community members has increased), then these results do not indicate a change in protein biosynthesis regulation. Perhaps I am mistaken in this particular hypothesis, but in order to make conclusions about the “down-regulation of protein biosynthesis and energy harvesting complexes under warming”, alternative hypotheses such as these should be explored. In any case, insufficient evidence of gene gain or loss has occurred within individual lineages for these genes that have long been established to be single copy. I note also that the authors' interpretations appear to only be present in the R2R, rather than the manuscript itself. The new Extended Data Figure 17 provides some further detail in visual form, but no interpretation of this figure is provided (or of EDFig 18), and in particular there is no discussion about the above point that I could see in the manuscript itself.

Another comment on this sentence. Even if a function is less prevalent in metagenomic data, this is not likely to indicate a “down-regulation”, since DNA is being measured, not RNA. The only way I'm aware it might be possible to be “regulation” measurable through DNA abundance is if the microorganisms are actively changing their genomic copy number. No evidence is presented that suggests this. It is much

more likely that the abundance of individual community members are changing, rather than an evolution of their genomes.

A final comment on this sentence. It is unclear how “energy harvesting” is related. Do the authors mean that there is less protein biosynthesis, and therefore less activity from the cell, and therefore less energy harvesting happening? If so, this should be spelled out for the reader.

217: “perturbations in unseen soils (by cross-validation)”, and the R2R response regarding small proportion of variance explained: It is unclear whether the cross validation is amongst sites, or samples. By reading Extended Data 14’s legend, it appears it is the latter. The former would be preferable, because it would show the model generalises well. Looking at Extended Data 14b, at least by eye, it does not seem as though the model accurately models variation within sites, only between them, which is concerning if each site is represented in the training set, because it might be that the model is simply learning the set of sites, not climatic patterns. Given the current analysis, concluding that this random forest model can reliably predict resilience and resistance based on soil properties alone is unsupported. This concern could be addressed by performing a “grouped” k-fold cross validation, where each of the sites forms a group.

Minor comments:

“These particular disturbances were chosen to represent extremes that are increasing with climate change², and are considered extreme across all our sampled environments (Methods), though not necessarily comparable in their severity.” I was unclear about the meaning of the last part of this sentence. Do the authors mean the treatments are not comparable e.g. severity of the drought is not comparable to the severity of the heatwave? Or the severity of one treatment is not comparable between sites? In the former case, this appears to contradict some of the main findings presented e.g. in the abstract “Heat most strongly impacted soil microbiomes,”.

117: There is no such thing as a “metagenomic community”, or “metagenomic community structure”. As well, metagenomic data can be interpreted in many ways e.g. for functional genes, by genome recovery, or taxonomic composition. How it is interpreted here should be made clearer to the reader in the main text.

120: “Prokaryotic, fungal, and metagenomic communities were strongly shaped by their origin .. reflecting the importance of environmental factors in the distribution of soil microbial communities globally” This sentence could be rephrased, because some of the environmental factors are imposed by the incubations, and so are not shaped by the soil origin

131: “With the inclusion of a higher number of countries in our study, the proportion of variance explained by the disturbances decreased and became less variable”. This sentence could be better phrased. Do the authors mean that the proportion of variance explained became “less”?

819: The “a” should be bolded.

Extended Data Table 2 - some of the numbers have been line wrapped or not displayed correctly

Extended Data Figure 2 - figure did not appear correctly for me even using Chrome or Adobe Acrobat reader, there is no colours or ellipses for instance.

The R2R states that Kimmel et al. suggests that ecologists might have exaggerated effect sizes. I found that using this unconvincing as an argument to more fully appreciate effect sizes observed in this study, because it is quite vague and not specific to microbial ecology. However, Extended Data Figure 3 quantified this well.

169: shot-gun should be spelled shotgun.

The data has been deposited at the ENA, which is excellent. However, there was almost no metadata associated with these deposits. This means the analyses cannot be reproduced by others. In particular, it isn't clear which metagenomes (or, presumably, amplicon datasets - I did not look at these specifically) correspond to the different timepoints and treatments discussed in the manuscript. The initial samples are grassland metagenome samples (not just 'metagenome'), and should be associated with GPS coordinates in their associated BioSample attributes and ideally with other associated data where known. The time of incubation sampling and treatment applied should also be recorded. This is a fantastic, large dataset, so effort should be expended to make the data FAIR.

195: No figure or data is pointed to by this sentence, making it hard to interrogate. A measure of statistical significance / effect size may also be appropriate.

Extended data figure 9: "growth capacity" is not a widely used term, and should be explained.

Fig 4: Initial properties are clustered according 868 to the resistance-resilience profiles" I did not understand this sentence. Is this clustering which determines the order of them on the y-axis?

479: This section is before the growth rate methods, but the results are presented in the main text in the reverse order. It would be helpful to switch them. I would also suggest adding subheadings as the methods section is challenging to navigate as it is.

In many cases, the methods sections are insufficiently detailed. In particular the software used or version number was missing. For instance, what was used to generate the Random Forest model? What was used to determine the partial dependence? What version of SAMSA (and SEED) was used?

Extended Data 17 and 18: The labels are very small, which necessitates zooming in. The labels on the right hand side can probably be made bigger, and in order to help the reader inspect link the right and left hand sides it may be helpful to include horizontal lines between each coloured box (i.e. gene group) or some extra colouring, or just have a separate figure for each of the conditions.

Referee #2 (Remarks to the Author):

I feel that the authors have done an adequate job in responding to the extensive reviewer comments. This is ready to go. It is an interesting and valuable paper and will make a substantial contribution to the literature.

Referee #3 (Remarks to the Author):

The manuscript was much improved. The authors adequately address reviewers' comments. Clarifying the contrast in explanatory power at the global compared to local scale strengthened the paper. However, based on Extended data fig 14c it appears the model is only predictive of the heat response. Perhaps the graphs are misleading given the differences in the spread of the data. It would be useful to separate the predictive power for each disturbance for clarity.

L215 Moreover, we were able to construct a predictive model that explains $58 \pm 4\%$ of the variance in the response to all perturbations in unseen soils (by cross-validation) based only on initial soil properties and climate (Extended Data Fig. 14a).

Additional attention to the concluding paragraph will strengthen the paper.

L247 The response of functional microbial community composition was linked to soil functioning and could be predicted by the type of disturbance and soil and climatic properties.

Specific language would make this sentence more meaningful - functional gene composition was related to substrate-induced respiration and potential enzyme activity and

L 253 would be more impactful to state what the novel insights are - Our findings provide novel insight into the microbial community and environmental attributes

Author Rebuttals to First Revision:

Referees' comments:

Referee #1 (Remarks to the Author):

The resubmitted version of the manuscript provides some further detail and analysis. However, some of the major criticisms discussed in the first review have not been, to me, adequately addressed.

We thank the reviewer for assessing our revised manuscript in detail. We are happy that they feel that we provided further detail and analysis, and below, we provide a response to the outstanding comments of the reviewer.

Ribosomal proteins are strikingly decreased in the heat-treated incubations according to the analysis presented. However I do not understand how this can be robustly interpreted to be a “down-regulation of protein biosynthesis and energy harvesting complexes under warming”. Most ribosomal proteins are always single copy, a fact which is used e.g. to assess the quality of metagenome-assembled genomes (see e.g. CheckM; Parks et. al. 2015, or Wu and Eisen 2008 10.1186/gb-2008-9-10-r151). Ribosomal rRNA may change its genomic copy number, as the authors state in the R2R, but ribosomal proteins do not. The analysis as stated appears to be based purely on a single result, recruiting reads to entire pathways with SAMSA/SEED. Extended Data Figure 17 only shows significantly over- or under-represented genes for brevity, but by eye I see a pattern that bacterial LSU bacterial genes appear to go down under heat, while the eukaryotic and archaeal LSU proteins go up. The same is true for SSU genes.

The R2R also highlights this observed pattern, but I contend that the pattern has been misinterpreted. Since LSU and SSU genes are usually single-copy in bacteria and archaea (whether or not they are lowering their expression of those genes in response to stress), this might indicate a change in the community profile to one more enriched in archaea or eukarya, which might mean that reference database bias towards particular species subverts the resulting read recruitment somehow. It does not indicate that individual cells are down-regulating their protein biosynthesis machinery. If my interpretation is correct, that bacteria comprise less of the community following heat treatment (or some other related explanation, like the average genome size of community members has increased), then these results do not indicate a change in protein biosynthesis regulation. Perhaps I am mistaken in this particular hypothesis, but in order to make conclusions about the “down-regulation of protein biosynthesis and energy harvesting complexes under warming”, alternative hypotheses such as these should be explored. In any case, insufficient

evidence of gene gain or loss has occurred within individual lineages for these genes that have long been established to be single copy. I note also that the authors' interpretations appear to only be present in the R2R, rather than the manuscript itself. The new Extended Data Figure 17 provides some further detail in visual form, but no interpretation of this figure is provided (or of ED Fig 18), and in particular there is no discussion about the above point that I could see in the manuscript itself.

The reviewer's main point relates to concerns that we have interpreted decreases in overall "ribosomal protein genes" as being reflective of warming-induced change in protein biosynthesis regulation.

Firstly, we did not of course mean to infer that our metagenomic analysis says anything about expression being down-regulated in our experiments. The phrase picked up by the reviewer explicitly referred to another paper (Söllinger et al. 2022, reference 32) which is about regulation.

Secondly, we concede that on closer inspection all ribosomal genes in the Protein Metabolism categories of our metagenomic analysis were ribosomal proteins, which are for the most part single copy, and therefore even at the genomic level we cannot infer changes in the potential activities of differentially abundant taxa directly from this data.

We have therefore removed this statement about regulation and the citation referring to the changes in protein metabolism in light of the reviewer's concerns. Given this, what is happening specifically in the ribosomal proteins and the discussion of single copy genes (and the ubiquitous potential for database bias) is not relevant to the main text. However, we now refer in the main text to the additional figures where these patterns are visible. In addition we thoroughly address changes in realised and potential growth to illustrate the 'bust and boom' effect of our heat treatment. This includes a new analysis which estimates change in maximum growth potential from expected rRNA operon copy number. This is further supported by the large increase in dormancy and sporulation genes. The above points are all made in the rewritten paragraph at lines 189-213.

Another comment on this sentence. Even if a function is less prevalent in metagenomic data, this is not likely to indicate a "down-regulation", since DNA is being measured, not RNA. The only way I'm aware it might be possible to be "regulation" measurable through DNA abundance is if the microorganisms are actively changing their genomic copy number. No evidence is presented that suggests this. It is much more likely that the abundance of individual community members are changing, rather than an evolution of their genomes.

A final comment on this sentence. It is unclear how "energy harvesting" is related. Do the authors

mean that there is less protein biosynthesis, and therefore less activity from the cell, and therefore less energy harvesting happening? If so, this should be spelled out for the reader.

We trust that the changes described above address both comments on this sentence. Again, we did not mean to infer that our metagenomic analysis says anything about expression being down-regulated in our experiments, and we have now removed this statement to avoid any confusion around this, including any reference to energy-harvesting.

217: “perturbations in unseen soils (by cross-validation)“, and the R2R response regarding small proportion of variance explained: It is unclear whether the cross validation is amongst sites, or samples. By reading Extended Data 14’s legend, it appears it is the latter. The former would be preferable, because it would show the model generalises well. Looking at Extended Data 14b, at least by eye, it does not seem as though the model accurately models variation within sites, only between them, which is concerning if each site is represented in the training set, because it might be that the model is simply learning the set of sites, not climatic patterns. Given the current analysis, concluding that this random forest model can reliably predict resilience and resistance based on soil properties alone is unsupported. This concern could be addressed by performing a “grouped” k-fold cross validation, where each of the sites forms a group.

Apologies for any lack of clarity – the analysis in Fig. 14 does already represent ‘a “grouped” k-fold cross validation, where each of the sites forms a group’. All samples associated with each replicate site, that is 11 metagenomes for this analysis, constitutes a group which is either included in the training or the test set for the cross validation, but not both. We have now made this clear in the legend to ED Fig. 14.

Minor comments:

“These particular disturbances were chosen to represent extremes that are increasing with climate change², and are considered extreme across all our sampled environments (Methods), though not necessarily comparable in their severity.” I was unclear about the meaning of the last part of this sentence. Do the authors mean the treatments are not comparable e.g. severity of the drought is not comparable to the severity of the heatwave? Or the severity of one treatment is not comparable between

sites? In the former case, this appears to contradict some of the main findings presented e.g. in the abstract “Heat most strongly impacted soil microbiomes,”.

We apologise for this omission. The reviewer’s first explanation is indeed correct; this sentence refers to the fact that the treatments are not comparable with each other in terms of severity. We responded extensively to the reviewer’s comment on this during the previous round of reviews, and we replaced “heat” with “the heat treatment” (which was indeed most severe) throughout the manuscript to make it clear that it is not necessarily heat in general that most strongly affects soil microbial communities. However, we missed this instance in the abstract, which we have now corrected.

117: There is no such thing as a “metagenomic community”, or “metagenomic community structure”. As well, metagenomic data can be interpreted in many ways e.g. for functional genes, by genome recovery, or taxonomic composition. How it is interpreted here should be made clearer to the reader in the main text.

The reviewer is correct, we have replaced these phrases with “functional gene community”, in line 118 and line 123 (of the clean version)

120: “Prokaryotic, fungal, and metagenomic communities were strongly shaped by their origin .. reflecting the importance of environmental factors in the distribution of soil microbial communities globally”
This sentence could be rephrased, because some of the environmental factors are imposed by the incubations, and so are not shaped by the soil origin

We have replaced this phrase with “reflecting the importance of soil and climatic factors in the distribution of soil microbial communities globally”, now line 121-122 (clean version)

131: “With the inclusion of a higher number of countries in our study, the proportion of variance explained by the disturbances decreased and became less variable”. This sentence could be better phrased. Do the authors mean that the proportion of variance explained became “less”?

We mean both that the proportion of variance explained became less and that the variability among different included countries became less. We have now made this explicit in line 131-134 (clean version)

819: The “a” should be bolded.

Done

Extended Data Table 2 - some of the numbers have been line wrapped or not displayed correctly

The first column of R^2 values had been inadvertently narrowed – this has now been corrected.

Extended Data Figure 2 - figure did not appear correctly for me even using Chrome or Adobe Acrobat reader, there is no colours or ellipses for instance.

We apologise for this. We have now replaced this figure, which now hopefully displays correctly in the file.

The R2R states that Kimmel et al. suggests that ecologists might have exaggerated effect sizes. I found that using this unconvincing as an argument to more fully appreciate effect sizes observed in this study, because it is quite vague and not specific to microbial ecology. However, Extended Data Figure 3 quantified this well.

We are glad that the reviewer found our figure helpful in appreciating the deviation in local effects and global effects, and underlining that we only quantify this common response when including a large number of different sites.

169: shot-gun should be spelled shotgun.

Done

The data has been deposited at the ENA, which is excellent. However, there was almost no metadata associated with these deposits. This means the analyses cannot be reproduced by others. In particular, it isn't clear which metagenomes (or, presumably, amplicon datasets - I did not look at these specifically) correspond to the different timepoints and treatments discussed in the manuscript. The initial samples are grassland metagenome samples (not just 'metagenome'), and should be associated with GPS coordinates in their associated BioSample attributes and ideally with other associated data where known. The time of incubation sampling and treatment applied should also be recorded. This is a fantastic, large dataset, so effort should be expended to make the data FAIR.

The reviewer is incorrect that almost no metadata is associated with the deposits - We have included all mandatory fields for soil derived sequences, as defined by the Genomic Standards Consortium (GSC MlxS soil ERC000022: including collection date, environmental context, geographic location, elevation). The large amounts of experimental factors and other soil and functional data are inappropriate for the ENA and so they have been deposited in Figshare. Importantly, both the larger metadata files there contain links to all ENA sequence accession files, and therefore all data are openly available. Moreover, the Figshare doi contains all our code, thus allowing to fully reproduce all analyses. We therefore believe we have done all we are able to in order to make our dataset FAIR and reproducible, and we have modified the statement on this in the data availability section to make this clearer: "Derived data are available at <https://doi.org/10.6084/m9.figshare.22128512> with explicit accessions for the raw data at the ENA." (line 820-822)

195: No figure or data is pointed to by this sentence, making it hard to interrogate. A measure of statistical significance / effect size may also be appropriate.

"Together with a decrease in carbohydrate metabolism ($2.5 \pm 0.6 \times 10^{-05}$, $P = 2.0 \times 10^{-15}$, Fig. 3a)" - now added (now line 207 in the clean version). We have now also included an effect size to the earlier statement in that paragraph in line 190-191.

Extended data figure 9: "growth capacity" is not a widely used term, and should be explained.

We had indeed omitted the appropriate references validating our approach to estimating growth capacity. These references and further explanation have now been added to the main text and legend of ED Fig. 9 as well as increased consistency in use of the term throughout.

Fig 4: Initial properties are clustered according 868 to the resistance-resilience profiles” I did not understand this sentence. Is this clustering which determines the order of them on the y-axis?

Yes, clustering based on the correlations with resistance and resilience determines the order on the y-axis, we have now clarified this in the figure legend as follows: "The order of initial properties on the y-axis is determined by complete linkage clustering according to their correlations with resistance-resilience across perturbations"

479: This section is before the growth rate methods, but the results are presented in the main text in the reverse order. It would be helpful to switch them. I would also suggest adding subheadings as the methods section is challenging to navigate as it is.

We have rearranged as suggested and added three further subheadings.

In many cases, the methods sections are insufficiently detailed. In particular the software used or version number was missing. For instance, what was used to generate the Random Forest model? What was used to determine the partial dependence? What version of SAMSA (and SEED) was used?

We have added the Random Forest software, version numbers on the R packages for which these were missing, and more detail on the metagenomics annotation including version numbers (line 424-428 of the clean version).

Extended Data 17 and 18: The labels are very small, which necessitates zooming in. The labels on the right hand side can probably be made bigger, and in order to help the reader inspect link the right and left hand sides it may be helpful to include horizontal lines between each coloured box (i.e. gene group) or some extra colouring, or just have a separate figure for each of the conditions.

Yes, the labels are very small, which does necessitate zooming in, but enables comprehensiveness. We have opted to keep the different conditions together in the figure (both to

aid comparison and because the analyses represented did themselves include all treatments). Nonetheless, we have, as suggested, both added line guides and made the right-hand side labels substantially larger (wrapping the text to ensure nothing else is made smaller as a result), which we trust will help the navigation of these figures.

Referee #2 (Remarks to the Author):

I feel that the authors have done an adequate job in responding to the extensive reviewer comments. This is ready to go. It is an interesting and valuable paper and will make a substantial contribution to the literature.

We are delighted that the reviewer is happy with our revisions, and that they feel that our paper is important and will make a significant addition to the literature.

Referee #3 (Remarks to the Author):

The manuscript was much improved. The authors adequately address reviewers' comments. Clarifying the contrast in explanatory power at the global compared to local scale strengthened the paper. However, based on Extended data fig 14c it appears the model is only predictive of the heat response. Perhaps the graphs are misleading given the differences in the spread of the data. It would be useful to separate the predictive power for each disturbance for clarity.

The reviewer is correct that the majority of the predictive power in the model comes through its prediction of the heat response. We have now included individual predicted-observed correlations for each panel of Fig. 14a which makes this point clearer, but also illustrates that indeed, correlations are substantial for Drought and Freeze.

However, we infer the point about explanatory power from EDFig. 14a, NOT EDFig. 14c as the reviewer suggests – EDFig14b and EDFig14c DO show differential prediction in non-heat

treatments based on particular properties of climate (EDFig14b) and measured properties (EDFig14c). This helps justify our usage of the model as a whole.

L215 Moreover, we were able to construct a predictive model that explains $58 \pm 4\%$ of the variance in the response to all perturbations in unseen soils (by cross-validation) based only on initial soil properties and climate (Extended Data Fig. 14a).

Additional attention to the concluding paragraph will strengthen the paper.

L247 The response of functional microbial community composition was linked to soil functioning and could be predicted by the type of disturbance and soil and climatic properties.

Specific language would make this sentence more meaningful - functional gene composition was related to substrate-induced respiration and potential enzyme activity and

We thank the reviewer for their suggestions for making the conclusions stronger. We have now changed this sentence to more clearly state these relationships and also their implications, as follows: "The response of functional microbial community composition could be predicted by the type of disturbance and by soil and climatic properties, and was linked to enzyme activities and the ability to use different carbon sources, suggesting that extreme events may have consequences for soil carbon storage through their impacts on soil microbial communities^{6,43}" in line 253-257 (clean version)

L 253 would be more impactful to state what the novel insights are - Our findings provide novel insight into the microbial community and environmental attributes

We have now changed this sentence to make it clearer that we are referring to the insights we have just specified in the preceding sentences, as follows: "These novel insights into the microbial community and environmental attributes that determine soil microbial responses to extreme climatic events provide a first step for making general predictions of the impact of extreme climatic events on soil functioning.", line 261-264.

Reviewer Reports on the Second Revision:

Referees' comments:

Referee #1 (Remarks to the Author):

At the outset of this review, I apologise for its length. The manuscript covers a lot of ground, and when things are unclear to me, it makes a fully comprehensive review at the outset a challenging undertaking.

With the first major point of the previous review, around ribosomal protein copy number, my concerns have been addressed.

With regards to the cross-validation, the authors have clarified some points, but unfortunately am still slightly confused about the methodology. In part this is because of the non-standard approach the authors appear to employ (see below), and the fact that the cross-validation and analysis procedures used are described only briefly in the supplementary figure legend, rather than the more ample space in the materials and methods section. Ultimately my conclusion is that the cross-validation should be undertaken again, in a more standard, rigorous way, and that the conclusions drawn from the results are too broad even if the current accuracy holds up to this more rigorous analysis.

During my previous review, I suggested the grouped cross-validation should be done per-site, but I was mistaken. I meant actually that it should be done per-country. This is because as stated in the manuscript, "Prokaryotic, fungal, and functional gene communities were strongly shaped by their origin (PERMANOVA $R^2 = 0.72, 0.74$ and 0.68 , respectively, for country and site main effects". If sites in a country are broadly similar to each other, but different to other countries, then country would be the more appropriate grouping of samples.

EDFig 14: "This was carried out by a grouped 3-fold cross-validation approach in which the data was split into training and test sets where each test set contained data from two replicate sites in each country, and the test set contained data from the third site." I presume the authors mean "where each *training* set contained data from two replicate sites".

The figure legend states that “Which sites were selected in which countries was chosen randomly in each of 10,000 training-test splits of the data”. But a 3 fold cross validation is only run 3 times (once for each fold), so where does the 10,000 come from?

Leaving this 10,000 sentence aside, and assuming that 3-fold cross validation was used, the way that this cross-validation was carried out is not consistent with a standard application of grouped KFold cross validation. Usually (and what was the intention of the comment in my previous review), grouped cross-validation means that all samples from the group (here, all samples from each site were used, but should be country (above)) are either all in the training set, or all in the test set. The reason for doing this is to avoid a situation where there are very similar samples in the training and test sets, because in that situation measures of accuracy are likely to be inappropriately inflated, giving false confidence in the model.

If I am understanding properly the approach used here, then the groups weren't kept together in either the train or the test sets, but intentionally spread out some were in the training and some were in the test sets.

It was also not clear from the figure legend methods whether each training set only contained data from one country, or that the training and test sets had samples from multiple countries. I suppose the latter, but there are no methods and it is hard to trust short hand notations when non-standard approaches have been employed.

From the ED14 figure legend, “The coefficient of determination (R^2) across all subdivisions is $0.58 (\pm 0.04, \text{sd})$.”. By “subdivisions” do the authors mean the panels of the figure? So all data points from the 4 panels were plotted on the same graph of predicted vs actual (10,000 of each? Or just 1 example?) and then an R^2 was taken from that, and that was 0.58? If so, the conclusion that

“Moreover, we were able to construct a predictive model that explains $58 \pm 4\%$ of the variance in the response to all perturbations in unseen soils (by cross-validation) based only on initial soil properties and climate (Extended Data Fig. 14a).”

.. is very much unwarranted, because by the looks of ED14a, there is only a correlation for heat - the others appear to be no different from noise. It was also unclear where the 0.04 s.d. is derived from, since R^2 is a single measure - from amongst the 10,000 replicates? The reader is left to guess.

224: "A similarly distinct response to flood and drought was associated with soils' water holding capacity (Extended data Fig. 14c). " The drought plot in 14c doesn't appear to support this conclusion. I'm also a little confused about what the point of these analyses are. A model based on soil properties and climate to predict resistance, all of which were known from observed data. Then the MAT/WHC100 was then plotted against the resistance predicted by the model. What is the point of using a model in this instance, when the actual data is available?

64: Abstract: It is unclear what "emphasise that local responses do not predict global responses." means. Is this a reference to ED3? If so, I do not agree. Observing more variation when there are fewer sites included does not mean that changes observed within-site are not predictive, just that their explanatory power is overestimated.

Minor:

Abstract: "The heat treatment most strongly impacted soil microbiomes, through enhancing dormancy and sporulation genes and decreasing metabolic versatility". The word "through" suggests the effect is causative, when only a correlation has been observed. The word could just be removed.

Abstract: "The magnitude of this response" it is not quite clear what "this" refers to.

>>The data has been deposited at the ENA, which is excellent. However, there was almost no metadata associated with these deposits.

>The large amounts of experimental factors and other soil and

functional data are inappropriate for the ENA and so they have been deposited in Figshare.

I strongly disagree with this statement. There are some types of data which are beyond ENA biosample's scope, but metadata such as that in GSC MIXS soil ERC000022 is not. Including these data in ENA biosample entries is much more reusable than requiring a reader to parse an arbitrarily formatted figshare dataset (after first having to find the specific file in the figshare which contains the information), because the sequence data is tied to the metadata in a

systematic way, and because entry in official upload forms provides a level of validation that reduces the incidence of data entry errors.

Extended Data Figure 2: “Ellipses have areas proportional to the variance explained”. This is not strictly true, since creation of these diagrams presents an unsolvable (over-determined) problem when there are many numbers to represent. For instance some areas marked 0 have non-zero size. The 0s could probably be omitted for brevity as well.

Extended Data Figure 3: “Metagenomically” should not be capitalised.

ED9. Missing full stop after figure title

ED9: “Growth is shown”: growth by what measure?

205: My understanding from ED9a is that the maintext statement “There was also a striking relationship of the bacterial response to heat with that of fungi – the more fungi were impacted, the more positive was the bacterial growth response” refers to a positive slope in ED9a. But that is relative to control. Why is it taken here relative to flood and not the control? The figure in ED9a is too small to enable a proper inspection, but as it is, the pattern does not seem much different to noise.

Paragraph 189-213: There are many statements made which do not have associated statistical metrics.

207: “Together with a decrease in carbohydrate metabolism (change in proportion = $2.5 \pm 0.6 \times 10^{-5}$, $P = 2.0 \times 10^{-15}$, Fig. 3a), which suggests lower metabolic versatility”.

and

230: “We also found that functional community profiles were more resistant and resilient to heat with higher abundance of genes involved in dormancy and sporulation, as well as carbohydrate, protein, amino acid, phosphorus, and DNA metabolism, suggesting that higher

metabolic versatility allows microbes to survive adverse resource conditions (Extended Data Fig. 10)31. “

This sentence is hard to follow. “Functional community profiles” are not resistant or resilient, just the communities that encode them. I suppose the authors mean that communities encoding more “versatile” functions (i.e. dormancy and sporulation, as well as carbohydrate, protein, amino acid, phosphorus, and DNA metabolism) are more resilient/resistant.

Assuming I have understood the meaning of this sentence, there are a few issues. First, only the heat treatment has very many statistically significant associations. Second, it is unclear what the meaning of the term “versatile” is. There are just as many functional groupings which are negatively correlated with resistance/resilience, according to ED10. This is an important point in particular because it is mentioned in the abstract.

ED14b: “Freeze of Heat” should be “and”?

ED14b: How were the partial dependencies determined?

Referee #3 (Remarks to the Author):

The authors adequately address reviewers’ comments. The paper is acceptable for publication. Congratulations!

Author Rebuttals to Second Revision:

We thank the editor and referee #1 for their careful evaluation of our manuscript. We have now included the additional analysis as suggested by the editor and reviewer #1, which helps strengthen our conclusions. We have also addressed all other point raised by the referee. Please find below a point-by-point response (in blue italics) to the reviewer's comments, including references to the line numbers in the clean version of our manuscript (note that we have also uploaded a version of the main text with track changes).

Referees' comments:

Referee #1 (Remarks to the Author):

At the outset of this review, I apologise for its length. The manuscript covers a lot of ground, and when things are unclear to me, it makes a fully comprehensive review at the outset a challenging undertaking.

With the first major point of the previous review, around ribosomal protein copy number, my concerns have been addressed.

With regards to the cross-validation, the authors have clarified some points, but unfortunately am still slightly confused about the methodology. In part this is because of the non-standard approach the authors appear to employ (see below), and the fact that the cross-validation and analysis procedures used are described only briefly in the supplementary figure legend, rather than the more ample space in the materials and methods section. Ultimately my conclusion is that the cross-validation should be undertaken again, in a more standard, rigorous way, and that the conclusions drawn from the results are too broad even if the current accuracy holds up to this more rigorous analysis.

We respond to the details of this point below, but in summary, we do not believe that our approach was particularly non-standard, however we have now made that approach much clearer within the Methods section. We have at the same time redone all the cross-validation, including both the approach we had taken before (grouping by site) and the approach advocated by the reviewer below (grouping by country), clarifying the different insight that each approach gives and reporting both of them in ED Fig. 14.

During my previous review, I suggested the grouped cross-validation should be done per-site, but I was mistaken. I meant actually that it should be done per-country. This is because as stated in the manuscript, “Prokaryotic, fungal, and functional gene communities were strongly shaped by their origin (PERMANOVA $R^2 = 0.72, 0.74$ and 0.68 , respectively, for country and site main effects”. If sites in a country are broadly similar to each other, but different to other countries, then country would be the more appropriate grouping of samples.

The Permanova results the reviewer refers to do show both large Country main effects and substantial Site main effects, all of which are highly significant ($P \leq 0.001$) across all four measurements (Prokaryotes, fungi, metagenome and function; ED Table 2, the numbers quoted from the main text are the sum of Country and Site effects). This makes either Country or Site an a priori reasonable level for grouping in the cross-validation. However, each asks a slightly different question. As shown in Fig. 1a, most countries represent whole biogeographic regions. Part of our argument is that we need to cover this range of regions in order to have a realistic understanding of what is going on (ED Fig. 3) – we do not expect a good prediction for an alpine sample from a model that has never seen any alpine sites.

We have now included the cross-validation grouped by country as the reviewer suggests, and consistent with the point we just made, cross-validation grouped by country typically has lower explanatory power than cross-validation grouped by site (0.31 vs. 0.58) and only has any explanatory power at all in the Heat perturbation (now shown in ED Fig. 14d). We now make this point in the main text (line 230-234 in the clean version): “As with the effect our experimental perturbations on community composition (Extended data Fig. 3), this model relies on inclusion of a broad range of countries, representing different biogeographic regions (Fig. 1a): predicting resistance in whole unseen countries had relatively poor explanatory power, restricted to the heat perturbation, and was itself more variable (26 ± 20 % mean \pm sd Extended Data Fig. 14d)”

EDFig 14: “This was carried out by a grouped 3-fold cross-validation approach in which the data was split into training and test sets where each test set contained data from two replicate sites in each country, and the test set contained data from the third site.” I presume the authors mean “where each *training* set contained data from two replicate sites”.

Yes, apologies for the error, now corrected.

The figure legend states that “Which sites were selected in which countries was chosen randomly in each of 10,000 training-test splits of the data”. But a 3 fold cross validation is only run 3 times (once for each fold), so where does the 10,000 come from?

3-fold cross-validation does indeed imply a split of the data into three with three possible models, each with different assignments of training and test sets. However, there is a large number of such splits that is possible while maintaining the grouped structure ($3^8 = 6,561$ different splits into three, giving 19,683 model runs in this case). Of those possible splits, we previously did 10,000 model runs, chosen at random. To make the explanation in the methods clearer, we have now redone this cross-validation to include all 19,683 possible training-test splits. This does not affect the findings or appearance of ED Fig. 14

Leaving this 10,000 sentence aside, and assuming that 3-fold cross validation was used, the way that this cross-validation was carried out is not consistent with a standard application of grouped KFold cross validation. Usually (and what was the intention of the comment in my previous review), grouped cross-validation means that all samples from the group (here, all samples from each site were used, but should be country (above)) are either all in the training set, or all in the test set. The reason for doing this is to avoid a situation where there are very similar samples in the training and test sets, because in that situation measures of accuracy are likely to be inappropriately inflated, giving false confidence in the model.

We agree

If I am understanding properly the approach used here, then the groups weren't kept together in either the train or the test sets, but intentionally spread out some were in the training and some were in the test sets.

No, in any one split of the data, all samples from a particular group (in this case site, but potentially country as in our new analysis, above and ED Fig. 14d) were kept together in either the training or the test set, but not both (as previously mentioned in the ED Fig. 14 legend, but now expanded and clarified in the Methods)

It was also not clear from the figure legend methods whether each training set only contained

data from one country, or that the training and test sets had samples from multiple countries. I suppose the latter, but there are no methods and it is hard to trust short hand notations when non-standard approaches have been employed.

In each cross-validation model, all data were used, either in the training or test set. This has now been expanded on in the Methods (line 565-579 clean version).

From the ED14 figure legend, “The coefficient of determination (R^2) across all subdivisions is $0.58 (\pm 0.04, \text{sd})$.”. By “subdivisions” do the authors mean the panels of the figure? So all data points from the 4 panels were plotted on the same graph of predicted vs actual (10,000 of each? Or just 1 example?) and then an R^2 was taken from that, and that was 0.58? If so, the conclusion that

“Moreover, we were able to construct a predictive model that explains $58 \pm 4\%$ of the variance in the response to all perturbations in unseen soils (by cross-validation) based only on initial soil properties and climate (Extended Data Fig. 14a).”

.. is very much unwarranted, because by the looks of ED14a, there is only a correlation for heat - the others appear to be no different from noise. It was also unclear where the 0.04 s.d. is derived from, since R^2 is a single measure - from amongst the 10,000 replicates? The reader is left to guess.

“subdivisions” referred to subdivision into training and test sets (i.e. splitting; language now simplified to include only ‘split’) not perturbations (which are the figure panels in ED Fig. 14a). So for one such split, points across all panels (but from one model) gave a single R^2 value, on average, 0.58. Looking across 10,000 such splits, each with its own R^2 value, that became $0.58 (\pm 0.04, \text{sd})$.

The strikingly different explanatory power for the different perturbations is why we have included the correlation values for each perturbation separately in ED Fig. 14a. However, while the correlation is tight for heat (0.84) we strongly disagree that a correlation of 0.43 (for freeze) across 54 samples is ‘no different from noise’. We have now included P values to back up this point and clarified the wording of the associated text, highlighting the differences in prediction among perturbations. (line 225-234 clean version)

224: “A similarly distinct response to flood and drought was associated with soils’ water holding capacity (Extended data Fig. 14c). “ The drought plot in 14c doesn’t appear to support this

conclusion. I'm also a little confused about what the point of these analyses are. A model based on soil properties and climate to predict resistance, all of which were known from observed data. Then the MAT/WHC100 was then plotted against the resistance predicted by the model. What is the point of using a model in this instance, when the actual data is available?

*The question addressed by ED Fig. 14b-c is **how** the model predicts resistance. In ED Fig. 14b this demonstrates the contrasting role of mean annual temperature (MAT) in predicting resistance to freeze and heat – using the model to account for all the other variables, increased MAT predicts nonlinear reduction of resistance to freeze and increase of resistance to heat. This backs up the direct correlations for individual variables in the actual data, which are shown in Fig. 4, but do not account for other variables. This does reveal more – for example, there is no relationship seen between MAT and the resistance and resilience to Freeze in Fig. 4, but there is in ED Fig. 14b. To make these points clearer, we've now simplified ED Fig. 14b-c to focus on the dependence (rather than suggesting a comparison with observations) and changed ED Fig. 14c to show Moisture rather than WHC100, since the contrast in the marginal model prediction between Drought and Flood is clearer.*

64: Abstract: It is unclear what “emphasise that local responses do not predict global responses.” means. Is this a reference to ED3? If so, I do not agree. Observing more variation when there are fewer sites included does not mean that changes observed within-site are not predictive, just that their explanatory power is overestimated.

We do not claim that ED Fig. 3 shows that within-country variation in response to perturbations is not predictive of what occurs in those countries, or even that it is poorly estimated, only that it requires a larger study, as we have here, to identify the smaller amount of variation that is common to the perturbations themselves rather than the choice of biogeographic regions or countries. We have now replaced this sentence with “Our results suggest that soil microbiomes from different climates share unified responses to extreme climatic events, but that predicting the extent of community change requires knowledge of the local microbiome” (line 62-64), which is also supported by the new cross-validation analysis in ED Fig. 14d.

Minor:

Abstract: “The heat treatment most strongly impacted soil microbiomes, through enhancing dormancy and sporulation genes and decreasing metabolic versatility”. The word “through” suggests the effect is causative, when only a correlation has been observed. The word could

just be removed.

Done

Abstract: “The magnitude of this response” it is not quite clear what “this” refers to.

Clarified to ‘heat response’

>>The data has been deposited at the ENA, which is excellent. However, there was almost no metadata associated with these deposits.

>The large amounts of experimental factors and other soil and functional data are inappropriate for the ENA and so they have been deposited in Figshare.

I strongly disagree with this statement. There are some types of data which are beyond ENA biosample’s scope, but metadata such as that in GSC MiXs soil ERC000022 is not. Including these data in ENA biosample entries is much more reusable than requiring a reader to parse an arbitrarily formatted figshare dataset (after first having to find the specific file in the figshare which contains the information), because the sequence data is tied to the metadata in a systematic way, and because entry in official upload forms provides a level of validation that reduces the incidence of data entry errors.

We agree with the reviewer wholeheartedly on this point - ideally all metadata should be with the sequences. However, our initial reluctance to submit all metadata to the EBI was specifically due to an issue we have encountered before: namely that the metadata submissions do not accept NULL values for checklisted numeric fields. This is an issue for our dataset, because we collected a different set of variables for the initial samples (which represent the field situation) and the samples from the experimental phase, in which samples that represent the original field situation were subjected to the extreme events. We initially submitted these combined as one dataset, which resulted into a large number of NAs, and precluded submitting all metadata. In response to the reviewers request we have however found a workaround, which was to split the data (experimental v initial), and only submit those vectors containing actual measured data for each set. Examples of the modified submissions are here:

<https://www.ebi.ac.uk/biosamples/samples/SAMEA14371547> (an experimental sample)

<https://www.ebi.ac.uk/biosamples/samples/SAMEA14372156> (an initial analyses of field collected soil)

Extended Data Figure 2: “Ellipses have areas proportional to the variance explained”. This is not strictly true, since creation of these diagrams presents an unsolvable (over-determined) problem when there are many numbers to represent. For instance some areas marked 0 have non-zero size. The 0s could probably be omitted for brevity as well.

This is a fair point and we have adjusted the legend text accordingly, explicitly highlighting the issue. However, because the ellipses represent an imperfect representation of the variance explained, we have kept in the zeros applied to areas with a non-zero size, to be explicit that they have a size due to an inadequacy of the representation, not any other reason.

Extended Data Figure 3: “Metagenomically” should not be capitalised.

Changed

ED9. Missing full stop after figure title

added

ED9: “Growth is shown”: growth by what measure?

Legend rearranged to make this clear

205: My understanding from ED9a is that the maintext statement “There was also a striking relationship of the bacterial response to heat with that of fungi – the more fungi were impacted, the more positive was the bacterial growth response” refers to a positive slope in ED9a. But that is relative to control. Why is it taken here relative to flood and not the control?

The figure in ED9a is too small to enable a proper inspection, but as it is, the pattern does not seem much different to noise.

The size of the figure panels has been increased to enable clearer inspection. There is noise in this data, but the effect of fungal impact on bacterial growth, in interaction with perturbation is very significant – overall this is $P= 0.007$, which we now report in the main text. Previously we did not compare directly to control because the response being analysed is the difference of each sample to its respective control. However, just reporting the slope for that heat perturbation is equivalent to comparing to control (which is, by design, a slope of zero), which we now do in the main text instead of the heat-flood contrast.

Paragraph 189-213: There are many statements made which do not have associated statistical metrics.

Via the modifications above and further reporting we have introduced five new statistical metrics into this paragraph (line 188-215 clean version).

207: “Together with a decrease in carbohydrate metabolism (change in proportion = $2.5 \pm 0.6 \times 10^{-5}$, $P = 2.0 \times 10^{-15}$, Fig. 3a), which suggests lower metabolic versatility”.

and

230: “We also found that functional community profiles were more resistant and resilient to heat with higher abundance of genes involved in dormancy and sporulation, as well as carbohydrate, protein, amino acid, phosphorus, and DNA metabolism, suggesting that higher metabolic versatility allows microbes to survive adverse resource conditions (Extended Data Fig. 10)³¹. “

This sentence is hard to follow. “Functional community profiles” are not resistant or resilient, just the communities that encode them. I suppose the authors mean that communities encoding more “versatile” functions (i.e. dormancy and sporulation, as well as carbohydrate, protein, amino acid, phosphorus, and DNA metabolism) are more resilient/resistant.

Assuming I have understood the meaning of this sentence, there are a few issues. First, only the heat treatment has very many statistically significant associations. Second, it is unclear what the meaning of the term “versatile” is. There are just as many functional groupings which are negatively correlated with resistance/resilience, according to ED10. This is an important point in

particular because it is mentioned in the abstract.

Yes, this drop across a range of functional categories with heat shown in Fig. 3a was noted in the main text and highlighted in the summary as a drop in 'versatility'. As elsewhere (e.g. ED Fig. 14, above) there are associations with other perturbations too, but we focus on heat because there are more and more clear associations than with the other treatments.

The point at issue is that ED Fig. 10 shows that different metagenomically defined functions in the initial samples have different signs of association with resistance/resilience, suggesting that it might be less about 'versatility' of function than the relative abundance of specific copiotrophic functions. We agree and have changed the wording around ED Fig. 10 and its appearance in the summary to be more explicit and distinct from the overall versatility referred to in relation to Fig. 3a. (line 329-244 clean version)

ED14b: "Freeze of Heat" should be "and"?

Correct - this has been changed

ED14b: How were the partial dependencies determined?

These are determined with a very generic tool (pdp, as given in the Methods) which visualises the marginal effect of the variable, i.e. the relationship between that variable and the response of a model, while accounting for the average effect of the other predictors. This is potentially rather simply constructed, as described in the reference given, by taking the average prediction of the model on versions of the training data with the variable of interest set to each value in turn. We have now added some further description to the Methods at this point (line 577-579) including referencing the paper that originated the approach (Friedman 2001).

Referee #3 (Remarks to the Author):

The authors adequately address reviewers' comments. The paper is acceptable for publication.
Congratulations!

Reviewer Reports on the Third Revision:

Referees' comments:

Referee #1 (Remarks to the Author):

The reviewers have carried out a cross validation both grouped by site and grouped by country as request. However, I maintain that the approach is not the most standard. For instance, see the commonly used scikit-learn library: https://scikit-learn.org/stable/modules/generated/sklearn.model_selection.GroupKFold.html - there each sample is given a group (i.e. a site or country), and then the randomisation procedure chooses which samples are in test and train sets, under the constraint that samples from one group cannot be in both sets.

For the site-wise grouped KFold, this differs from the approach used in the manuscript, where the groupings guarantee that 2 sites from each country will be in the training set, and 1 site from the country will be in the test set. In contrast to the more standard approach, this means that the predictor is trained on data from each country (2 sites from each). In the standard approach, this may or may not be true, depending on the randomisation. I would suggest the authors change the wording to make it clear the method isn't standard e.g. "A *modified* grouped cross-validation was carried out.." or otherwise just do the cross-validation in the standard way.

The approach also differs in the sense that all possible partitions between test and train datasets have been assessed (19k of them). This deviation seems reasonable to me though, and probably provides a better assessment than the standard procedure (at the cost of increased computational load, which the authors have carried out).

I thank the authors for taking the time to do the country-wise cross validation. Indeed, models tested this way do not predict the test datasets. However, I would suggest the cross-validation procedure here is probably too strict. Instead the authors might do a leave-one-country-out analysis rather than leaving 4 countries out at a time - this would help the predictive accuracy by providing more (and more diverse) training data, which might improve accuracy. But I leave this as a more optional suggestion in this review.

>"Our results suggest that soil microbiomes from different climates share unified responses to extreme climatic events, but that predicting the extent of community change requires knowledge of the local microbiome" (line 62-64), which is also supported by the new cross-validation analysis in ED Fig. 14d.

I feel that this conclusion is too strong. The ED14d analysis is a lack of a strong result, not proof that the task is impossible. Perhaps the predictive accuracy could be improved by some methodology not attempted by the authors, some improvement to the cross-validation as suggested above, inclusion of some other non-microbiome data as input, or simply a larger dataset. Therefore the phrasing “predicting the extent of community change *may require* knowledge of the local microbiome” would be more appropriate.

The authors have adequately responded to my other previous comments.

Minor:

ED14: It would help to include both an R^2 and r values on the plots a and d - on first reading I thought perhaps by r the authors meant R^2 since that is the first mention of r in relation to these plots, but this isn't the case.

Author Rebuttals to Third Revision:

> We thank the referee for their thorough and positive assessment of our revised manuscript and for their final, helpful suggestions. We have now revised our manuscript and incorporated these final revisions. We provide a point-by-point response below (in italics).

Referee #1 (Remarks to the Author):

The reviewers have carried out a cross validation both grouped by site and grouped by country as request. However, I maintain that the approach is not the most standard. For instance, see the commonly used scikit-learn library: https://scikit-learn.org/stable/modules/generated/sklearn.model_selection.GroupKFold.html - there each sample is given a group (i.e. a site or country), and then the randomisation procedure chooses which samples are in test and train sets, under the constraint that samples from one group cannot be in both sets.

For the site-wise grouped KFold, this differs from the approach used in the manuscript, where the groupings guarantee that 2 sites from each country will be in the training set, and 1 site from the country will be in the test set. In contrast to the more standard approach, this means that the predictor is trained on data from each country (2 sites from each). In the standard approach, this may or may not be true, depending on the randomisation. I would suggest the authors change the wording to make it clear the method isn't standard e.g. "A *modified* grouped cross-validation was carried out.." or otherwise just do the cross-validation in the standard way.

> We have adapted the wording as suggested by the referee in line 783-786 (clean version) of the Methods section, which now reads: "A modified grouped cross-validation of this model was carried out in which the data was split into training and test sets where each training set contained data from two replicate sites in each country, and the test set contained data from the third site."

> We have in addition carried out another alternative cross-validation as the reviewer suggests is 'standard', i.e. without guaranteeing that 2 sites from each country are in the training set (only that no site appears in both test and training set, randomly assigning each site to training or test set with a 2:1 probability). This gives results very similar to those in the reported grouped cross-validation by site (e.g. r for Heat = 0.84 in each case). Nonetheless, in the cross-validation the reviewer describes as the standard version, grouped by site, all the data from a particular country could, by chance, be put in either training or test set, more like the cross-validation grouped by country. We feel this is potentially confusing and therefore prefer to stick with reporting the two cross-validations given as these more clearly contrast the approach of grouping by site versus grouping by country.

The approach also differs in the sense that all possible partitions between test and train datasets have been assessed (19k of them). This deviation seems reasonable to me though, and probably provides a better assessment than the standard procedure (at the cost of increased computational load, which the authors have carried out).

> We agree.

I thank the authors for taking the time to do the country-wise cross validation. Indeed, models tested this way do not predict the test datasets. However, I would suggest the cross-validation procedure here is probably too strict. Instead the authors might do a leave-one-country-out analysis rather than leaving 4 countries out at a time - this would help the predictive accuracy by providing more (and more diverse) training data, which might improve accuracy. But I leave this as a more optional suggestion in this review.

> We too wondered if leave-one-out cross-validation might improve accuracy. However, having performed this analysis, we find that the accuracy is very similar to leaving out 4 countries. We choose to report the version leaving out four countries because a) this is closer in proportions and therefore more comparable to the 3-fold cross-validation grouped by site already presented and b) leaving out 4 countries enables us to estimate an error for each country (leaving one out only leaves each country out once, whereas all different combinations of 4 countries leaves each country out multiple times, across which we estimate an error). We have now clarified this by stating these reasons for leaving out four countries at line 792-793 (clean version)

“Our results suggest that soil microbiomes from different climates share unified responses to extreme climatic events, but that predicting the extent of community change requires knowledge of the local microbiome” (line 62-64), which is also supported by the new cross-validation analysis in ED Fig. 14d.

I feel that this conclusion is too strong. The ED14d analysis is a lack of a strong result, not proof that the task is impossible. Perhaps the predictive accuracy could be improved by some methodology not attempted by the authors, some improvement to the cross-validation as suggested above, inclusion of some other non-microbiome data as input, or simply a larger dataset. Therefore the phrasing “predicting the extent of community change *may require* knowledge of the local microbiome” would be more appropriate.

> We agree, and we have changed the wording as suggested by the referee (line 64 in the clean version)

The authors have adequately responded to my other previous comments.

Minor:

ED14: It would help to include both an R^2 and r values on the plots a and d - on first reading I thought perhaps by r the authors meant R^2 since that is the first mention of r in relation to these plots, but this isn't the case.

> As suggested, we have now added a coefficient of determination (R^2) in each of these panels to minimise such potential confusion and aid interpretation as a proportion of variance explained.

Reviewer Reports on the Forth Revision:

Referees' comments:

Referee #1 (Remarks to the Author):

The authors have adequately responded to my previous review comments.